# Contextual Decision-Making with Knapsacks Beyond the Worst Case

**Zhaohua Chen**
School of Computer Science
Peking University
Haidian, Beijing, China
chenzhaohua@pku.edu.cn

**Rui Ai**
IDSS & LIDS
Massachusetts Institute of Technology
Cambridge, MA 02139, USA
ruiai@mit.edu

**Mingwei Yang**
Dept. of Management Science and Engineering
Stanford University
Stanford, CA 94305, USA
mwyang@stanford.edu

**Yuqi Pan**
School of Engineering and Applied Sciences
Harvard University
Cambridge, MA 02138, USA
yuqipan@g.harvard.edu

**Chang Wang**
Dept. of Computer Science
Northwestern University
Evanston, IL 60208, USA
wc@u.northwestern.edu

**Xiaotie Deng**
School of Computer Science
Institute for Artificial Intelligence
Peking University
Haidian, Beijing, China
xiaotie@pku.edu.cn

## Abstract

We study the framework of a dynamic decision-making scenario with resource constraints. In this framework, an agent, whose target is to maximize the total reward under the initial inventory, selects an action in each round upon observing a random request, leading to a reward and resource consumptions that are further associated with an unknown random external factor. While previous research has already established an $\widetilde{O}(\sqrt{T})$ worst-case regret for this problem, this work offers two results that go beyond the worst-case perspective: one for the worst-case gap between benchmarks and another for logarithmic regret rates. We first show that an $\Omega(\sqrt{T})$ distance between the commonly used fluid benchmark and the online optimum is unavoidable when the former has a degenerate optimal solution. On the algorithmic side, we merge the re-solving heuristic with distribution estimation skills and propose an algorithm that achieves an $\widetilde{O}(1)$ regret as long as the fluid LP has a unique and non-degenerate solution. Furthermore, we prove that our algorithm maintains a near-optimal $\widetilde{O}(\sqrt{T})$ regret even in the worst cases and extend these results to the setting where the request and external factor are continuous. Regarding information structure, our regret results are obtained under two feedback models, respectively, where the algorithm accesses the external factor at the end of each round and at the end of a round only when a non-null action is executed.

## 1 Introduction

In online contextual decision-making problems with knapsack constraints (CDMK for short), an agent is required to make sequential decisions over a finite time horizon to maximize the accumulated

38th Conference on Neural Information Processing Systems (NeurIPS 2024).

reward under initial resource constraints [14, 15]. To be more specific, in each round $t = 1, \ldots, T$, a request $\theta_t$ and an external factor $\gamma_t$ are independently generated from two distributions, and only $\theta_t$ is revealed to the agent. Based on the request, the agent should irrevocably choose an action $a_t$, which results in a reward $r(\theta_t, a_t, \gamma_t)$ and a consumption vector $\boldsymbol{c}(\theta_t, a_t, \gamma_t)$ of resources. The agent's target is to optimize the sum of rewards $\sum_{t=1}^{T} r(\theta_t, a_t, \gamma_t)$ before the resources are depleted.

The contextual decision-making with knapsacks problem presents two key challenges when compared to closely related problems (e.g., the network revenue management problem): (1) choices are made without observing the *external factor*, and (2) distributions of requests and external factors are *unknown*. However, the complexity of CDMK makes it a suitable mathematical abstraction for many real-life scenarios, and there are extensive application scenarios with this kind of information structure. We use the following examples as illustrations and motivation.

**Example 1.1** (Supply chain management). In supply chain management, a factory needs to allocate among $T$ repositories consecutively and is constrained by the total inventory. Given the request $\theta_t$ for each repository, the factory chooses the number of goods transported to it as action $a_t$. However, the factory has randomized transportation costs for different locations and traffic conditions denoted by an external factor $\gamma_t$ for the $t$-th repository, which finally influences rewards. The factory needs to form an optimal scheme to allocate its goods under uncertainty to all these repositories.

**Example 1.2** (Dynamic bidding in repeated auctions with budgets [9, 10]). In this circumstance, an advertiser acquires the value of the ad slot $\theta_t$ at the start of each auction and chooses a bid $a_t$ accordingly. The agent's gain in this auction, as a consequence, is collaboratively determined by the value, the bid, and the highest competing bid $\gamma_t$, and has a form of $\theta_t \mathbb{1}(a_t > \gamma_t)$. Additionally, the payment is $a_t \mathbb{1}(a_t > \gamma_t)$ for the first-price auction and $\gamma_t \mathbb{1}(a_t > \gamma_t)$ for the second-price auction, respectively. It is to be noted that the highest competing bid is inaccessible to the agent before committing to the bid, as all advertisers bid simultaneously. Meanwhile, its distribution is decided by other advertisers, which is also unknown to the agent before the auctions.

The CDMK model can also capture other well-studied problems, including multi-secretary, online linear programming, online matching, etc., as discussed in Balseiro et al. [9]. Previous studies of the CDMK problem have shown that the worst-case regret is $\widetilde{O}(\sqrt{T})$ when the initial resources are linearly proportional to the horizon length $T$ [36, 24]. However, it is still unclear whether we can achieve a better regret guarantee for the CDMK problem beyond worst-case scenarios. In particular, can we design algorithms to obtain an $o(\sqrt{T})$ regret only under mild assumptions that hold for almost all possible CDMK instances? Meanwhile, can these algorithms still obtain good regret guarantees even in the worst cases? At last, previous works would adopt specific benchmarks to measure the regret of algorithms, but how are these benchmarks close to the rewards that the optimal online algorithm can achieve? This work widely addresses these questions.

## 1.1 Our Contributions

This work makes three main contributions, summarized as follows.

**The fluid optimum can be $\Omega(\sqrt{T})$ away from the online optimum.** Since the online optimum is hard to characterize, previous works always use an alternative benchmark to measure the performance of any online algorithm, and the fluid optimum (also known as the deterministic LP) is a common choice [24, 35]. However, we demonstrate that when the fluid benchmark has a unique and degenerate solution, then an $\Omega(\sqrt{T})$ gap is unavoidable between these two optima (cf. Theorem 2.1). While Han et al. [24] has also provided a similar lower bound result for the related contextual bandits with knapsacks (CBwK) problem, their condition depends on the inseparability of the possible expected reward/consumption function set. In other words, their condition may not perform well when this feasible set is small. Furthermore, their condition is rather complicated to verify. In contrast, our condition only depends on the underlying problem instance and is concise and easy to check. The proof of our result extends the approach of Vera and Banerjee [38] to the CDMK problem.

**An $\widetilde{O}(1)$ regret via re-solving under mild assumptions with full/partial information feedback.** Since an $\widetilde{O}(\sqrt{T})$ worst-case regret is already known [36], we investigate how well an online algorithm can perform beyond worst cases by applying the re-solving heuristic in conjunction with distribution estimation techniques, as given in Algorithm 1. This method has been considered in the problems

Table 1: A summary of our algorithmic results on Algorithm 1. Constants are omitted.

| | Beyond the Worst Case | Worst Case | |
| | Uniq., Non-Degen. LP | Discrete | Continuous |
|---|---|---|---|
| Full-Info. | $O(1)$ | $O(\sqrt{T\log T})$ | $O((T^{\alpha_u} + T^{\alpha_v} + T^{1/2})\sqrt{\log T})$ |
| Part.-Info. | $O(\log T)$ | $O(\sqrt{T}\log T)$ | $O((T^{\alpha_u} + T^{1/2})\sqrt{\log T} + T^{\alpha_v}(\log T)^{3/2-\alpha_v})$ |

$u, v$: the mass/density function of the context and the external factor.
$\alpha_{p\in\{u,v\}}$: $(\beta + d)/(2\beta + d)$ if $p$ is a $d$-dimension distribution and $p \in \Sigma(\beta, L)$. (See Appendix A.)

of network revenue management (NRM) and bandits with knapsacks (BwK). (See Section 1.2 for a literature review.) However, to our knowledge, we are the first to extend this method to the CDMK problem, which poses new challenges as decisions should be made according to the request. To avoid worst cases, we explicitly suppose that the fluid problem has a unique and non-degenerate solution (cf. Assumption 3.1). This assumption is mild in three aspects: (1) it captures almost all CDMK problem instances, as slightly perturbing any LP can help it satisfy the unique optimality and non-degeneracy conditions; (2) it is less restrictive than the assumptions given in Sankararaman and Slivkins [32], which require that there are at most two resources; and (3) it is almost necessary for an $o(\sqrt{T})$ regret bound to be established by Theorem 2.1, when using the fluid optimum as the benchmark. Under the assumption, our main results show that the re-solving heuristic reaches an $O(1)$ regret with full information (cf. Theorem 3.1) and an $O(\log T)$ regret with partial information (cf. Theorem 4.1). To our knowledge, these are the first $\widetilde{O}(1)$ regret results in the CDMK problem beyond the worst case with only mild assumptions. Importantly, unlike previous results, these regret bounds are also independent of the number of actions.

Within our results, the full information model assumes that the agent sees the external factor at the end of each round. In contrast, in the partial information model, the agent acquires the external factor only when a non-null action is adopted. In Example 1.1, if the factory can learn the road condition via map services, it then observes the external factor no matter its chosen action, reflecting the full information feedback. However, it can sometimes only observe transportation costs when it transports goods, resembling a non-null action. This is a case of partial information feedback. In the auction market illustrated in Example 1.2, agents might also face these two kinds of information models. In some situations, bidders can always view others' bids after the auction, while in other cases, only those who bid non-zero values can observe others' bids. Non-zero bidding here reflects a non-null action. Therefore, these two information models hold strong practical significance.

Other state-of-the-art results consider bandit information feedback, in which the agent only sees the reward and the consumption rather than the external factor. However, they explicitly assume a specific (e.g., linear) relationship between the conditional expected reward-consumption pair and the request [3, 32, 24, 36], whereas our results do not impose any underlying distribution structures, bypassing realizability issues [24]. On this side, our information model is comparable to those in existing work.

**A near-optimal regret even in worst cases with full/partial information feedback and an extension to continuous randomness.** We further explore how well our Algorithm 1 performs even in worst-case scenarios. With full information feedback, we show that an $O(\sqrt{T\log T})$ regret is achieved (cf. Theorem 5.1). This bound is asymptotically equal to the state-of-the-art with this information model [24, 36]. Even with partial information, we can still guarantee a universal $O(\sqrt{T}\log T)$ regret (cf. Theorem 5.2), which is optimal up to a logarithmic factor. These results demonstrate the applicability and robustness of the re-solving heuristic in CDMK problems, regardless of some specific instances. For completeness, we extend our algorithm and analysis to the situation in which the distributions of request and external factor are continuous and derive corresponding regret results (cf. Theorems A.1 and A.2).

We summarize our algorithmic results on Algorithm 1 in Table 1.

## 1.2 Related Work

**Contextual decision-making/bandits with knapsacks.** The issue most closely related to the CDMK problem is the problem of contextual bandits with knapsacks (CBwK), introduced by Agrawal and Devanur [3]. The main difference between the CDMK problem and the CBwK problem is that in the latter, an explicit model of the external factor is missed, and the bandit information feedback is considered. That is, only the consumption and the reward are revealed to the agent at the end of a round rather than the external factor. Along this research line, two primary methodologies have been proposed to solve the problem. The first approach aims to select the best probabilistic strategy within the policy set [8], and Agrawal et al. [4] adopts this approach to achieve an $\widetilde{O}(\sqrt{T})$ regret. This heuristic originates from the subject of contextual bandits [17, 2], and requires a cost-sensitive classification oracle to achieve computation efficiency.

On the other hand, another approach views the problem from the perspective of the Lagrangian dual space. It uses a dual update method that reduces the CBwK problem to the online convex optimization (OCO) problem. In particular, some work [3, 32, 35, 30] assumes a linear relationship between the conditional expectation of the reward-consumption pair and the request-action pair. This line adopts techniques for estimating linear function classes [1, 6, 34, 18] and combines them with OCO methods to achieve sub-linear regret.

Apart from the above studies, some results [24, 36, 37] are not restricted to linear expectation functions. To deal with more general problems with bandit feedback, they plug model-reliable online regression methods [22, 21] into the dual update framework. As a result, their algorithms' regret is the sum of the regret on online regression and online convex optimization, respectively. Nevertheless, the online regression technique still limits the conditionally expected reward-consumption functions.

In the CDMK literature, Liu and Grigas [30] have considered full information feedback, where the agent sees the external factor at the end of each round. Motivated by practice, our work further considers a partial feedback model, in which the agent observes the external factor when a non-null action is chosen (cf. Section 2).

**The re-solving heuristic and related problems.** Unlike the above approaches, our work adopts the re-solving method, also known as the "certainty equivalence" (CE) heuristic. Under this approach, the agent (in)frequently solves the fluid optimization problem with the remaining resources to obtain a probability control in each round. This method comes from the literature on the network revenue management (NRM) problem, which can be seen as a simplification of the CDMK problem without the existence of external factors or the external factor not getting involved in the resource consumption [42]. Some researches in this setting also assumes known request distributions [26, 5, 25, 19, 13, 28, 16, 11, 39, 12, 27]. These works show that the re-solving-based method can obtain a constant regret under certain non-degeneracy assumptions and can generally obtain a square-root regret [16]. Recently, the re-solving method is also extended to the general dynamic resource-constrained reward collection problem in Balseiro et al. [9], which assumes the knowledge of request and external factor distributions and achieves $O(1)$ to $O(\log T)$ regret for different action space cardinalities.

We should mention that the re-solving technique, together with other methods, has also been adopted for the bandits with knapsacks (BwK) problem [23, 20, 42, 39, 29, 32] to achieve an $O(\log T)$ regret under different assumptions. For example, an essential result by Sankararaman and Slivkins [32] achieves $O(\log T)$ regret in BwK under the best-arm assumption and two resources. However, CDMK is a more challenging problem than BwK in that the decision has to be based on the received request. Thus, no optimal static action mode is irrelevant to the round, which adds a layer of complexity to the re-solving method.

## 2 Preliminaries

We consider an agent interacting with the environment for $T$ rounds. There are $n$ kinds of resources, with an average amount of $\rho^i$ for resource $i$ in each round, resulting in a total of $\rho^i T$ amount of resource $i$. We suppose that $0 < \boldsymbol{\rho} = \boldsymbol{\rho}_1 = (\rho^i)_{i \in [n]} \leq \mathbf{1}$ is independent of $T$, with a maximum entry of $\rho^{\max} \leq 1$ and a minimum entry of $\rho^{\min} > 0$.

At the beginning of each round $t \geq 1$, the agent observes a request $\theta_t \in \Theta$ drawn i.i.d. from a distribution $\mathcal{U}$ and should choose an action $a_t$ from a set of actions $A$. Given the request $\theta_t$ and the action $a_t$, the agent receives a random reward $r_t \in [0, 1]$ and a consumption vector of resources $\boldsymbol{c}_t \in [0, 1]^n$, both of which are related to an external factor $\gamma_t \in \Gamma$ drawn i.i.d. from a distribution $\mathcal{V}$. In other words, there is a reward function $r : \Theta \times A \times \Gamma \rightarrow [0, 1]$ and a consumption vector function $\boldsymbol{c} : \Theta \times A \times \Gamma \rightarrow [0, 1]^n$, such that $r_t = r(\theta_t, a_t, \gamma_t)$ and $\boldsymbol{c}_t = \boldsymbol{c}(\theta_t, a_t, \gamma_t)$. We suppose these two functions are pre-known to the agent. We further define $R(\theta, a) := \mathbb{E}_\gamma[r(\theta, a, \gamma)]$, and $\boldsymbol{C}(\theta, a) := \mathbb{E}_\gamma[\boldsymbol{c}(\theta, a, \gamma)]$.

We impose minimum restrictions on the distributions $\mathcal{U}$ and $\mathcal{V}$. In the main body of this work, we only suppose that the support sets of both distributions are finite. Specifically, we let $k = |\Theta|$ be the size of the request set. We denote the mass function of $\mathcal{U}$ and $\mathcal{V}$ by $u(\theta)$ and $v(\gamma)$, respectively. We will extend to the situation that these two distributions can be continuous in Appendix A.

The agent's objective is to maximize the cumulative rewards over the period under initial resource constraints, which is a sequential decision-making problem. To ensure feasibility, we assume the existence of a null action (denoted by 0) in the action set $A$. Under the null action, the reward and the consumption of any resource are zero, regardless of the request and the external factor. In other words, we have $r(\theta_t, 0, \gamma_t) = 0$ and $\boldsymbol{c}(\theta_t, 0, \gamma_t) = \boldsymbol{0}$ for any $(\theta_t, \gamma_t) \in \Theta \times \Gamma$. We use $A^+ := A \setminus \{0\}$ to denote the set of non-null actions and let $m := |A^+|$ be its size.

We would like to discuss here the necessity of the null action, which is widely used in related works [8, 3, 4, 36]. An alternative common choice for the null action is the so-called "early stop when resource exhausted" [32] in the BwK problem with no contexts. In reality, when the agent faces some "bad" contexts, a better choice is not "entering the market" to avoid, for example, small rewards but large consumption. As a comparison, struggling to come up with a non-null action here could occupy the space for serving those "good" contexts, and stopping before these contexts arrive may inevitably cause an $\Omega(T)$ regret. This illustrates that introducing a null action is necessary in the CDMK problem. In fact, in this problem, contexts play the role of revealing information and deterring unreasonable deals.

We consider the set of *non-anticipating* strategies $\Pi$. In particular, let $\mathcal{H}_t$ be the history the agent could access at the start of round $t$. Then, for any non-anticipating strategy $\pi \in \Pi$, $a_t$ should depend only on $(\theta_t, \mathcal{H}_t)$, that is, $a_t = a_t^\pi(\theta_t, \mathcal{H}_t)$. For abbreviation, we write $a_t^\pi = a_t^\pi(\theta_t, \mathcal{H}_t)$ when there is no confusion.

Therefore, we can define the agent's optimization problem as below:

$$V^{\mathrm{ON}} := \max_{\pi \in \Pi} \mathbb{E}_{\boldsymbol{\theta} \sim \mathcal{U}^T, \boldsymbol{\gamma} \sim \mathcal{V}^T} \left[ \sum_{t=1}^T r(\theta_t, a_t^\pi, \gamma_t) \right], \quad \text{s.t.} \quad \sum_{t=1}^T \boldsymbol{c}(\theta_t, a_t^\pi, \gamma_t) \leq \boldsymbol{\rho} T, \ \forall \boldsymbol{\theta} \in \Theta^T, \boldsymbol{\gamma} \in \Gamma^T.$$

**The fluid benchmark.** In practice, however, computing the expected reward of the optimal online strategy would require solving a high-dimension (probably infinite) dynamic programming, which is intractable. Hence, we turn to consider the fluid benchmark to measure the performance of a strategy, which is defined as follows:

$$V^{\mathrm{FL}} := T \cdot \max_{\phi : \Theta \times A^+ \rightarrow \mathbb{R}} \mathbb{E}_{\theta \sim \mathcal{U}} \left[ \sum_{a \in A^+} R(\theta, a) \phi(\theta, a) \right],$$

$$\text{s.t.} \ \mathbb{E}_{\theta \sim \mathcal{U}} \left[ \sum_{a \in A^+} \boldsymbol{C}(\theta, a) \phi(\theta, a) \right] \leq \boldsymbol{\rho}; \ \sum_{a \in A^+} \phi(\theta, a) \leq 1, \forall \theta \in \Theta; \ \phi(\theta, a) \geq 0, \forall (\theta, a) \in \Theta \times A^+.$$

For a better understanding, $V^{\mathrm{FL}}$ reflects the maximum expected total rewards an agent can win when a static strategy is adopted and the resource constraints are only to be satisfied in expectation. Therefore, this optimization problem is a linear program in which the decision variable $\phi(\theta, a)$ represents the probability that the agent chooses action $a$ upon seeing request $\theta$. It is a well-known result that $V^{\mathrm{FL}}$ gives an upper bound on $V^{\mathrm{ON}}$.

**Proposition 2.1** (Balseiro et al. [9]). $V^{\mathrm{FL}} \geq V^{\mathrm{ON}}$.

Thus, we evaluate the performance of a non-anticipating strategy $\pi$ by comparing its expected accumulated reward $Rew^\pi$ with the fluid benchmark $V^{\mathrm{FL}}$, which is a common choice in literature [35,

24]. However, we prove that such a benchmark choice may lead to a $\Omega(\sqrt{T})$ gap as long as $V^{\mathrm{FL}}$ is degenerate.

**Theorem 2.1** (Worst-case gap). *When $V^{\mathrm{FL}}$ has a unique and degenerate optimal solution, $V^{\mathrm{FL}} - V^{\mathrm{ON}} = \Omega(\sqrt{T})$.*

Despite the worst-case lower bound, we prove in this work that for any CDMK instance in which $V^{\mathrm{FL}}$ has a *unique non-degenerate* optimal solution (cf. Assumption 3.1), we can obtain an $\widetilde{O}(1)$ gap compared to the fluid benchmark. Thus, it is still a good choice in most cases.

**Information feedback model.** In this work, we consider two types of information feedback models, with increasing difficulty obtaining a sample of the external factor $\gamma$.

- [Full information feedback.] The agent is able to observe $\gamma_t$ at the end of each round $t$.
- [Partial information feedback.] The agent can observe $\gamma_t$ at the end of round $t$ *only if $a_t \neq 0$.*

In general, with full information feedback, the agent can observe an i.i.d. sample from $\mathcal{V}$ each round, which is the optimal scenario for learning the distribution. Nevertheless, such an assumption could be strong since the reward and consumption vector are irrelevant to the external factor when the agent chooses the null action $a = 0$. Thereby, a more realistic information model is the partial feedback one, where the external factor is only accessible when $a \neq 0$. This limitation also increases the difficulty of learning the distribution $\mathcal{V}$ since the agent observes fewer samples under this model than under full information feedback. It is important to note that the partial information model represents a transition from full to bandit information feedback, under which only the reward and consumption vector are accessible in each round, rather than the external factor. Real-life instances of partial information feedback include Examples 1.1 and 1.2, as we have discussed in the introduction.

## 3 The Re-Solving Heuristic

In this work, we introduce the re-solving heuristic to the CDMK problem. The resulting algorithm is presented in Algorithm 1.

To briefly describe the algorithm, we start by defining an optimization problem that captures the optimal fluid control for each round, assuming complete knowledge of $\mathcal{U}$ and $\mathcal{V}$. For any $\boldsymbol{\kappa} \in [0,1]^n$, we define $J(\boldsymbol{\kappa})$ as the following optimization problem:

$$J(\boldsymbol{\kappa}) := \max_{\phi:\Theta \times A^+ \to \mathbb{R}} \mathbb{E}_{\theta \sim \mathcal{U}} \left[ \sum_{a \in A^+} R(\theta, a)\phi(\theta, a) \right],$$

$$\text{s.t. } \mathbb{E}_{\theta \sim \mathcal{U}} \left[ \sum_{a \in A^+} \boldsymbol{C}(\theta, a)\phi(\theta, a) \right] \leq \boldsymbol{\kappa}; \ \sum_{a \in A^+} \phi(\theta, a) \leq 1, \forall \theta \in \Theta; \ \phi(\theta, a) \geq 0, \forall (\theta, a) \in \Theta \times A^+.$$

Evidently, we have $V^{\mathrm{FL}} = T \cdot J(\boldsymbol{\rho}) = T \cdot J(\boldsymbol{\rho}_1)$ by definition. Intuitively, in each round $t$, the best fluid choice of the agent is given by the optimal solution $\phi_t^*$ of LP $J(\boldsymbol{\rho}_t)$, where $\boldsymbol{\rho}_t$ is the average budget of the remaining rounds, including round $t$. Nevertheless, since full knowledge of the exact distributions $\mathcal{U}$ and $\mathcal{V}$ is lacking, the agent can only solve an estimated programming $\widehat{J}(\boldsymbol{\rho}_t, \mathcal{H}_t)$ as outlined in Algorithm 1, with the following realization:

$$\widehat{J}(\boldsymbol{\rho}_t, \mathcal{H}_t) := \max_{\phi:\Theta \times A^+ \to \mathbb{R}} \mathbb{E}_{\theta \sim \widehat{\mathcal{U}}_t} \left[ \sum_{a \in A^+} \mathbb{E}_{\gamma \sim \widehat{\mathcal{V}}_t} \left[ r(\theta, a, \gamma) \right] \phi(\theta, a) \right],$$

$$\text{s.t. } \mathbb{E}_{\widehat{\mathcal{U}}_t} \left[ \sum_{a \in A^+} \mathbb{E}_{\widehat{\mathcal{V}}_t} \left[ c(\theta, a, \gamma) \right] \phi(\theta, a) \right] \leq \boldsymbol{\rho}_t; \ \sum_{a \in A^+} \phi(\theta, a) \leq 1, \forall \theta \in \Theta; \ \phi(\theta, a) \geq 0, \forall (\theta, a) \in \Theta \times A^+.$$

Here, $\widehat{\mathcal{U}}_t$ and $\widehat{\mathcal{V}}_t$ represent the empirical distribution of $\theta$ and $\gamma$, respectively, according to the sample history given by $\mathcal{H}_t$. Specifically, let $\mathcal{I}_t$ be the set of rounds that the agent accesses the external factor. The mass functions of these two estimated distributions are standard as follows: $\widehat{u}_t(\theta) := \#[\theta \text{ appears in previous } t-1 \text{ rounds}]/t-1; \ \widehat{v}_t(\gamma) := \#[\gamma \text{ appears in rounds in } \mathcal{I}_t]/|\mathcal{I}_t|.$

---

**Algorithm 1** Re-Solving with Empirical Estimation.

---
**Input:** $\boldsymbol{\rho}, T$.
**Initialization:** $\mathcal{I}_1 \leftarrow \emptyset$, $\boldsymbol{B}_1 \leftarrow \boldsymbol{\rho} T$.
**for** $t = 1$ **to** $T$ **do**
    Observe $\theta_t$;
    $\boldsymbol{\rho}_t \leftarrow \boldsymbol{B}_t / (T - t + 1)$;
    $\widehat{\phi}_t^* \leftarrow$ the solution to $\widehat{J}(\boldsymbol{\rho}_t, \mathcal{H}_t)$;
    Choose $a_t \in A$ randomly such that for $a \in A^+$, $\Pr[a_t = a] = \widehat{\phi}_t^*(\theta_t, a)$, and $\Pr[a_t = 0] = 1 - \sum_{a \in A^+} \widehat{\phi}_t^*(\theta_t, a)$;
    **if** (*FULL-INFO*) $\vee$ (*PARTIAL-INFO* $\wedge$ $a_t \neq 0$) **then**
        Observe $\boldsymbol{\gamma}_t$;
        $\mathcal{I}_{t+1} \leftarrow \mathcal{I}_t \cup \{t\}$;
    **else**
        $\mathcal{I}_{t+1} \leftarrow \mathcal{I}_t$;
    **end if**
    $\boldsymbol{B}_{t+1} \leftarrow \boldsymbol{B}_t - \boldsymbol{c}_t$;
    **if** $\boldsymbol{B}_{t+1}^i < 1$ for some $i \in [n]$ **then**
        **break**;
    **end if**
**end for**

---

It is worth noting that the estimated distribution of $\theta$, $\widehat{\mathcal{U}}_t$, is always based on $t - 1$ samples since the agent received an independent sample from $\mathcal{U}$ at the beginning of each round. On the other hand, the empirical distribution of the external factor $\gamma$, $\widehat{\mathcal{V}}_t$, is estimated from $|\mathcal{I}_t|$ independent samples. With full information feedback, $|\mathcal{I}_t| = t - 1$; whereas with partial information feedback, $|\mathcal{I}_t| \leq t - 1$ equals the number of times the agent chooses an action $a \neq 0$ before round $t$. For brevity, for the estimated programming, we write $\widehat{\boldsymbol{C}}_t(\theta, a) := \mathbb{E}_{\gamma \sim \widehat{\mathcal{V}}_t}[\boldsymbol{c}(\theta, a, \gamma)]$ and $\widehat{R}_t(\theta, a) := \mathbb{E}_{\gamma \sim \widehat{\mathcal{V}}_t}[r(\theta, a, \gamma)]$. As per Algorithm 1, the agent's decision mode in round $t$ is given by the optimal solution $\widehat{\phi}_t^*$ of programming $\widehat{J}(\boldsymbol{\rho}_t, \mathcal{H}_t)$. The algorithm stops when the resources are near depletion, that is, $\boldsymbol{B}^i < 1$ for some resource $i \in [n]$, and we use $T_0$ to denote the stopping time of Algorithm 1, i.e., $T_0 := \min\{T, \min\{t : \exists i \in [n], \boldsymbol{B}_{t+1}^i < 1\}\}$.

For an analysis beyond the worst-case scenario, a crucial assumption we will make is that the fluid problem possesses good regularity properties, i.e., it is an LP with a unique and non-degenerate solution.

**Assumption 3.1.** The optimal solution to $J(\boldsymbol{\rho}_1)$ is unique and non-degenerate.

As pointed out by Bumpensanti and Wang [13], uniqueness and non-degeneracy are a critical factor for an $o(\sqrt{T})$ regret bound to hold in the CDMK problem, at least for the frequent re-solving technique we use in this work [26]. Intuitively, if $J(\boldsymbol{\rho}_1)$ is degenerate, then with any minor error on the estimation, the optimal solution to $\widehat{J}(\boldsymbol{\rho}_t, \mathcal{H}_t)$ can have a major different landscape with the optimal solution to $J(\boldsymbol{\rho}_1)$ in the sense of basic variables and binding constraints, and this will lead to an $O(\sqrt{T})$ accumulated regret. For completeness, we formally define the above concepts.

**Definition 3.1.** A context-action pair $(\theta, a)$ is a *basic variable* for $J(\boldsymbol{\rho}_1)$ if $\phi_1^*(\theta, a) > 0$, or else, it is a non-basic variable. Similarly, define basic/non-basic variables for $\widehat{J}(\boldsymbol{\rho}_t, \mathcal{H}_t)$.

**Definition 3.2.** $i \in [n]$ is a *binding constraint* for $J(\boldsymbol{\rho}_1)$ if $\sum_{\theta \in \Theta, a \in A^+} u(\theta) \boldsymbol{C}^i(\theta, a) \phi_1^*(\theta, a) = \boldsymbol{\rho}_1^i$, or else it is a non-binding constraint. We let $\mathcal{S} := \{i$ is a binding constraint for $J(\boldsymbol{\rho}_1)\}$, $\mathcal{T} = [n] \setminus \mathcal{S}$, and we use $\boldsymbol{\kappa}|_{\mathcal{S}}$ or $\boldsymbol{\kappa}|_{\mathcal{T}}$ to define the sub-vector of $\boldsymbol{\kappa}$ confined on $\mathcal{S}$ or $\mathcal{T}$, respectively. Further, $\theta \in \Theta$ is a binding constraint for $J(\boldsymbol{\rho}_1)$ if $\sum_{a \in A^+} \phi_1^*(\theta, a) = 1$, or else it is a non-binding constraint. Similarly, define binding/non-binding constraints for $\widehat{J}(\boldsymbol{\rho}_t, \mathcal{H}_t)$.

As stated above, we want to guarantee that when the "distance" between $\widehat{J}(\boldsymbol{\rho}_t, \mathcal{H}_t)$ and $J(\boldsymbol{\rho}_1)$ is sufficiently small, the optimal solution to these two programmings have the same landscapes. In this sense, we consider a *stability factor* $D$ to measure such a threshold, as presented by Mangasarian and Shiau [31].

**Proposition 3.1** (Stability). *Under Assumption 3.1, there is a maximum $D > 0$, such that when the following holds:*

$$\max\left\{\|(u(\theta) - \widehat{u}_t(\theta))_{\theta \in \Theta}\|_\infty, \|(v(\gamma) - \widehat{v}_t(\gamma))_{\gamma \in \Gamma}\|_1\right\} \leq D,$$
$$\max\left\{\|\boldsymbol{\rho}_1|_{\mathcal{S}} - \boldsymbol{\rho}_t|_{\mathcal{S}}\|_\infty, \max\{\boldsymbol{\rho}_1|_{\mathcal{T}} - \boldsymbol{\rho}_t|_{\mathcal{T}}\}\right\} \leq D, \tag{1}$$

*$J(\boldsymbol{\rho}_1)$ and $\widehat{J}(\boldsymbol{\rho}_t, \mathcal{H}_t)$ share the same sets of basic/non-basic variables and binding/non-binding constraints.*

With the assumption, below we present the main result of this work, which is proved in Appendix C.1.

**Theorem 3.1.** *Under Assumption 3.1, with full information feedback, the expected accumulated reward $Rew$ brought by Algorithm 1 when $T \to \infty$ satisfies:*

$$V^{\mathrm{FL}} - Rew = O\left(\frac{n^2 + k}{D^2}\right),$$

*which is independent of $T$.*

The intuition behind Theorem 3.1 is to conduct a regret decomposition in a Lagrangian manner, motivated by Chen et al. [16]. This leads to three remaining terms (cf. Appendix C). For the first two Lagrangian product terms, thanks to Proposition 3.1, they equal 0 as long as the estimates of the distributions are sufficiently accurate with an error of $O(D)$, which will happen with high probability after a constant number of rounds. The last term reflects how the stopping time of Algorithm 1 is close to the total time-span $T$. On this front, we are left to demonstrate that the resources are spent smoothly. Intuitively, this property is guaranteed by combining two observations: (1) In each round, the action mode ensures that resources are spent evenly in expectation in the estimation world due to the re-solving step, and (2) the distance between the estimation world and the real world diminishes to zero, with the accumulation of samples. The complete reasoning is much more detailed.

We now compare Theorem 3.1 with results in prior work. We first mention that our benchmark $V^{\mathrm{FL}}$ is larger than the benchmark used in Slivkins and Foster [36] and Slivkins et al. [37], as proved in Appendix B. Thus, our result provides a stronger regret upper bound. As for the constants in the regret bound, first, our regret does not involve $m$ explicitly. This is superior to existing results, which report an $\widetilde{O}(\sqrt{m})$ reliance [8, 4, 36, 24]. As an intuitive reason, the number of actions does not explicitly appear in our Algorithm 1, but only contributes to the dimension of the linear program. However, $m$ could appear in some complex and problem-specific constants that we omit in the bound. Interested readers can refer to Appendix C for more details. Second, although $k$ does not always appear in previous works, this is inevitable in our bound, brought by the estimation error of the context distribution. Third, for the BwK problem, the well-known $O(\log T)$ result given by Sankararaman and Slivkins [32] supposes that $n \leq 2$, and it is still unclear whether their analysis can be extended to an arbitrary number of resources. Our result does not suffer from such a limit. Finally, we remark that in the absence of resource constraints, $D$ is precisely half the gap between the mean rewards of the best and second-best arms. Thus, $D$ resembles the *reward-gap-like parameter* in the multi-armed bandit literature. The dependence on $D$ of our result is similar to the *first* result in Sankararaman and Slivkins [32] and is the same with Chen et al. [16], and it is still unclear whether the dependence can be improved. We should also note that we omit the dependence of our regret bound on the unknown size of the external factor set in all our results, which could be improved via parameterized estimation techniques.

One key implication of Theorem 3.1 is that the re-solving heuristic's regret is independent of the number of rounds beyond the worst-case with full information. This result significantly improved over previous state-of-the-art results under mild assumptions, surpassing the solutions proposed by Han et al. [24] and Slivkins and Foster [36]. In particular, their solutions come from the BwK literature and rely on dual update and upper confidence bound (UCB) heuristics, which only provide a worst-case regret of $O(\sqrt{T \log T})$.

## 4 Partial Information Feedback

We now shift to consider the re-solving method's performance with partial information feedback, under which the agent only sees the external factor $\gamma_t$ when her choice is non-null in round $t$, i.e.,

$a_t \neq 0$. Apparently, with less information, the learning speed of the distribution $\mathcal{V}$ decreases, hindering the re-solving procedure's quick convergence to an optimal solution. Nevertheless, we demonstrate that the performance of the re-solving method only faces an $O(\log T)$ multiplicative degradation under partial information feedback. Our primary theorem in this section is as follows:

**Theorem 4.1.** *Under Assumption 3.1, with partial information feedback, the expected accumulated reward Rew brought by Algorithm 1 when $T \to \infty$ satisfies:*

$$V^{\mathrm{FL}} - Rew = O\left(\frac{n^2 + k + \log T}{D^2}\right).$$

Before we come to the technical parts, we first place Theorem 4.1 within the literature. As previously mentioned, $\Omega(\sqrt{T})$ is a worst-case lower bound on the regret even with full information feedback and thus also extends as a lower bound with partial information feedback. However, Theorem 4.1 steps beyond the worst case by providing an $O(\log T)$ upper bound for regular problem instances. This result outperforms the universal $O(\sqrt{T \log T})$ regret by Han et al. [24] and Slivkins and Foster [36]. While the result is asymptotically equivalent to that of Sankararaman and Slivkins [32], it imposes fewer restrictions on the problem structure, as previously discussed.

We now provide an intuitive understanding of the proof of Theorem 4.1. The crux lies in analyzing the frequency with which Algorithm 1 can access an independent sample of the external factor. To this end, we use $Y_t = |\mathcal{I}_t| \leq t - 1$ to denote the number of times a non-null action is chosen before time $t$, or equivalently, the number of i.i.d. samples from $\mathcal{V}$ observed by the agent before time $t$ under partial information feedback. The following crucial technical lemma provides a lower bound on $Y_t$.

**Lemma 4.1.** *There is a constant $0 < C_b < 1/2$, such that with probability $1 - O(1/T)$, the following hold for Algorithm 1:*

1. *For any $\Theta(\log T) \leq t \leq C_b \cdot T$, $Y_t \geq C_f \cdot (t-1)/\log T$ for some constant $C_f$;*

2. *For any $t > C_b \cdot T$, $Y_t \geq C_r \cdot T$ for some constant $C_r$.*

The proof of Lemma 4.1 is deferred to Appendix D.2. In simple terms, during the initial $\Theta(\log T)$ rounds (the shaded segment), the re-solving method cannot guarantee the accessing frequency since the learning of the request distribution $\mathcal{U}$ has yet to converge sufficiently. However, after $\Theta(\log T)$ rounds, Algorithm 1 ensures a constant probability of obtaining a new example in each round, provided that the remaining resources are sufficient. As a consequence, before $\Theta(T)$ rounds, we can guarantee an $\Omega(1/\log T)$ accessing frequency at any time step and an overall $\Omega(1)$ frequency with high probability, as established by a concentration inequality. The remaining proof of Theorem 4.1 is provided in Appendix D.1.

## 5 Relaxing the Regularity Assumption – A Worst-Case Guarantee

In Sections 3 and 4, we have proved that Algorithm 1 can achieve an $\widetilde{O}(1)$ regret for CDMK problems under full and partial information feedbacks, assuming certain regular conditions (cf. Assumption 3.1). Put differently, the re-solving heuristic nicely deals with regular scenarios. In this section, we complement the above by showing that this method can attain nearly optimal regret in the worst cases. Furthermore, in Appendix A, we extend our analysis to cases where the context and external factor distributions can be continuous.

Our main results are given below, and their proofs are provided in Appendices E.1 and E.2, respectively.

**Theorem 5.1.** *With full information feedback, the expected accumulated reward Rew brought by Algorithm 1 satisfies: $V^{\mathrm{FL}} - Rew = O(k\sqrt{T \log T} + n)$, as $T \to \infty$.*

**Theorem 5.2.** *With partial information feedback, the expected accumulated reward Rew brought by Algorithm 1 satisfies: $V^{\mathrm{FL}} - Rew = O(k\sqrt{T} \log T + n)$, as $T \to \infty$.*

As given by Theorem 2.1, the worst-case regret of any online CDMK algorithm is $\Omega(\sqrt{T})$, while Theorems 5.1 and 5.2 indicate that the re-solving heuristic reaches near-optimality in such cases. Further, state-of-the-art algorithms [24, 36] can at most obtain an $\widetilde{O}(\sqrt{T})$ regret even with full/partial information feedback. Our algorithm also achieves this near-optimal regret bound in worst cases.

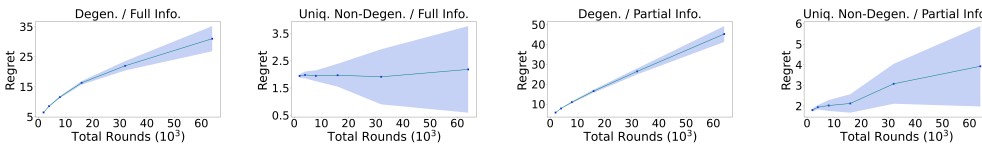

Figure 1: Regret of Algorithm 1 under different number of total rounds $T = 2000 \cdot 2^k$ for integer $0 \le k \le 5$.

It is worth noticing that we omit some problem-specific constants in the previous bounds, e.g., the fluid optimum $J(\boldsymbol{\rho}_1)$, which could be related to the number of non-null arms $m$. Therefore, our results do not conflict with the well-known $\Omega(\sqrt{mT})$ regret lower bound as given by Auer et al. [7].

## 6    Numerical Validations

In this section, we use numerical experiments to verify our analysis. We perform our simulation experiments with either full or partial information feedback under two cases. The first is with a degenerate optimal solution, and the second is with a unique and non-degenerate optimal solution. We delay more details, including the choice of the problem instances, to Appendix F.

Figure 1 describes the relationship between the regret and the number of total rounds $T$ under all four settings. We set the horizon $T$ to be $2000 \cdot 2^k$ for integer $0 \le k \le 5$. The figure displays both the sample mean (the line) and the 99%-confidence interval (the light color zone) calculated by the results of 50 estimations for the regret, where each estimation comprises 400 independent trials. Observe that when the LP $J(\boldsymbol{\rho})$ is degenerate, the regret grows on the order of $\tilde{O}(\sqrt{T})$ under both full information and partial information settings. Further, when the underlying LP $J(\boldsymbol{\rho})$ has a unique and non-degenerate optimal solution, the regret does not scale with $T$ under the full information setting. In the partial information setting, the regret slowly grows with $T$, which matches our $\widetilde{O}(1)$ theoretical guarantee.

## 7    Concluding Remarks

This work establishes the effectiveness of the re-solving heuristic in the contextual decision-making problem with knapsack constraints. We first prove that the gap between the fluid optimum and online optimum is $\Omega(\sqrt{T})$ when the fluid LP has a unique and degenerate optimal solution. Further, we show that the re-solving method reaches an $O(1)$ regret with full information and an $O(\log T)$ regret with partial information when the fluid LP has a unique and non-degenerate optimal solution, even compared to the fluid benchmark. Considering the sufficient condition for the $\Omega(\sqrt{T})$ lower bound, our non-degeneracy assumption is mild, especially when comparing with the two-resource condition required in Sankararaman and Slivkins [32].

Further, we show that even in worst cases, the re-solving method achieves an $O(\sqrt{T \log T})$ regret with full information feedback and an $O(\sqrt{T} \log T)$ regret with partial information feedback. These results are comparable to start-of-the-art results [24, 36]. We also extend our analysis to the continuous randomness case for completeness.

## Acknowledgement

This work is supported by the National Natural Science Foundation of China (Grant No. 62172012), by Alibaba Group through Alibaba Innovative Research Program, and by Peking University-Alimama Joint Laboratory of AI Innovation. Part of this work was done when Rui Ai, Mingwei Yang, Yuqi Pan, and Chang Wang were undergraduates in Peking University. The authors thank Yinyu Ye for his kind suggestions and all anonymous reviewers for their helpful feedback.

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

# Appendix of "Contextual Decision-Making with Knapsacks Beyond the Worst Case"

We begin by outlining the structure of the whole appendix. In Appendix A, we state our regret results with continuous randomness. In Appendix B, we prove the regret worst-case result (cf. Theorem 2.1). Appendix C.1 devotes to proving the main result (cf. Theorem 3.1), where Appendix C.1.1 presents the Lagrangian regret decomposition, and Appendices C.1.2 and C.1.3 analyze different terms in the decomposition. Appendices C.2 to C.6 complement missing proofs of lemmas arising in Appendix C.1. Appendix D focuses on proving the regret results in the partial information setting, where Appendix D.1 derives Theorem 4.1, and Appendix D.2 proves the crucial Lemma 4.1 for the partial feedback model. Appendices D.3 and D.4 prove lemmas arising in previous parts of Appendix D. Appendix E deals with Theorems 5.1 and 5.2, which show that the re-solving method is near-optimal even in worst cases. At last, Appendix G complements the missing details of Appendix A with continuous randomness.

## A    From Discrete Randomness to Continuous Randomness

In the main body of this work, we explicitly assume that both the context set and the external factor set are discrete. Such an assumption can suitably capture most real-life situations. For example, in an agent's online bidding problem with budget constraints, if we presume that the context is the agent's actual value and the external factor is the highest competing bid, it is natural to suppose that all these three values are discrete. Nevertheless, for theoretical completeness, we expand our results in this section to circumstances where these two sets are infinite, i.e., the two underlying randomnesses are continuous. It is imperative to note that the scenario where one randomness is discrete and the other is continuous would be analogous in analysis by incorporating the techniques presented in Section 5.

Conceptually, the re-solving heuristic still works: we solve the optimization problem in each round concerning the remaining resources based on previous estimates. However, technically, since the distributions of context and external factors are continuous, we should further elaborate on the setting. In this section, we suppose that the context set $\Theta = [0,1]^{d_u}$ and the external factor set $\Gamma = [0,1]^{d_v}$. We denote $u(\theta)$ and $v(\gamma)$ as the density function of $\mathcal{U}$ and $\mathcal{V}$, respectively. We assume that $p \in \{u, v\}$ belongs to the $\beta_p$-order $L_p$-Hölder smooth class $\Sigma(\beta_p, L_p)$. Here, for the foundation, given a vector $s = (s_1, ..., s_d)$, define

$$|s| = s_1 + \cdots + s_d, \quad D^s = \frac{\partial^{s_1 + \cdots + s_d}}{\partial x_1^{s_1} \cdots \partial x_d^{s_d}}.$$

Subsequently, for a positive integer $\beta$, the $\beta$-order $L$-Hölder smooth class is defined as

$$\Sigma(\beta, L) := \{g : |D^s g(x) - D^s g(y)| \leq L\|x - y\|_2, \text{ for all } s \text{ such that } |s| = \beta - 1, \text{ and all } x, y\}.$$

Now, suppose $X_1, \cdots, X_k$ are $k$ i.i.d. samples from a distribution with density function $p \in \Sigma(\beta, L)$. According to Wasserman [40], we have the following result, which implies that we can calculate an estimator from these samples that converges to the density function.

**Proposition A.1** (Wasserman [40]). *Suppose $X_1, \cdots, X_k$ are drawn i.i.d. from a $d$-dimension distribution $\mathcal{P}$, with density $p \in \Sigma(\beta, L)$ for some $L > 0$, and $k$ is sufficiently large. Then there exists an estimator $\widehat{p}_k$ such that for any $\epsilon > 0$,*

$$\Pr\left[\sup_x |p(x) - \widehat{p}_k(x)| > \frac{C\sqrt{\log(k/\epsilon)}}{k^{\beta/(2\beta+d)}}\right] \leq \epsilon,$$

*with $C$ a constant.*

The details of constructing such a density estimator are postponed to Appendix G.1. We now return to the re-solving heuristic and Algorithm 1. In the algorithm, with continuous randomness, the constrained optimization problem to be solved in each round $\widehat{J}(\boldsymbol{\rho}_t, \mathcal{H}_t)$ for $t = 1, 2, \cdots$ becomes:

$$\widehat{J}(\boldsymbol{\rho}_t, \mathcal{H}_t) := \max_{\phi:\Theta \times A^+ \to \mathbb{R}} \int_\theta \sum_{a \in A^+} \phi(\theta, a) \int_\gamma r(\theta, a, \gamma) \widehat{v}_t(\gamma) \widehat{u}_t(\theta) \, \mathrm{d}\gamma \, \mathrm{d}\theta,$$

$$\text{s.t.} \quad \int_\theta \sum_{a\in A^+} \phi(\theta,a) \int_\gamma \boldsymbol{c}(\theta,a,\gamma)\widehat{v}_t(\gamma)\widehat{u}_t(\theta)\,\mathrm{d}\gamma\,\mathrm{d}\theta \le \boldsymbol{\rho}_t,$$

$$\sum_{a\in A^+} \phi(\theta,a) \le 1, \quad \forall\theta\in\Theta,$$

$$\phi(\theta,a) \ge 0, \quad \forall(\theta,a)\in\Theta\times A^+.$$

Correspondingly, the reference optimization problem $J(\boldsymbol{\rho}_t)$ is given below:

$$J(\boldsymbol{\rho}_t) := \max_{\phi:\Theta\times A^+\to\mathbb{R}} \int_\theta \sum_{a\in A^+} \phi(\theta,a) \int_\gamma r(\theta,a,\gamma)v(\gamma)u(\theta)\,\mathrm{d}\gamma\,\mathrm{d}\theta,$$

$$\text{s.t.} \quad \int_\theta \sum_{a\in A^+} \phi(\theta,a) \int_\gamma \boldsymbol{c}(\theta,a,\gamma)v(\gamma)u(\theta)\,\mathrm{d}\gamma\,\mathrm{d}\theta \le \boldsymbol{\rho}_t,$$

$$\sum_{a\in A^+} \phi(\theta,a) \le 1, \quad \forall\theta\in\Theta,$$

$$\phi(\theta,a) \ge 0, \quad \forall(\theta,a)\in\Theta\times A^+.$$

At this point, it is worth mentioning that solving $\widehat{J}(\boldsymbol{\rho}_t, \mathcal{H}_t)$ in each round could be hard as it could be a continuous yet non-convex constrained optimization problem. Nevertheless, we assume the existence of an oracle that aids us in solving this optimization, and we focus on the regret of the re-solving method. Let $\alpha_u := (\beta_u + d_u)/(2\beta_u + d_u)$ and $\alpha_v := (\beta_v + d_v)/(2\beta_v + d_v)$, and we have the following two results, respectively, under full and partial information feedback.

**Theorem A.1.** *Under continuous randomness, with full information feedback, the expected accumulated reward Rew brought by Algorithm 1 satisfies:*

$$V^{\mathrm{FL}} - Rew = O((T^{\alpha_u} + T^{\alpha_v} + T^{1/2})\sqrt{\log T} + n), \quad T \to \infty.$$

**Theorem A.2.** *Under continuous randomness, with partial information feedback, the expected accumulated reward Rew brought by Algorithm 1 satisfies:*

$$V^{\mathrm{FL}} - Rew = O((T^{\alpha_u} + T^{1/2})\sqrt{\log T} + T^{\alpha_v}\log^{3/2-\alpha_v} T + n), \quad T \to \infty.$$

The proofs of the above theorems are presented in Appendices G.2 and G.3, respectively, which almost follow the threads of Theorems 5.1 and 5.2.

## B   Specifying the Worst-Case Gap – Proof of Theorem 2.1

To prove the lemma, we first introduce an intermediate value, which we denote as $V^{\mathrm{Hyb}}$, to upper bound $V^{\mathrm{ON}}$, and show that the gap between $V^{\mathrm{Hyb}}$ and $V^{\mathrm{FL}}$ is $O(\sqrt{T})$ under the given condition. Specifically, we have the following definition:

$$V^{\mathrm{Hyb}} := \mathbb{E}_{\theta_1,\cdots,\theta_T}\left[ \max_{\phi_1,\cdots,\phi_T:A^+\to\mathbb{R}} \sum_{t=1}^{T} \sum_{a\in A^+} R(\theta_t,a)\phi_t(a) \right],$$

$$\text{s.t.} \quad \sum_{t=1}^{T} \sum_{a\in A^+} \boldsymbol{C}(\theta_t,a)\phi_t(a) \le \boldsymbol{\rho}T, \tag{2}$$

$$\sum_{a\in A^+} \phi_t(a) \le 1, \quad \forall t\in[T],$$

$$\phi_t(a) \ge 0, \quad \forall(t,a)\in[T]\times A^+.$$

To see that $V^{\mathrm{Hyb}}$ gives an upper bound on $V^{\mathrm{ON}}$, we fix a request trajectory $\theta_1,\cdots,\theta_T$. Now, for any non-anticipating strategy $\pi$, we let

$$p_t^\pi(a) = \Pr[a_t^\pi = a \mid \theta_1,\cdots,\theta_t]$$

be the total probability that $a_t^\pi = a$ conditioning on the pre-determined request sequence, with respect to $\gamma_1, \cdots, \gamma_{t-1}$ and the randomness of strategy $\pi$. We show that $\{p_t^\pi\}_{t=1,\cdots,T}$ is a feasible solution to $V^{\text{Hyb}}$ under $\theta_1, \cdots, \theta_T$. Here, a key observation is that for any $t \in [T]$:

$$\mathbb{E}\left[\boldsymbol{c}(\theta_t, a_t^\pi, \gamma_t) \mid \theta_1, \cdots, \theta_t\right] = \mathbb{E}_{\gamma_t}\left[\sum_{a \in A^+} \boldsymbol{c}(\theta_t, a_t^\pi, \gamma_t) \cdot \Pr[a_t^\pi = a \mid \theta_1, \cdots, \theta_t]\right]$$

$$= \sum_{a \in A^+} \boldsymbol{C}(\theta_t, a)p_t^\pi(a).$$

In the above, the first expectation is taken on $\gamma_1, \cdots, \gamma_t$ and the random choice of strategy $\pi$. Since $\sum_{t=1}^T \boldsymbol{c}(\theta_t, a_t^\pi, \gamma_t) \leq \boldsymbol{\rho}T$ always holds, we derive that

$$\sum_{t=1}^T \sum_{a \in A^+} \boldsymbol{C}(\theta_t, a)p_t^\pi(a) = \mathbb{E}\left[\sum_{t=1}^T \boldsymbol{c}(\theta_t, a_t^\pi, \gamma_t) \mid \theta_1, \cdots, \theta_T\right] \leq \boldsymbol{\rho}T,$$

which indicates that $\{p_t^\pi\}_{t=1,\cdots,T}$ is feasible to $V^{\text{Hyb}}$ under $\theta_1, \cdots, \theta_T$. To the same reason, we also have

$$\sum_{t=1}^T \sum_{a \in A^+} R(\theta_t, a)p_t^\pi(a) = \mathbb{E}\left[\sum_{t=1}^T \boldsymbol{r}(\theta_t, a_t^\pi, \gamma_t) \mid \theta_1, \cdots, \theta_T\right]$$

equals the conditional expected reward of strategy $\pi$. Thus, since $V^{\text{Hyb}}$ is a maximization problem for any request trajectory, we conclude that $V^{\text{Hyb}} \geq V^{\text{ON}}$.

It remains to show that when $V^{\text{FL}}$, or $J(\boldsymbol{\rho})$ has a unique and degenerate solution, $V^{\text{FL}} - V^{\text{Hyb}} = \Omega(\sqrt{T})$. We first present a transformation of $V^{\text{Hyb}}$. We let

$$x(\theta) := \frac{\#[\text{appearance of } \theta]}{T}$$

be the random variable indicating the frequency of $\theta$ when $\theta$ is drawn $T$ times i.i.d. from $\mathcal{U}$. Obviously, the mean of $\boldsymbol{x}$ is $\boldsymbol{u}$. We now demonstrate that

$$V^{\text{Hyb}} = T \cdot \mathbb{E}_{\boldsymbol{x}}\left[\max_{\phi:\Theta \times A^+ \to \mathbb{R}} \sum_{\theta \in \Theta} x(\theta) \sum_{a \in A^+} R(\theta, a)\phi(\theta, a)\right],$$

$$\text{s.t.} \quad \sum_{\theta \in \Theta} x(\theta) \sum_{a \in A^+} \boldsymbol{C}(\theta, a)\phi(\theta, a) \leq \boldsymbol{\rho},$$

$$\sum_{a \in A^+} \phi(\theta, a) \leq 1, \quad \forall \theta \in \Theta, \tag{3}$$

$$\phi(\theta, a) \geq 0, \quad \forall (\theta, a) \in \Theta \times A^+.$$

To see this, in form (2), it is not hard to see that conditioning on $\theta_1, \cdots, \theta_T$, the value of the optimization is only related to the number of times that any $\theta \in \Theta$ appears in the sequence, and irrelevant with their arriving order. Therefore, by taking an average, it is without loss of generality to suppose that $\phi_{t_1}^* = \phi_{t_2}^*$ as long as $\theta_{t_1} = \theta_{t_2}$. Under such an observation, it is natural that (2) is equivalent to (3).

For convenience, we now recall the definition of $V^{\text{FL}}$:

$$V^{\text{FL}} = T \cdot \max_{\phi:\Theta \times A^+ \to \mathbb{R}} \sum_{\theta \in \Theta} u(\theta) \sum_{a \in A^+} R(\theta, a)\phi(\theta, a),$$

$$\text{s.t.} \quad \sum_{\theta \in \Theta} u(\theta) \sum_{a \in A^+} \boldsymbol{C}(\theta, a)\phi(\theta, a) \leq \boldsymbol{\rho},$$

$$\sum_{a \in A^+} \phi(\theta, a) \leq 1, \quad \forall \theta \in \Theta,$$

$$\phi(\theta, a) \geq 0, \quad \forall (\theta, a) \in \Theta \times A^+.$$

By Sierksma [33], we know that when $J(\boldsymbol{\rho})$ has a unique and degenerate solution, then its dual form has multiple solutions. We then adopt the framework of Vera and Banerjee [38]. In particular, we let $\boldsymbol{\lambda} \geq \mathbf{0}$ be the dual variable vector for the resource constraints, and $\boldsymbol{\mu} \geq \mathbf{0}$ be the dual variable vector for the probability feasibility constraints. If we take $\omega(\theta) = \mu(\theta)/u(\theta)$, then the dual programming of $V^{\mathrm{FL}}/T$ is the following as a function of $\boldsymbol{u}$:

$$\mathcal{D}[Z(\boldsymbol{u})] = \min_{\boldsymbol{\lambda},\boldsymbol{\omega}} \boldsymbol{\rho}^\top \boldsymbol{\lambda} + \boldsymbol{u}^\top \boldsymbol{\omega},$$

$$\text{s.t.} \quad \boldsymbol{\lambda}^\top \boldsymbol{C}(\theta, a) + \omega(\theta) \geq R(\theta, a), \quad \forall (\theta, a) \in \Theta \times A^+,$$
$$\boldsymbol{\lambda} \geq \mathbf{0}, \quad \boldsymbol{\omega} \geq \mathbf{0}.$$

Now, suppose $(\boldsymbol{\lambda}^1, \boldsymbol{\omega}^1)$ and $(\boldsymbol{\lambda}^2, \boldsymbol{\omega}^2)$ are two different optimal solutions to $\mathcal{D}[Z(\boldsymbol{u})]$, which directly leads to $\boldsymbol{\lambda}^1 \neq \boldsymbol{\lambda}^2$ by the programming formation. We let $\boldsymbol{\lambda}' = \boldsymbol{\lambda}^1 - \boldsymbol{\lambda}^2$ and $\boldsymbol{\omega}' = \boldsymbol{\omega}^1 - \boldsymbol{\omega}^2$. Then,

$$\boldsymbol{\rho}^\top \boldsymbol{\lambda}^1 + \boldsymbol{u}^\top \boldsymbol{\omega}^1 = \boldsymbol{\rho}^\top \boldsymbol{\lambda}^2 + \boldsymbol{u}^\top \boldsymbol{\omega}^2 \implies \boldsymbol{\rho}^\top \boldsymbol{\lambda}' + \boldsymbol{u}^\top \boldsymbol{\omega}' = 0. \tag{4}$$

Further, notice that $(\boldsymbol{\lambda}^1, \boldsymbol{\omega}^1)$ and $(\boldsymbol{\lambda}^2, \boldsymbol{\omega}^2)$ are both feasible for $\mathcal{D}[Z(\boldsymbol{x})]$ for any $\boldsymbol{x}$. Since $\mathcal{D}[Z(\boldsymbol{x})]$ is a minimization problem, by a convex combination, we have

$$\mathcal{D}[Z(\boldsymbol{x})] \leq (\boldsymbol{\rho}^\top \boldsymbol{\lambda}^1 + \boldsymbol{x}^\top \boldsymbol{\omega}^1) \mathbf{1}[\boldsymbol{\rho}^\top \boldsymbol{\lambda}' + \boldsymbol{x}^\top \boldsymbol{\omega}' \leq 0] + (\boldsymbol{\rho}^\top \boldsymbol{\lambda}^2 + \boldsymbol{x}^\top \boldsymbol{\omega}^2) \mathbf{1}[\boldsymbol{\rho}^\top \boldsymbol{\lambda}' + \boldsymbol{x}^\top \boldsymbol{\omega}' > 0].$$

Further, by optimality, we know that for any $\boldsymbol{x}$,

$$\mathcal{D}[Z(\boldsymbol{u})] = (\boldsymbol{\rho}^\top \boldsymbol{\lambda}^1 + \boldsymbol{u}^\top \boldsymbol{\omega}^1) \mathbf{1}[\boldsymbol{\rho}^\top \boldsymbol{\lambda}' + \boldsymbol{x}^\top \boldsymbol{\omega}' \leq 0] + (\boldsymbol{\rho}^\top \boldsymbol{\lambda}^2 + \boldsymbol{u}^\top \boldsymbol{\omega}^2) \mathbf{1}[\boldsymbol{\rho}^\top \boldsymbol{\lambda}' + \boldsymbol{x}^\top \boldsymbol{\omega}' > 0].$$

Now, by weak duality, since $V^{\mathrm{Hyb}}/T$ for any given $\boldsymbol{x}$ is a maximization problem, we know from the above two equations that

$$(V^{\mathrm{FL}} - V^{\mathrm{Hyb}})/T$$
$$\geq \mathcal{D}[Z(\boldsymbol{u})] - \mathbb{E}_{\boldsymbol{x}}\left[\mathcal{D}[Z(\boldsymbol{x})]\right]$$
$$\geq \mathbb{E}_{\boldsymbol{x}}\left[((\boldsymbol{u} - \boldsymbol{x})^\top \boldsymbol{\omega}^1)\mathbf{1}[\boldsymbol{\rho}^\top \boldsymbol{\lambda}' + \boldsymbol{x}^\top \boldsymbol{\omega}' \leq 0] + ((\boldsymbol{u} - \boldsymbol{x})^\top \boldsymbol{\omega}^2)\mathbf{1}[\boldsymbol{\rho}^\top \boldsymbol{\lambda}' + \boldsymbol{x}^\top \boldsymbol{\omega}' > 0]\right]$$
$$\overset{(a)}{=} \mathbb{E}_{\boldsymbol{x}}\left[((\boldsymbol{u} - \boldsymbol{x})^\top \boldsymbol{\omega}^1)\mathbf{1}[(\boldsymbol{u} - \boldsymbol{x})^\top \boldsymbol{\omega}' \geq 0] + ((\boldsymbol{u} - \boldsymbol{x})^\top \boldsymbol{\omega}^2)(1 - \mathbf{1}[(\boldsymbol{u} - \boldsymbol{x})^\top \boldsymbol{\omega}' \geq 0])\right]$$
$$\overset{(b)}{=} \mathbb{E}_{\boldsymbol{x}}\left[((\boldsymbol{u} - \boldsymbol{x})^\top \boldsymbol{\omega}')\mathbf{1}[(\boldsymbol{u} - \boldsymbol{x})^\top \boldsymbol{\omega}' \geq 0]\right].$$

Here, (a) is due to (4), and (b) is since the mean of $\boldsymbol{x}$ is $\boldsymbol{u}$. Now, we let $\xi = \sqrt{T}(\boldsymbol{u} - \boldsymbol{x})^\top \boldsymbol{\omega}'$ be the normalized scaled variable. By the Central Limit Theorem, $\xi \mathbf{1}[\xi \geq 0]$ converges to a half-normal distribution, which has a constant expectation. Thus, we arrive at $V^{\mathrm{FL}} - V^{\mathrm{Hyb}} = \Omega(\sqrt{T})$, which finishes the proof.

## C   Missing Proofs in Section 3

### C.1   Proof of Theorem 3.1

We now give a proof of Theorem 3.1. The proof draws inspiration from that of Chen et al. [16], but significantly diverges in terms of the problem setting.

#### C.1.1   Regret Decomposition

We start by presenting a regret decomposition approach, which stands on the dual viewpoint. We first recall the optimization problem $V^{\mathrm{FL}} = T \cdot J(\boldsymbol{\rho}_1)$:

$$J(\boldsymbol{\rho}_1) := \max_{\phi:\Theta \times A^+ \to \mathbb{R}} \mathbb{E}_{\theta \sim \mathcal{U}}\left[\sum_{a \in A^+} R(\theta, a)\phi(\theta, a)\right],$$

$$\text{s.t.} \quad \mathbb{E}_{\theta \sim \mathcal{U}}\left[\sum_{a \in A^+} \boldsymbol{C}(\theta, a)\phi(\theta, a)\right] \leq \boldsymbol{\rho}_1,$$

$$\sum_{a \in A^+} \phi(\theta, a) \le 1, \quad \forall \theta \in \Theta,$$

$$\phi(\theta, a) \ge 0, \quad \forall (\theta, a) \in \Theta \times A^+.$$

Recall that $u(\theta)$ denotes the mass function of $\mathcal{U}$, then the above linear program can be expanded as

$$J(\boldsymbol{\rho}_1) := \max_{\phi: \Theta \times A^+ \to \mathbb{R}} \sum_{\theta \in \Theta, a \in A^+} u(\theta) R(\theta, a) \phi(\theta, a),$$

$$\text{s.t.} \quad \sum_{\theta \in \Theta, a \in A^+} u(\theta) \boldsymbol{C}(\theta, a) \phi(\theta, a) \le \boldsymbol{\rho}_1,$$

$$\sum_{a \in A^+} \phi(\theta, a) \le 1, \quad \forall \theta \in \Theta,$$

$$\phi(\theta, a) \ge 0, \quad \forall (\theta, a) \in \Theta \times A^+.$$

Now let $\boldsymbol{\lambda} \ge \mathbf{0}$ be the dual vector for the consumption constraint and $\{\mu^*(\theta)\}_{\theta \in \Theta} \ge \mathbf{0}$ be the dual variables for the action distribution constraint. By the strong duality of linear program, there is an optimal dual variable tuple $(\boldsymbol{\lambda}^*, \{\mu^*(\theta)\}_{\theta \in \Theta}) \ge \mathbf{0}$ such that:

$$
\begin{aligned}
J(\boldsymbol{\rho}_1) &= \sum_{\theta \in \Theta, a \in A^+} \left( u(\theta) \left( R(\theta, a) - (\boldsymbol{\lambda}^*)^\top \boldsymbol{C}(\theta, a) \right) - \mu^*(\theta) \right) \phi_1^*(\theta, a) + (\boldsymbol{\lambda}^*)^\top \boldsymbol{\rho}_1 + \sum_{\theta \in \Theta} \mu^*(\theta) \\
&= \sum_{\theta \in \Theta, a \in A^+} u(\theta) \left( R(\theta, a) - (\boldsymbol{\lambda}^*)^\top \boldsymbol{C}(\theta, a) \right) \phi_1^*(\theta, a) + (\boldsymbol{\lambda}^*)^\top \boldsymbol{\rho}_1.
\end{aligned}
$$

(5)

Here $\phi_1^*$ is the optimal solution to $J(\boldsymbol{\rho}_1)$. With (5), we have the following lemma for regret decomposition.

**Lemma C.1.** *For any stopping time $T_e \le T_0$ adapted to the process $\{\boldsymbol{B}_t\}$'s, we have*

$$
\begin{aligned}
&V^{\text{FL}} - Rew \\
&\le \mathbb{E} \left[ \sum_{t=1}^{T_e} \sum_{\theta \in \Theta, a \in A^+} \left( u(\theta) \left( R(\theta, a) - (\boldsymbol{\lambda}^*)^\top \boldsymbol{C}(\theta, a) \right) - \mu^*(\theta) \right) \left( \phi_1^*(\theta, a) - \widehat{\phi}_t^*(\theta, a) \right) \right] \\
&\quad + \mathbb{E} \left[ \sum_{t=1}^{T_e} \sum_{\theta \in \Theta} \mu^*(\theta) \left( 1 - \sum_{a \in A^+} \widehat{\phi}_t^*(\theta, a) \right) \right] \\
&\quad + (\boldsymbol{\lambda}^*)^\top \mathbb{E} [\boldsymbol{B}_{T_e + 1}] + \max_{\theta \in \Theta, a \in A^+} \left( R(\theta, a) - (\boldsymbol{\lambda}^*)^\top \boldsymbol{C}(\theta, a) \right) \cdot \mathbb{E} [T - T_e].
\end{aligned}
$$

(6)

The proof of Lemma C.1 is deferred to Appendix C.2. We now give a brief explanation on this result. The first two terms in (6) depicts the gap between the choice of Algorithm 1 and the optimal decision. This is apparent for the first term. For the second term, we should notice that by complementary slackness, for each $\theta \in \Theta$,

$$\mu^*(\theta) \cdot \left( 1 - \sum_{a \in A^+} \phi_1^*(\theta, a) \right) = 0.$$

Therefore, the second term in (6) is bounded if $\widehat{\phi}_t^*$ is close to $\phi_1^*$.

On the other hand, the last two terms are closely related to the choice of stopping time $T_e$ and the consumption behavior of Algorithm 1. Intuitively, if $T_e$ is sufficiently close to $T$, then $\mathbb{E}[T - T_e]$ should be appropriately bounded. Nevertheless, if the algorithm spends the resources too fast, then such a sufficiently large $T_e$ would be impossible. Conversely, if the resources are consumed substantially slower than the optimal, then the term $\mathbb{E}[\boldsymbol{B}_{T_e + 1}]$, the remaining resources at the stopping time, would be unbounded.

In the following, we will deal with these two parts correspondingly. A crux to the analysis is to pick a satisfying stopping time $T_e$, which we will first cover.

### C.1.2 The Gap to Optimal Decision

We first give a realization of the stopping time $T_e$. With Proposition 3.1 in hand, we can derive that when condition (1) is met, it holds that

$$\left(u(\theta)\left(R(\theta,a)-(\boldsymbol{\lambda}^*)^\top \boldsymbol{C}(\theta,a)\right)-\mu^*(\theta)\right)\left(\phi_1^*(\theta,a)-\widehat{\phi}_t^*(\theta,a)\right)=0, \tag{7}$$

$$\sum_{\theta\in\Theta}\mu^*(\theta)\left(1-\sum_{a\in A^+}\widehat{\phi}_t^*(\theta,a)\right)=0. \tag{8}$$

To see these, notice that by the dual feasibility of $J(\boldsymbol{\rho}_1)$, we have $u(\theta)\left(R(\theta,a)-(\boldsymbol{\lambda}^*)^\top \boldsymbol{C}(\theta,a)\right)-\mu^*(\theta)\leq 0$. When $u(\theta)\left(R(\theta,a)-(\boldsymbol{\lambda}^*)^\top \boldsymbol{C}(\theta,a)\right)-\mu^*(\theta)<0$, by primal optimality, $\phi_1^*(\theta,a)=0$ and thus $(\theta,a)$ is non-basic for $J(\boldsymbol{\rho}_1)$. By Proposition 3.1, $(\theta,a)$ is also non-basic for $\widehat{J}(\boldsymbol{\rho}_t,\mathcal{H}_t)$ and $\widehat{\phi}_t^*(\theta,a)=0$ holds as well. This finishes the deduction of (7). A similar reasoning on binding constraints would help us achieve (8), which we omit here.

As the above goes, it is then natural for us to define $T_e$ the stopping time in our analysis as follows:

$$T_e := \min\{T_0,\min\{t:\max\{\|\boldsymbol{\rho}_1|_{\mathcal{S}}-\boldsymbol{\rho}_t|_{\mathcal{S}}\|_\infty,\max\{\boldsymbol{\rho}_1|_{\mathcal{T}}-\boldsymbol{\rho}_t|_{\mathcal{T}}\}\}>D\}-1\}, \tag{9}$$

where $T_0$ is the stopping time of Algorithm 1. With the definition, we always have $\max\{\|\boldsymbol{\rho}_1|_{\mathcal{S}}-\boldsymbol{\rho}_t|_{\mathcal{S}}\|_\infty,\max\{\boldsymbol{\rho}_1|_{\mathcal{T}}-\boldsymbol{\rho}_t|_{\mathcal{T}}\}\}\leq D$ when $t\leq T_e$. What we are left is to bound the situation when $\max\{\|(u(\theta)-\widehat{u}_t(\theta))_{\theta\in\Theta}\|_\infty,\|(v(\gamma)-\widehat{v}_t(\gamma))_{\gamma\in\Gamma}\|_1\}>D$ for $1\leq t\leq T_e$. In total, we arrive at the following result for this part, with the proof given in Appendix C.4:

**Lemma C.2.** *Under Assumption 3.1, with full information feedback, we have when $T\to\infty$:*

$$\mathbb{E}\left[\sum_{t=1}^{T_e}\sum_{\theta\in\Theta,a\in A^+}\left(u(\theta)\left(R(\theta,a)-(\boldsymbol{\lambda}^*)^\top \boldsymbol{C}(\theta,a)\right)-\mu^*(\theta)\right)\left(\phi_1^*(\theta,a)-\widehat{\phi}_t^*(\theta,a)\right)\right]$$

$$=O\left(\frac{k}{D^2}\right),$$

$$\mathbb{E}\left[\sum_{t=1}^{T_e}\sum_{\theta\in\Theta}\mu^*(\theta)\left(1-\sum_{a\in A^+}\widehat{\phi}_t^*(\theta,a)\right)\right]=O\left(\frac{k}{D^2}\right).$$

We are now only left to bound the last two terms in (6).

### C.1.3 The Gap to Optimal Consumption

As presented in (6), we now bound the remaining two terms, respectively $\mathbb{E}[\boldsymbol{B}_{T_e+1}]$ and $\mathbb{E}[T-T_e]$ for $T_e$ defined in (9). It turns out that these two terms are closely related. Due to this observation, we would first bound $(\boldsymbol{\lambda}^*)^\top\cdot\mathbb{E}[\boldsymbol{B}_{T_e+1}]$ by $\mathbb{E}[T-T_e]$, and then bound $\mathbb{E}[T-T_e]$.

Now by the strong duality of $J(\boldsymbol{\rho}_1)$, we know that complementary slackness holds, that is $\boldsymbol{\lambda}^*|_{\mathcal{T}}=\mathbf{0}$. We therefore have

$$(\boldsymbol{\lambda}^*)^\top\mathbb{E}[\boldsymbol{B}_{T_e+1}]\leq(\boldsymbol{\lambda}^*)^\top\mathbb{E}[\boldsymbol{B}_{T_e}]=(\boldsymbol{\lambda}^*|_{\mathcal{S}})^\top\mathbb{E}[\boldsymbol{B}_{T_e}|_{\mathcal{S}}]=(\boldsymbol{\lambda}^*|_{\mathcal{S}})^\top\mathbb{E}[(T-T_e+1)\boldsymbol{\rho}_{T_e}|_{\mathcal{S}}]$$

$$\overset{(a)}{\leq}n(\rho^{\max}+D)\|\boldsymbol{\lambda}^*\|_\infty\cdot\mathbb{E}[T-T_e+1]. \tag{10}$$

In the above, recall that $\rho^{\max}$ denotes the maximum coordinate of $\boldsymbol{\rho}_1$, and $D$ is specified in Proposition 3.1. Consequently, (a) is due to the definition of $T_e$ and that $\|\boldsymbol{\rho}_1+D\mathbf{1}\|_\infty\leq\rho^{\max}+D$.

We are left to bound $\mathbb{E}[T-T_e]$. Nevertheless, this part would be rather technical and involved. Therefore we defer the analysis to Appendix C.5, and only give the final bounds.

**Lemma C.3.** *Under Assumption 3.1, with full information feedback, we have when $T\to\infty$:*

$$(\boldsymbol{\lambda}^*)^\top\mathbb{E}[\boldsymbol{B}_{T_e+1}]+\max_{\theta\in\Theta,a\in A^+}\left(R(\theta,a)-(\boldsymbol{\lambda}^*)^\top\cdot \boldsymbol{C}(\theta,a)\right)\mathbb{E}[T-T_e]=O\left(\frac{n^2}{D^2}\right).$$

Combining Lemmas C.1 to C.3, we arrive at Theorem 3.1.

## C.2 Proof of Lemma C.1

The proof is obtained by the following set of (in)equalities.

$$V^{\mathrm{FL}} - Rew$$

$$= T \cdot J(\boldsymbol{\rho}_1) - \mathbb{E}\left[\sum_{t=1}^{T_0} r(\theta_t, a_t, \gamma_t)\right]$$

$$\overset{(a)}{\leq} T \cdot J(\boldsymbol{\rho}_1) - \mathbb{E}\left[\sum_{t=1}^{T_e} r(\theta_t, a_t, \gamma_t)\right]$$

$$\overset{(b)}{=} T \cdot J(\boldsymbol{\rho}_1) - \mathbb{E}\left[\sum_{t=1}^{T_e} \sum_{\theta \in \Theta, a \in A^+} u(\theta) R(\theta, a) \widehat{\phi}_t^*(\theta, a)\right]$$

$$\overset{(c)}{=} T \cdot \left(\sum_{\theta \in \Theta, a \in A^+} \left(u(\theta)(R(\theta, a) - (\boldsymbol{\lambda}^*)^\top \boldsymbol{C}(\theta, a)) - \mu^*(\theta)\right) \phi_1^*(\theta, a) + (\boldsymbol{\lambda}^*)^\top \boldsymbol{\rho}_1 + \sum_{\theta \in \Theta} \mu^*(\theta)\right)$$

$$- \mathbb{E}\left[\sum_{t=1}^{T_e} \sum_{\theta \in \Theta, a \in A^+} u(\theta) R(\theta, a) \widehat{\phi}_t^*(\theta, a)\right]$$

$$\overset{(d)}{=} \mathbb{E}\left[\sum_{t=1}^{T_e} \sum_{\theta \in \Theta, a \in A^+} \left(u(\theta)\left(R(\theta, a) - (\boldsymbol{\lambda}^*)^\top \boldsymbol{C}(\theta, a)\right) - \mu^*(\theta)\right)\left(\phi_1^*(\theta, a) - \widehat{\phi}_t^*(\theta, a)\right)\right]$$

$$+ \mathbb{E}\left[\sum_{t=1}^{T_e} \sum_{\theta \in \Theta} \mu^*(\theta)\left(1 - \sum_{a \in A^+} \widehat{\phi}_t^*(\theta, a)\right)\right] + \left(\sum_{\theta \in \Theta^*} \mu^*(\theta)\left(1 - \sum_{a \in A^+} \phi_1^*(\theta, a)\right)\right) \cdot \mathbb{E}\left[T - T_e\right]$$

$$+ (\boldsymbol{\lambda}^*)^\top \mathbb{E}\left[T \boldsymbol{\rho}_1 - \sum_{t=1}^{T_e} \sum_{\theta \in \Theta, a \in A^+} u(\theta) \boldsymbol{C}(\theta, a) \widehat{\phi}_t^*(\theta, a)\right]$$

$$+ \left(\sum_{\theta \in \Theta, a \in A^+} \left(u(\theta)(R(\theta, a) - (\boldsymbol{\lambda}^*)^\top \boldsymbol{C}(\theta, a))\right) \phi_1^*(\theta, a)\right) \cdot \mathbb{E}\left[T - T_e\right]$$

$$\overset{(e)}{\leq} \mathbb{E}\left[\sum_{t=1}^{T_e} \sum_{\theta \in \Theta, a \in A^+} \left(u(\theta)\left(R(\theta, a) - (\boldsymbol{\lambda}^*)^\top \boldsymbol{C}(\theta, a)\right) - \mu^*(\theta)\right)\left(\phi_1^*(\theta, a) - \widehat{\phi}_t^*(\theta, a)\right)\right]$$

$$+ \mathbb{E}\left[\sum_{t=1}^{T_e} \sum_{\theta \in \Theta} \mu^*(\theta)\left(1 - \sum_{a \in A^+} \widehat{\phi}_t^*(\theta, a)\right)\right]$$

$$+ (\boldsymbol{\lambda}^*)^\top \mathbb{E}\left[\boldsymbol{B}_{T_e+1}\right] + \max_{\theta \in \Theta, a \in A^+} \left(R(\theta, a) - (\boldsymbol{\lambda}^*)^\top \boldsymbol{C}(\theta, a)\right) \cdot \mathbb{E}\left[T - T_e\right].$$

In the above set of derivations, (a) holds since $T_0 \geq T_e$, (b) is due to Optional Stopping Theorem since $T_e$ is a stopping time, (c) is by the strong duality of $J(\boldsymbol{\rho}_1)$ as given by (5), (d) establishes by rearranging terms. At last, for (e), the diminishing term is by strong duality, the transformation from the fourth term in (d) to the third term in (e) is derived by another application of Optional Stopping Theorem on the accumulated consumption vector, and for the last term, the upper bound is achieved since $\sum_{a \in A^+} \phi_1^*(\theta, a) \leq 1$ for any $\theta \in \Theta$ and $\sum_{\theta \in \Theta} u(\theta) = 1$.

## C.3 Proof of Proposition 3.1

We will apply the stability result in Chen et al. [16] as an intermediate to prove our version. As given, we know that $J(\boldsymbol{\rho}_1)$ and $\widehat{J}(\boldsymbol{\rho}_t, \mathcal{H}_t)$ has the same set of basic/non-basic variables and binding/non-

binding constraints as long as the following conditions hold for some constant $D_0 > 0$:

$$\left\| \left( u(\theta) \sum_\gamma v(\gamma) r(\theta, a, \gamma) - \widehat{u}_t(\theta) \sum_\gamma \widehat{v}_t(\gamma) r(\theta, a, \gamma) \right)_{(\theta, a) \in \Theta \times A^+} \right\|_\infty \leq D_0,$$

$$\left\| \left( u(\theta) \sum_\gamma v(\gamma) \boldsymbol{c}^i(\theta, a, \gamma) - \widehat{u}_t(\theta) \sum_\gamma \widehat{v}_t(\gamma) \boldsymbol{c}^i(\theta, a, \gamma) \right)_{(\theta, a) \in \Theta \times A^+} \right\|_\infty \leq D_0, \quad \forall i \in [n], \tag{11}$$

$$\|\boldsymbol{\rho}_1|_{\mathcal{S}} - \boldsymbol{\rho}_t|_{\mathcal{S}}\|_\infty \leq D_0, \quad \max\{\boldsymbol{\rho}_1|_{\mathcal{T}} - \boldsymbol{\rho}_t|_{\mathcal{T}}\} \leq D_0.$$

Now, by a standard insertion technique, we have

$$u(\theta) \sum_\gamma v(\gamma) r(\theta, a, \gamma) - \widehat{u}_t(\theta) \sum_\gamma \widehat{v}_t(\gamma) r(\theta, a, \gamma)$$

$$= (u(\theta) - \widehat{u}_t(\theta)) \sum_\gamma v(\gamma) r(\theta, a, \gamma) + \widehat{u}_t(\theta) \sum_\gamma (v(\gamma) - \widehat{v}_t(\gamma)) r(\theta, a, \gamma)$$

$$\overset{(a)}{\leq} \|(u(\theta) - \widehat{u}_t(\theta))_{\theta \in \Theta}\|_\infty + \|(v(\gamma) - \widehat{v}_t(\gamma))_{\gamma \in \Gamma}\|_1. \tag{12}$$

For (a), the first term is bounded since $r(\theta, a, \gamma) \leq 1$ and $\sum_\gamma v(\gamma) = 1$. The second term is similarly bounded as $\widehat{u}_t(\theta) \leq 1$. Therefore, we let $D = D_0/2$, then when we have

$$\|(u(\theta) - \widehat{u}_t(\theta))_{\theta \in \Theta}\|_\infty \leq D, \quad \|(v(\gamma) - \widehat{v}_t(\gamma))_{\gamma \in \Gamma}\|_1 \leq D,$$

the first condition in (11) is met. An almost identical reasoning also holds for the second condition in (11). Consequently we finish the proof of the lemma.

### C.4 Proof of Lemma C.2

Recall that we are going to prove that

$$\mathbb{E}\left[ \sum_{t=1}^{T_e} \sum_{\theta \in \Theta, a \in A^+} \left( u(\theta) \left( R(\theta, a) - (\boldsymbol{\lambda}^*)^\top \boldsymbol{C}(\theta, a) \right) - \mu^*(\theta) \right) \left( \phi_1^*(\theta, a) - \widehat{\phi}_t^*(\theta, a) \right) \right]$$

$$= O\left( \frac{k}{D^2} \right),$$

$$\mathbb{E}\left[ \sum_{t=1}^{T_e} \sum_{\theta \in \Theta} \mu^*(\theta) \left( 1 - \sum_{a \in A^+} \widehat{\phi}_t^*(\theta, a) \right) \right] = O\left( \frac{k}{D^2} \right),$$

when $T \to \infty$ under Assumption 3.1. For simplicity, we give the following abbreviations:

$$P_t := \sum_{\theta \in \Theta, a \in a^+} \left( u(\theta) \left( R(\theta, a) - (\boldsymbol{\lambda}^*)^\top \boldsymbol{C}(\theta, a) \right) - \mu^*(\theta) \right) \left( \phi_1^*(\theta, a) - \widehat{\phi}_t^*(\theta, a) \right),$$

$$Q_t := \sum_{\theta \in \Theta} \mu^*(\theta) \left( 1 - \sum_{a \in A^+} \widehat{\phi}_t^*(\theta, a) \right),$$

$$\mathcal{E}_{u,t} := [\|(u(\theta) - \widehat{u}_t(\theta))_{\theta \in \Theta}\|_\infty \leq D], \quad \mathcal{E}_{v,t} := [\|(v(\gamma) - \widehat{v}_t(\gamma))_{\gamma \in \Gamma}\|_1 \leq D].$$

On this end, we first utilize Proposition 3.1 to show that when condition (1) holds, we have

$$P_t = Q_t = 0.$$

Specifically, for $P_t$, by the dual feasibility of $J(\boldsymbol{\rho}_1)$, we have $u(\theta) \left( R(\theta, a) - (\boldsymbol{\lambda}^*)^\top \boldsymbol{C}(\theta, a) \right) - \mu^*(\theta) \leq 0$. When $u(\theta) \left( R(\theta, a) - (\boldsymbol{\lambda}^*)^\top \boldsymbol{C}(\theta, a) \right) - \mu^*(\theta) < 0$, by primal optimality, $\phi_1^*(\theta, a) = 0$

and thus $(\theta, a)$ is non-basic for $J(\boldsymbol{\rho}_1)$. By Proposition 3.1, $(\theta, a)$ is also non-basic for $\widehat{J}(\boldsymbol{\rho}_t, \mathcal{H}_t)$ and $\widehat{\phi}_t^*(\theta, a) = 0$ holds as well. In conjunction with the case that $u(\theta)\left(R(\theta, a) - (\boldsymbol{\lambda}^*)^\top \boldsymbol{C}(\theta, a)\right) - \mu^*(\theta) = 0$, we obtain that $P_t = 0$.

For $Q_t$, notice that we have $\mu^*(\theta) \geq 0$ for any $\theta \in \Theta$. The case that $\mu^*(\theta) = 0$, again, does not contribute to the total sum. When $\mu^*(\theta) > 0$, by complementary slackness, $\sum_{a \in A^+} \phi_1^*(\theta, a) = 1$, i.e., $\theta$ is a binding constraint for $J(\boldsymbol{\rho}_1)$. This, by Proposition 3.1, implies that $\theta$ is also binding for $\widehat{J}(\boldsymbol{\rho}_t, \mathcal{H}_t)$, which shows that the second term is also zero.

With the above, it remains to consider the situation that condition (1) does not hold when $t \leq T_e$, or in other words, $\mathcal{E}_{u,t} \wedge \mathcal{E}_{v,t}$ does not hold. Note that $P_t \leq 1$ and $Q_t \leq 1$ always hold. Thus, we only need to bound the probability that $\neg(\mathcal{E}_{u,t} \wedge \mathcal{E}_{v,t})$. By a union bound, we have

$$\Pr[\neg(\mathcal{E}_{u,t} \wedge \mathcal{E}_{v,t})] = \Pr[\neg\mathcal{E}_{u,t} \vee \neg\mathcal{E}_{v,t}] \leq \Pr[\neg\mathcal{E}_{u,t}] + \Pr[\neg\mathcal{E}_{v,t}].$$

For the first term above, we apply the Hoeffding's inequality and a union bound to derive that

$$\Pr[\neg\mathcal{E}_{u,t}] = \Pr[\|(u(\theta) - \widehat{u}_t(\theta))_{\theta \in \Theta}\|_\infty > D] \leq 2k \exp\left(-2D^2(t-1)\right).$$

Whereas for the second term, we use the concentration result in Weissman et al. [41] to derive that

$$\Pr[\neg\mathcal{E}_{v,t}] = \Pr[\|(v(\gamma) - \widehat{v}_t(\gamma))_{\gamma \in \Gamma}\|_1 > D] \leq \left(2^{|\Gamma|} - 2\right) \exp\left(-D^2(t-1)/2\right).$$

Synthesizing the above all, we have

$$\begin{aligned}
\mathbb{E}[P_t] &= \mathbb{E}[P_t \mid \mathcal{E}_{u,t} \wedge \mathcal{E}_{v,t}] \cdot \Pr[\mathcal{E}_{u,t} \wedge \mathcal{E}_{v,t}] + \mathbb{E}[P_t \mid \neg(\mathcal{E}_{u,t} \wedge \mathcal{E}_{v,t})] \cdot \Pr[\neg(\mathcal{E}_{u,t} \wedge \mathcal{E}_{v,t})] \\
&\leq 0 + 1 \cdot \Pr[\neg(\mathcal{E}_{u,t} \wedge \mathcal{E}_{v,t})] \\
&\leq 2k \exp\left(-2D^2(t-1)\right) + \left(2^{|\Gamma|} - 2\right) \exp\left(-D^2(t-1)/2\right), \quad (13) \\
\mathbb{E}[Q_t] &\leq 2k \exp\left(-2D^2(t-1)\right) + \left(2^{|\Gamma|} - 2\right) \exp\left(-D^2(t-1)/2\right). \quad (14)
\end{aligned}$$

Summing (13) and (14) from 1 to $T_e$, we achieve that

$$\begin{aligned}
&\left\{\mathbb{E}\left[\sum_{t=1}^{T_e} P_t\right], \mathbb{E}\left[\sum_{t=1}^{T_e} Q_t\right]\right\} \\
&\leq \sum_{t=1}^{T} \left(2k \exp\left(-2D^2(t-1)\right) + \left(2^{|\Gamma|} - 2\right) \exp\left(-D^2(t-1)/2\right)\right) \\
&\leq \frac{2k}{1 - \exp\left(-2D^2\right)} + \frac{2^{|\Gamma|} - 2}{1 - \exp\left(-D^2/2\right)} = O\left(\frac{k}{D^2}\right),
\end{aligned}$$

which conclude the proof of the lemma.

### C.5 Proof of Lemma C.3

As implied by (10), the proof of this lemma reduces to bound $\mathbb{E}[T - T_e]$, i.e., showing that $T_e$ is sufficiently close to $T$. On this side, we first recall the definition of $T_e$ in (9):

$$T_e := \min\{T_0, \min\{t : \max\{\|\boldsymbol{\rho}_1|_{\mathcal{S}} - \boldsymbol{\rho}_t|_{\mathcal{S}}\|_\infty, \max\{\boldsymbol{\rho}_1|_{\mathcal{T}} - \boldsymbol{\rho}_t|_{\mathcal{T}}\}\} > D\} - 1\},$$

where $T_0$ is the stopping time of Algorithm 1, and $\mathcal{S}$ and $\mathcal{T}$ correspondingly represent the set of binding/non-binding resource constraints in LP $J(\boldsymbol{\rho}_1)$. For simplicity, we define

$$\mathcal{N}(\boldsymbol{\rho}_1, D, \mathcal{S}) := \{\boldsymbol{\kappa} : \max\{\|\boldsymbol{\rho}_1|_{\mathcal{S}} - \boldsymbol{\kappa}|_{\mathcal{S}}\|_\infty, \max\{\boldsymbol{\rho}_1|_{\mathcal{T}} - \boldsymbol{\kappa}|_{\mathcal{T}}\}\} \leq D\}.$$

It is without loss of generality to suppose that $D < \rho^{\min}$. We let

$$T_D := \min\{t : \boldsymbol{\rho}_t \notin \mathcal{N}(\boldsymbol{\rho}_1, D, \mathcal{S})\} - 1, \quad T_- = \lfloor T + 1 - 1/(\rho^{\min} - D)\rfloor.$$

We show that if $t \leq T_-$ and $t \leq T_D$, then $t \leq T_e$. In fact, under the condition, we derive that

$$\boldsymbol{B}_t \geq (T - t + 1)(\boldsymbol{\rho}_1 - D\mathbf{1}) \geq \frac{1}{\rho^{\min} - D}(\boldsymbol{\rho}_1 - D\mathbf{1}) \geq \mathbf{1},$$

which implies that $t \leq T_0$, and therefore $t \leq T_e$. As a result, we have

$$\mathbb{E}[T_e] = \sum_{t=1}^{T} \Pr[T_e \geq t] \geq \sum_{t=1}^{T_-} \Pr[T_e \geq t] \geq \sum_{t=1}^{T_-} \Pr[T_D \geq t] = T_- - \sum_{t=1}^{T_-} \Pr[t > T_D]. \quad (15)$$

Before we continue to bound (15), we first give an observation on the dynamics of $\rho_t$. By the update process of the budget, we have for any $t \geq 1$,

$$\boldsymbol{B}_{t+1} = \boldsymbol{B}_t - \boldsymbol{c}_t \implies \boldsymbol{\rho}_{t+1}(T-t) = \boldsymbol{\rho}_t(T-t+1) - \boldsymbol{c}_t$$

$$\implies \boldsymbol{\rho}_{t+1} = \boldsymbol{\rho}_t + \frac{\boldsymbol{\rho}_t - \boldsymbol{c}_t}{T-t}.$$

Now let

$$\boldsymbol{M}_t^C := \frac{\boldsymbol{\rho}_t - \mathbb{E}_{\theta \sim \mathcal{U}}\left[\sum_{a \in A^+} \widehat{\phi}_t^*(\theta, a) \boldsymbol{C}(\theta, a)\right]}{T-t}, \quad \boldsymbol{N}_t^C := \frac{\mathbb{E}_{\theta \sim \mathcal{U}}\left[\sum_{a \in A^+} \widehat{\phi}_t^*(\theta, a) \boldsymbol{C}(\theta, a)\right] - \boldsymbol{c}_t}{T-t}.$$

We then have

$$\boldsymbol{\rho}_{t+1} - \boldsymbol{\rho}_t = \frac{\boldsymbol{\rho}_t - \boldsymbol{c}_t}{T-t} = \boldsymbol{M}_t^C + \boldsymbol{N}_t^C. \quad (16)$$

We now define an auxiliary process that benefits the analysis. Specifically, for $t \in [T]$, let

$$\tilde{\boldsymbol{\rho}}_t := \begin{cases} \boldsymbol{\rho}_t, & t \leq T_D; \\ \boldsymbol{\rho}_{T_D}, & t > T_D. \end{cases}$$

Therefore,

$$\tilde{\boldsymbol{\rho}}_{t+1} - \tilde{\boldsymbol{\rho}}_t = \begin{cases} \boldsymbol{M}_t^C + \boldsymbol{N}_t^C, & t \leq T_D; \\ 0, & t > T_D. \end{cases}$$

We further define the following two auxiliary variables for $t \in [T]$:

$$\widetilde{\boldsymbol{M}}_t^C := \begin{cases} \boldsymbol{M}_t^C, & t \leq T_D; \\ 0, & t > T_D. \end{cases}, \quad \widetilde{\boldsymbol{N}}_t^C := \begin{cases} \boldsymbol{N}_t^C, & t \leq T_D; \\ 0, & t > T_D. \end{cases}$$

As a result, we have

$$\tilde{\boldsymbol{\rho}}_{t+1} - \tilde{\boldsymbol{\rho}}_t = \widetilde{\boldsymbol{M}}_t^C + \widetilde{\boldsymbol{N}}_t^C.$$

Now we come back to (15). Notice that

$$\Pr[t > T_D] \quad (17)$$
$$= \Pr[\boldsymbol{\rho}_s \notin \mathcal{N}(\boldsymbol{\rho}_1, D, \mathcal{S}) \text{ for some } s \leq t] = \Pr[\tilde{\boldsymbol{\rho}}_t \notin \mathcal{N}(\boldsymbol{\rho}_1, D, \mathcal{S})]$$
$$\leq \Pr\left[\left\|\sum_{\tau=1}^{t-1}\left(\widetilde{\boldsymbol{M}}_\tau^C + \widetilde{\boldsymbol{N}}_\tau^C\right)|_{\mathcal{S}}\right\|_\infty > D \text{ or } \min \sum_{\tau=1}^{t-1}\left(\widetilde{\boldsymbol{M}}_\tau^C + \widetilde{\boldsymbol{N}}_\tau^C\right)|_{\mathcal{T}} < -D\right]$$
$$\leq \Pr\left[\left\|\sum_{\tau=1}^{t-1}\widetilde{\boldsymbol{M}}_\tau^C|_{\mathcal{S}}\right\|_\infty > D/2 \text{ or } \min \sum_{\tau=1}^{t-1}\widetilde{\boldsymbol{M}}_\tau^C|_{\mathcal{T}} < -D/2\right] + \Pr\left[\left\|\sum_{\tau=1}^{t-1}\widetilde{\boldsymbol{N}}_\tau^C\right\|_\infty \geq D/2\right]. \quad (18)$$

For the second term in (18), we observe that each entry of $\{\sum_{\tau < t}\widetilde{\boldsymbol{N}}_\tau^C\}_t$ is a martingale with the absolute value of the $\tau$-th increment bounded by $1/(T-\tau)$. Since

$$\sum_{\tau=1}^{t-1} \frac{1}{(T-\tau)^2} \leq \frac{1}{T-t},$$

by applying the Azuma–Hoeffding inequality and a union bound, we achieve that

$$\Pr\left[\left\|\sum_{\tau=1}^{t-1}\widetilde{\boldsymbol{N}}_\tau^C\right\|_\infty \geq D/2\right] \leq 2n \exp\left(-\frac{(T-t)D^2}{8}\right).$$

We now come back to the first term in (18), for any $\{D_1, \cdots, D_{t-1}\}$ such that $\sum_{\tau=1}^{t-1} D_\tau/(T-\tau) \leq D/2$, we have

$$\left\{ \left\| \sum_{\tau=1}^{t-1} \widetilde{\boldsymbol{M}}_\tau^C|_{\mathcal{S}} \right\|_\infty > D/2 \text{ or } \min \sum_{\tau=1}^{t-1} \widetilde{\boldsymbol{M}}_\tau^C|_{\mathcal{T}} < -D/2 \right\}$$

$$\implies \left\{ \left\| \widetilde{\boldsymbol{M}}_\tau^C|_{\mathcal{S}} \right\|_\infty > \frac{D_\tau}{T-\tau} \text{ or } \min \widetilde{\boldsymbol{M}}_\tau^C|_{\mathcal{T}} < -\frac{D_\tau}{T-\tau} \right\} \text{ for some } \tau \in [T-1].$$

We now define

$$\mathcal{E}_\tau(D_\tau) := \left( \left\| \boldsymbol{M}_\tau^C|_{\mathcal{S}} \right\|_\infty \leq \frac{D_\tau}{T-\tau} \right) \wedge \left( \min \boldsymbol{M}_\tau^C|_{\mathcal{T}} \geq -\frac{D_\tau}{T-\tau} \right) \text{ holds for } \forall \boldsymbol{\rho}_\tau \in \mathcal{N}(\boldsymbol{\rho}_1, D, \mathcal{S}).$$

Since $\widetilde{\boldsymbol{M}}_\tau^C \neq 0$ only when $t \leq T_D$, i.e., $\boldsymbol{\rho}_t \in \mathcal{N}(\boldsymbol{\rho}_1, D, \mathcal{S})$, by the definition of $\mathcal{E}_\tau(D_\tau)$, we have the following claim:

$$\left\{ \left\| \sum_{\tau=1}^{t-1} \widetilde{\boldsymbol{M}}_\tau^C|_{\mathcal{S}} \right\|_\infty > D/2 \text{ or } \min \sum_{\tau=1}^{t-1} \widetilde{\boldsymbol{M}}_\tau^C|_{\mathcal{T}} < -D/2 \right\} \subseteq \bigcup_{\tau=1}^{t-1} \neg\mathcal{E}_\tau(D_\tau), \quad \forall \sum_{\tau=1}^{t-1} \frac{D_\tau}{T-\tau} \leq D/2. \tag{19}$$

Thus, we forward to bound $\Pr[\neg\mathcal{E}_\tau(D_\tau)]$ for a suitable choice of $\{D_\tau\}_{1 \leq \tau \leq T}$. Recall that we have defined events $\mathcal{E}_{u,\tau}$ and $\mathcal{E}_{v,\tau}$ as follows:

$$\mathcal{E}_{u,\tau} := [\|(u(\theta) - \widehat{u}_\tau(\theta))_{\theta \in \Theta}\|_\infty \leq D], \quad \mathcal{E}_{v,\tau} := [\|(v(\gamma) - \widehat{v}_\tau(\gamma))_{\gamma \in \Gamma}\|_1 \leq D].$$

We have the following lemma, which we are going to prove in Appendix C.6:

**Lemma C.4.** *When $\boldsymbol{\rho}_\tau \in \mathcal{N}(\boldsymbol{\rho}_1, D, \mathcal{S})$ and $\mathcal{E}_{u,\tau} \wedge \mathcal{E}_{v,\tau}$ hold,*

$$(T-\tau) \left\| \boldsymbol{M}_\tau^C|_{\mathcal{S}} \right\|_\infty \leq \|(u(\theta) - \widehat{u}_\tau(\theta))_{\theta \in \Theta}\|_1 + \|(v(\gamma) - \widehat{v}_\tau(\gamma))_{\gamma \in \Gamma}\|_1,$$

$$(T-\tau) \min \boldsymbol{M}_\tau^C|_{\mathcal{T}} \geq -\|(u(\theta) - \widehat{u}_\tau(\theta))_{\theta \in \Theta}\|_1 - \|(v(\gamma) - \widehat{v}_\tau(\gamma))_{\gamma \in \Gamma}\|_1.$$

Further, it is clear that $(T-\tau) \left\| \boldsymbol{M}_\tau^C|_{\mathcal{S}} \right\|_\infty \leq 1$ and $(T-\tau)\boldsymbol{M}_\tau^C|_{\mathcal{T}} \geq -1$ holds. Inspired by the above observations, we let the series of $D_1, \cdots, D_{T-1}$ be the following form:

$$D_\tau = \begin{cases} 1, & \tau \leq \eta T; \\ (\tau-1)^{-1/4}, & \tau > \eta T, \end{cases}$$

where $\eta \in (0, 1)$ is a constant to be specified. We need to satisfy the following constraints:

$$\sum_{t=1}^{T-1} \frac{D_t}{T-t} \leq D/2, \quad (\eta T)^{-1/4} < D.$$

Here, the first constraint is instructed by (19), and the second is to guarantee that when $\|(u(\theta) - \widehat{u}_\tau(\theta))_{\theta \in \Theta}\|_1 + \|(v(\gamma) - \widehat{v}_\tau(\gamma))_{\gamma \in \Gamma}\|_1 < (\tau-1)^{-1/4}$ for $\tau > \eta T$, $\mathcal{E}_{u,\tau} \wedge \mathcal{E}_{v,\tau}$ naturally holds, and therefore we can apply Lemma C.4. For the first one, we notice that

$$\sum_{\tau=1}^{T-1} \frac{D_\tau}{T-\tau} = \sum_{\tau=1}^{\eta T} \frac{1}{T-\tau} + \sum_{\tau=\eta T+1}^{T-1} \frac{1}{(T-\tau)(\tau-1)^{1/4}} \leq \log \frac{T-1}{(1-\eta)T-1} + \frac{\log T}{(\eta T)^{1/4}}.$$

Therefore, for some $\eta$ such that $\log(1-\eta) \geq -D/4$, $\sum_{t=1}^{T} D_t/(T-t) \leq D/2$ establishes for sufficiently large $T \gg 1$, and the second constraint is also satisfied.

We are now prepared to bound $\Pr[\neg\mathcal{E}_\tau(D_\tau)]$ for the $\{D_\tau\}$ we just proposed. To start with, when $\tau \leq \eta T$, $\mathcal{E}_\tau(D_\tau)$ always holds, thus $\Pr[\neg\mathcal{E}_\tau(D_\tau)] = 0$. When $\tau > \eta T$, since $\tau^{-1/4}/2 < D$, by Hoeffding's inequality and union bound, we have

$$\Pr[\neg\mathcal{E}_\tau(D_\tau)]$$

$$\overset{(a)}{\leq} \Pr\left[\|(u(\theta) - \widehat{u}_\tau(\theta))_{\theta\in\Theta}\|_1 \leq (\tau-1)^{-1/4}/2\right] + \Pr\left[\|(v(\gamma) - \widehat{v}_\tau(\gamma))_{\gamma\in\Gamma}\|_1 \leq (\tau-1)^{-1/4}/2\right]$$

$$\leq 2k\exp\left(-\frac{(\tau-1)^{1/2}}{8k^2}\right) + 2|\Gamma|\exp\left(-\frac{(\tau-1)^{1/2}}{8|\Gamma|^2}\right).$$

Here, (a) is by Lemma C.4 and a union bound. Therefore, according to (19), we have

$$\Pr\left[\left\|\sum_{\tau=1}^{t-1}\widetilde{\boldsymbol{M}}_\tau^C|_{\mathcal{S}}\right\|_\infty > D/2 \text{ or } \min\sum_{\tau=1}^{t-1}\widetilde{\boldsymbol{M}}_\tau^C|_{\mathcal{T}} < -D/2\right] \leq \sum_{\tau=1}^{t-1}\Pr[\neg\mathcal{E}_\tau(D_\tau)],$$

and therefore,

$$\Pr\left[\left\|\sum_{\tau=1}^{t-1}\widetilde{\boldsymbol{M}}_\tau^C|_{\mathcal{S}}\right\|_\infty > D/2 \text{ or } \min\sum_{\tau=1}^{t-1}\widetilde{\boldsymbol{M}}_\tau^C|_{\mathcal{T}} < -D/2\right] \leq \begin{cases} 0, & t \leq \eta T + 1; \\ \sum_{\tau=\eta T+1}^{t-1}\exp\left\{-\tau^{1/2}\right\}, & t > \eta T + 1. \end{cases}$$

Plugging the into (18) and (15), we obtain that when $T \to \infty$,

$$\mathbb{E}\left[T - T_e\right]$$
$$\leq T - T_-$$
$$+ \sum_{t=1}^{T_-}\left(\Pr\left[\left\|\sum_{\tau=1}^{t-1}\widetilde{\boldsymbol{M}}_\tau^C|_{\mathcal{S}}\right\|_\infty > D/2 \text{ or } \min\sum_{\tau=1}^{t-1}\widetilde{\boldsymbol{M}}_\tau^C|_{\mathcal{T}} < -D/2\right] + 2n\exp\left(-\frac{(T-t)D^2}{8}\right)\right)$$
$$\leq \frac{1}{\rho^{\min} - D} + 2n(1 - \exp(-D^2/8))^{-1} + O(T^2)\exp\left(-T^{1/2}\right) = O\left(\frac{n}{D^2}\right).$$

At last, combining with (10), we finally finish the proof of Lemma C.3.

### C.6 Proof of Lemma C.4

To start with, we notice that

$$(T - \tau)\boldsymbol{M}_\tau^C = \boldsymbol{\rho}_\tau - \mathbb{E}_{\theta\sim\mathcal{U}}\left[\sum_{a\in A^+}\widehat{\phi}_\tau^*(\theta, a)\boldsymbol{C}(\theta, a)\right].$$

Now, notice that $\boldsymbol{\rho}_\tau \in \mathcal{N}(\boldsymbol{\rho}_1, D, \mathcal{S})$ and $\mathcal{E}_{u,\tau} \wedge \mathcal{E}_{v,\tau}$ are the condition of Proposition 3.1, therefore, the set of resource binding constraints of $\widehat{J}(\boldsymbol{\rho}_t, \mathcal{H}_t)$ are identical to that of $J(\boldsymbol{\rho}_1)$, i.e., $\mathcal{S}$. Hence, for any $i \in [n]$,

$$\boldsymbol{\rho}_\tau^i|_{\mathcal{S}} - \mathbb{E}_{\theta\sim\mathcal{U}}\left[\sum_{a\in A^+}\widehat{\phi}_\tau^*(\theta, a)\boldsymbol{C}^i(\theta, a)|_{\mathcal{S}}\right]$$
$$= \sum_{\theta\in\Theta, a\in A^+}\widehat{u}_\tau(\theta)\widehat{\phi}_\tau^*(\theta, a)\sum_\gamma \widehat{v}_\tau(\gamma)\boldsymbol{c}^i(\theta, a, \gamma)|_{\mathcal{S}} - \sum_{\theta\in\Theta, a\in A^+}u(\theta)\widehat{\phi}_\tau^*(\theta, a)\sum_\gamma v(\gamma)\boldsymbol{c}^i(\theta, a, \gamma)|_{\mathcal{S}}$$
$$= \sum_{\theta\in\Theta, a\in A^+}(u(\theta) - \widehat{u}_\tau(\theta))\widehat{\phi}_\tau^*(\theta, a)\sum_\gamma v(\gamma)\boldsymbol{c}^i(\theta, a, \gamma)|_{\mathcal{S}}$$
$$+ \sum_{\theta\in\Theta, a\in A^+}\widehat{u}_\tau(\theta)\widehat{\phi}_\tau^*(\theta, a)\sum_\gamma(\widehat{v}_\tau(\gamma) - v(\gamma))\boldsymbol{c}^i(\theta, a, \gamma)|_{\mathcal{S}}$$
$$\overset{(a)}{\leq} \|(u(\theta) - \widehat{u}_\tau(\theta))_{\theta\in\Theta}\|_1 + \|(v(\gamma) - \widehat{v}_\tau(\gamma))_{\gamma\in\Gamma}\|_1.$$

Here, the bound on the first term in (a) establishes because for any $\theta \in \Theta$,

$$\sum_{a\in A^+}\widehat{\phi}_\tau^*(\theta, a)\sum_\gamma v(\gamma)\boldsymbol{c}^i(\theta, a, \gamma)|_{\mathcal{S}} \leq 1$$

since $\sum_{a\in A^+}\widehat{\phi}_\tau^*(\theta, a) \leq 1$. The bound on the second term is similar. Thus, we achieve the result for binding constraints. The proof for non-binding constraints resembles the above by noticing that

$$\boldsymbol{\rho}_\tau|_{\mathcal{T}} \geq \sum_{\theta\in\Theta, a\in A^+}\widehat{u}_\tau(\theta)\widehat{\phi}_\tau^*(\theta, a)\sum_\gamma \widehat{v}_\tau(\gamma)\boldsymbol{c}(\theta, a, \gamma)|_{\mathcal{T}}.$$

# D Missing Proofs in Section 4

## D.1 Proof of Theorem 4.1

With Lemma 4.1 in hand, we now show how to derive Theorem 4.1. Specifically, the regret decomposition technique in Lemma C.1 still works fine. We only need to re-derive corresponding results for Lemmas C.2 and C.3. We have the following results on this side, which are proved respectively in Appendices D.3 and D.4.

**Lemma D.1.** *Under Assumption 3.1, with partial information feedback, we have when $T \to \infty$:*

$$
\mathbb{E}\left[\sum_{t=1}^{T_e} \sum_{\theta \in \Theta, a \in A^+} \left(u(\theta)\left(R(\theta, a) - (\boldsymbol{\lambda}^*)^\top \boldsymbol{C}(\theta, a)\right) - \mu^*(\theta)\right)\left(\phi_1^*(\theta, a) - \widehat{\phi}_t^*(\theta, a)\right)\right]
$$
$$
= O\left(\frac{k}{D^2}\log T\right),
$$
$$
\mathbb{E}\left[\sum_{t=1}^{T_e} \sum_{\theta \in \Theta} \mu^*(\theta)\left(1 - \sum_{a \in A^+} \widehat{\phi}_t^*(\theta, a)\right)\right] = O\left(\frac{k + \log T}{D^2}\right).
$$

**Lemma D.2.** *Under Assumption 3.1, with partial information feedback, we have when $T \to \infty$:*

$$
(\boldsymbol{\lambda}^*)^\top \mathbb{E}\left[\boldsymbol{B}_{T_e+1}\right] + \max_{\theta \in \Theta, a \in A^+}\left(R(\theta, a) - (\boldsymbol{\lambda}^*)^\top \cdot \boldsymbol{C}(\theta, a)\right)\mathbb{E}\left[T - T_e\right] = O\left(\frac{n}{D^2}\right).
$$

Lemmas C.1, D.1 and D.2 in together leads to Theorem 4.1.

## D.2 Proof of Lemma 4.1

Some preparations are required before we come to prove the lemma. To start with, we notice that $Y_\tau = \Pr[a_1 \neq 0] + \cdots + \Pr[a_{t-1} \neq 0]$. By the control rule of Algorithm 1, we have

$$
\Pr[a_\tau \neq 0] = \mathbb{E}_{\theta \sim \mathcal{U}}\left[\sum_{a \in A^+} \widehat{\phi}_\tau^*(\theta, a) \mid \mathcal{H}_\tau\right].
$$

We first give a lower bound on $\mathbb{E}_{\theta \sim \mathcal{U}}[\sum_{a \in A^+} \widehat{\phi}_\tau^*(\theta, a) \mid \mathcal{H}_\tau]$ with $\boldsymbol{\rho}_\tau$, taking $\mathbb{E}_{\theta \sim \widehat{\mathcal{U}}_\tau}[\sum_{a \in A^+} \widehat{\phi}_\tau^*(\theta, a) \mid \mathcal{H}_\tau]$ as an intermediate.

**Lemma D.3.**

$$
\mathbb{E}_{\theta \sim \widehat{\mathcal{U}}_\tau}\left[\sum_{a \in A^+} \widehat{\phi}_\tau^*(\theta, a) \mid \mathcal{H}_\tau\right] \geq \min\{1, \min \boldsymbol{\rho}_\tau\}.
$$

*Proof of Lemma D.3.* To start with, when $\boldsymbol{\rho}_\tau \geq 1$, then clearly, all the resource constraints in $\widehat{J}(\boldsymbol{\rho}_\tau, \mathcal{H}_\tau)$ are satisfied even when $\sum_{a \in A^+} \phi(\theta, a) = 1$ holds for any $\theta \in \Theta$. Therefore, an optimal solution should have this form.

We now consider the case that $\min \boldsymbol{\rho}_\tau < 1$. In this case, if there is a feasible solution that $\sum_{a, \in A^+} \widehat{\phi}_\tau^*(\theta, a) = 1$ holds for any $\theta \in \Theta$, then the proof is also finished. Otherwise, there is at least a binding resource constraint in $\widehat{J}(\boldsymbol{\rho}_\tau, \mathcal{H}_\tau)$, which we denote by $i^*$. Consequently,

$$
\mathbb{E}_{\theta \sim \widehat{\mathcal{U}}_\tau}\left[\sum_{a \in A^+} \widehat{\phi}_\tau^*(\theta, a)\right] \geq \mathbb{E}_{\theta \sim \widehat{\mathcal{U}}_\tau}\left[\sum_{a \in A^+} \widehat{\phi}_\tau^*(\theta, a)\widehat{\boldsymbol{C}}_\tau^{i^*}(\theta, a)\right] = \boldsymbol{\rho}_\tau^{i^*} \geq \min \boldsymbol{\rho}_\tau.
$$

This finishes the proof of the lemma. $\qquad\square$

Thus, we have

$$\Pr[a_\tau \neq 0] = \mathbb{E}_{\theta \sim \mathcal{U}}\left[\sum_{a \in A^+} \widehat{\phi}_\tau^*(\theta, a) \mid \mathcal{H}_\tau\right] \geq \mathbb{E}_{\theta \sim \widehat{\mathcal{U}}_\tau}\left[\sum_{a \in A^+} \widehat{\phi}_\tau^*(\theta, a) \mid \mathcal{H}_\tau\right] - \|u(\theta) - \widehat{u}_\tau(\theta)\|_1$$

$$\geq \min\{1, \min \boldsymbol{\rho}_\tau\} - \|u(\theta) - \widehat{u}_\tau(\theta)\|_1. \qquad (20)$$

Further, we have the following result bounding $\min \boldsymbol{\rho}_\tau$ when $t$ is no larger than a fraction of $T$.

**Lemma D.4.** *When $t \leq (\rho^{\min}/2) \cdot T$, $\min \boldsymbol{\rho}_\tau \geq \rho^{\min}/2$.*

*Proof of Lemma D.4.* In fact, for $t \leq (\rho^{\min}/2) \cdot T$,

$$\boldsymbol{\rho}_\tau = \frac{T \cdot \boldsymbol{\rho}_1 - \sum_{\tau=1}^{t-1} \boldsymbol{c}_\tau}{T - t + 1} \geq \frac{T \cdot \boldsymbol{\rho}_1 - t \cdot \mathbf{1}}{T} \geq \frac{\boldsymbol{\rho}_1}{2}.$$

This concludes the proof. $\qquad \square$

Now, by Weissman et al. [41], with probability $1 - O(1/T)$, we have

$$\|u(\theta) - \widehat{u}_\tau(\theta)\|_1 \leq \frac{\rho^{\min}}{4}, \quad \forall \tau \geq \Theta(\log T).$$

Taking into (20), we derive that

$$\Pr[a_\tau \neq 0] \geq \frac{\rho^{\min}}{4}, \quad \forall \Theta(\log T) \leq t \leq \frac{\rho^{\min}}{2} \cdot T.$$

Consequently, within the period, the probability that there are $\Omega(\log T)$ consecutive rounds in which the agent chooses to quit in all these rounds is $O(1/T)$. This proves the first part. Meanwhile, at time $t = \lceil(\rho^{\min}/2) \cdot T\rceil + 1$, by Azuma–Hoeffding inequality, we derive that with probability $1 - O(1/T)$, $Y_t = \sum_{\tau=1}^{t-1} \Pr[a_\tau \neq 0] \geq \Omega(T)$, which proves the second part.

### D.3 Proof of Lemma D.1

We concentrate on adapting the proof of Lemma C.2 into the partial information feedback setting. To start with, we suppose that the conditions given in Lemma 4.1 hold. In fact, since the failure probability is only $O(1/T)$, and the sum is upper bounded by $O(T)$, therefore the failure case only contributes $O(1)$ to the total expectation.

Now, recall the following definitions:

$$P_t := \sum_{\theta \in \Theta, a \in a^+} \left(u(\theta)\left(R(\theta, a) - (\boldsymbol{\lambda}^*)^\top \boldsymbol{C}(\theta, a)\right) - \mu^*(\theta)\right)\left(\phi_1^*(\theta, a) - \widehat{\phi}_t^*(\theta, a)\right),$$

$$Q_t := \sum_{\theta \in \Theta} \mu^*(\theta)\left(1 - \sum_{a \in A^+} \widehat{\phi}_t^*(\theta, a)\right),$$

$$\mathcal{E}_{u,t} := [\|(u(\theta) - \widehat{u}_t(\theta))_{\theta \in \Theta}\|_\infty \leq D], \quad \mathcal{E}_{v,t} := [\|(v(\gamma) - \widehat{v}_t(\gamma))_{\gamma \in \Gamma}\|_1 \leq D],$$

and by (13) and (14), we have

$$\mathbb{E}[P_t] \leq \Pr[\neg \mathcal{E}_{u,t}] + \Pr[\neg \mathcal{E}_{v,t}], \quad \mathbb{E}[Q_t] \leq \Pr[\neg \mathcal{E}_{u,t}] + \Pr[\neg \mathcal{E}_{v,t}].$$

Now, the bound on $\Pr[\neg \mathcal{E}_{u,t}]$ inherits the analysis in the proof of Lemma C.2, as partial information feedback does not affect the learning of the request distribution. That is,

$$\Pr[\neg \mathcal{E}_{u,t}] = \Pr[\|(u(\theta) - \widehat{u}_t(\theta))_{\theta \in \Theta}\|_\infty > D] \leq 2k \exp\left(-2D^2(t-1)\right).$$

For $\Pr[\neg \mathcal{E}_{v,t}]$, when $t \leq \Theta(\log T)$, it is obviously bounded by 1. By Lemma 4.1, when $\Theta(\log T) \leq t \leq C_b \cdot T$, by Weissman et al. [41], we have

$$\Pr[\neg \mathcal{E}_{v,t}] = \Pr[\|(v(\gamma) - \widehat{v}_t(\gamma))_{\gamma \in \Gamma}\|_1 > D] \leq \left(2^{|\Gamma|} - 2\right) \exp\left(-\frac{D^2 C_f(t-1)}{2 \log T}\right).$$

Further, when $t > C_b \cdot T$, we correspondingly derive

$$\Pr[\neg \mathcal{E}_{v,t}] \leq \left(2^{|\Gamma|} - 2\right) \exp\left(-\frac{D^2 C_r(t-1)}{2}\right).$$

Putting the above together, we achieve that

$$\left\{\mathbb{E}\left[\sum_{t=1}^{T_e} P_t\right], \mathbb{E}\left[\sum_{t=1}^{T_e} Q_t\right]\right\}$$

$$\leq \sum_{t=1}^{T} 2k \exp\left(-2D^2(t-1)\right) + \Theta(\log T)$$

$$+ \left(2^{|\Gamma|} - 2\right) \left(\sum_{t=\Theta(\log T)}^{C_b \cdot T} \exp\left(-\frac{D^2 C_f(t-1)}{2 \log T}\right) + \sum_{t=C_b \cdot T+1}^{T} \exp\left(-\frac{D^2 C_r(t-1)}{2}\right)\right)$$

$$\leq \Theta\left(\frac{k}{D^2}\right) + \Theta(\log T) + \left(2^{|\Gamma|} - 2\right) \left(\frac{\Theta(1)}{1 - \exp\left(-\Theta(D^2/\log T)\right)} + \exp(-\Theta(T))\right)$$

$$\leq O\left(\frac{k + \log T}{D^2}\right).$$

This finishes the proof.

### D.4 Proof of Lemma D.2

As in Appendix D.3 when we prove Lemma C.2, we only consider the case when the conditions in Lemma 4.1 establish, as the contribution of the failure cases on the expectation-sum is $O(1)$. We now bound $\mathbb{E}[T - T_e]$ in the good case when the sample accessing frequency under partial information feedback is guaranteed. Specifically, as predefined in the proof of Lemma C.2, we only need to re-calculate the following, as the other terms remain unchanged with partial information:

$$\sum_{t=\eta T+2}^{T_-} \sum_{\tau=\eta T+1}^{t-1} \Pr\left[\|(v(\gamma) - \widehat{v}_\tau(\gamma))_{\gamma \in \Gamma}\|_1 \leq (\tau-1)^{-1/4}/2\right].$$

Here, $\eta$ is specified in the definition of $D_\tau$. It is hard for us to directly compare $\eta$ and $C_b$ in Lemma 4.1. Nevertheless, in any case, we know that when $T$ is sufficiently large, $Y_\tau/(\tau-1) = \Omega(1/\log T)$ for $\tau \geq \eta T$. Therefore, we have

$$\Pr\left[\|(v(\gamma) - \widehat{v}_\tau(\gamma))_{\gamma \in \Gamma}\|_1 \leq (\tau-1)^{-1/4}/2\right] \leq 2|\Gamma| \exp\left(-\frac{(\tau-1)^{1/2}}{|\Gamma|^2 O(\log T)}\right).$$

Hence,

$$\sum_{t=\eta T+2}^{T_-} \sum_{\tau=\eta T+1}^{t-1} \Pr\left[\|(v(\gamma) - \widehat{v}_\tau(\gamma))_{\gamma \in \Gamma}\|_1 \leq (\tau-1)^{-1/4}/2\right]$$

$$\leq 2|\Gamma| \sum_{t=\eta T+2}^{T_-} \sum_{\tau=\eta T+1}^{t-1} \exp\left(-\frac{(\tau-1)^{1/2}}{|\Gamma|^2 O(\log T)}\right)$$

$$= O(T^2) \exp\left(-\Omega\left(\frac{T^{1/2}}{\log T}\right)\right) = O(1).$$

Combining with the other parts, Lemma D.2 is proved.

## E  Missing Proofs in Section 5

### E.1  Proof of Theorem 5.1

We will prove Theorem 5.1 in the following, and we are inspired by the analysis in Chen et al. [16].

### E.1.1 Another Regret Decomposition

Different from our analysis for the regular cases, in general circumstances, we introduce another regret decomposition method. The reason for involving such an alternative is that without the regularity assumptions, we no longer have any local stability guarantee even when the estimates are close. Therefore, the decision given by Algorithm 1 does not coincides with the optimal decision even when the distribution learning process converges well, and the corresponding analysis in Section 3 does not work out anymore.

We now present a more general regret decomposition as follows:

$$
\begin{aligned}
V^{\mathrm{FL}} - Rew &= T \cdot J(\boldsymbol{\rho}_1) - \mathbb{E}\left[\sum_{t=1}^{T_0} r(\theta_t, a_t, \gamma_t)\right] \\
&\overset{(a)}{=} T \cdot J(\boldsymbol{\rho}_1) - \mathbb{E}\left[\sum_{t=1}^{T_0} \mathbb{E}_\theta\left[\sum_{a \in A^+} \widehat{\phi}_t^*(\theta, a) R(\theta, a)\right]\right] \\
&\overset{(b)}{=} J(\boldsymbol{\rho}_1) \cdot \mathbb{E}\left[T - T_0\right] + \mathbb{E}\left[\sum_{t=1}^{T_0} \left(J(\boldsymbol{\rho}_1) - \mathbb{E}_\theta\left[\sum_{a \in A^+} \widehat{\phi}_t^*(\theta, a) R(\theta, a)\right]\right)\right].
\end{aligned}
\tag{21}
$$

Here, (a) holds due to the Optimal Stopping Theorem, since $T_0$ is a stopping time. Meanwhile, by the decision process, we have for any $\theta_t$:

$$
\mathbb{E}_{a_t, \gamma_t}[r(\theta_t, a_t, \gamma_t) \mid \theta_t] = \sum_{a \in A^+} \widehat{\phi}_t^*(\theta_t, a) R(\theta_t, a).
$$

Further, (b) is by a re-arrangement. To give a bound for (21), we respectively analyze $\mathbb{E}[T - T_0]$ the stopping time, and difference between the optimal accumulated rewards and the real ones.

### E.1.2 Bounding the Stopping Time

To settle the stopping time, we first reduce it to $\max(\boldsymbol{\rho}_1 - \boldsymbol{\rho}_t, 0)$ for $t \leq T_0$, and then deals with these values. We notice that $t \leq T_0$ as long as that $\boldsymbol{B}_t \geq \mathbf{1}$, or $\boldsymbol{\rho}_t \geq 1/(T - t + 1)$. Now, since for any $i \in [n]$,

$$
\boldsymbol{\rho}_t^i = \boldsymbol{\rho}_1^i - (\boldsymbol{\rho}_1 - \boldsymbol{\rho}_t)^i \geq \rho^{\min} - \max(\boldsymbol{\rho}_1 - \boldsymbol{\rho}_t, 0),
$$

we have $\min \boldsymbol{\rho}_t \geq \rho^{\min} - \max(\boldsymbol{\rho}_1 - \boldsymbol{\rho}_t, 0)$. Therefore,

$$
\begin{aligned}
t \leq T_0 &\impliedby \rho^{\min} - \max(\boldsymbol{\rho}_1 - \boldsymbol{\rho}_t, 0) \geq \frac{1}{T - t + 1} \\
&\iff \max(\boldsymbol{\rho}_1 - \boldsymbol{\rho}_t, 0) \leq \rho^{\min} - \frac{1}{T - t + 1}.
\end{aligned}
\tag{22}
$$

Since $\mathbb{E}[T_0] \geq \Pr[T_0 \geq t] \cdot t$ for any $t \in [T]$, we only need to bound the following term for some certain $t$:

$$
\Pr\left[\max(\boldsymbol{\rho}_1 - \boldsymbol{\rho}_t, 0) \leq \rho^{\min} - \frac{1}{T - t + 1}\right].
$$

We will further prove the following lemma in Appendix E.3:

**Lemma E.1.** *It holds for any $t < T$ that*

$$
\Pr\left[\max(\boldsymbol{\rho}_1 - \boldsymbol{\rho}_t, 0) \geq \Theta\left(\frac{1}{T-1} + k\sum_{\tau=2}^{t-1}\sqrt{\frac{\log T}{(T-\tau)^2(\tau-1)}} + \sqrt{\frac{\log T}{T-t}}\right)\right] \leq O\left(\frac{k+n}{T}\right).
$$

With the light of Lemma E.1, it is natural for us to compute

$$
\sum_{\tau=2}^{t-1}\sqrt{\frac{\log T}{(T-\tau)^2(\tau-1)}} \leq
\begin{cases}
\sqrt{\log T} \cdot \dfrac{4\sqrt{t-2}}{T-1}, & 2 \leq t \leq (T+1)/2; \\[2mm]
\sqrt{\log T} \cdot \dfrac{2}{\sqrt{T-t}}, & t > (T+1)/2.
\end{cases}
$$

In fact, to derive the above, we notice that when $2 \leq t \leq (T+1)/2$,

$$\sum_{\tau=1}^{t-1} \frac{1}{(T-\tau)(\tau-1)^{1/2}} \leq \frac{2}{T-1} \sum_{\tau=2}^{t-1} \frac{1}{(\tau-1)^{1/2}} \leq \frac{4\sqrt{t-1}}{T-1}.$$

Meanwhile, when $t > (T+1)/2$, we have $T - t < t - 1$, which leads to

$$\sum_{\tau=2}^{t-1} \frac{1}{(T-\tau)(\tau-1)^{1/2}} \leq \sqrt{\frac{8}{T-1}} + \sum_{\tau=(T+1)/2}^{t-1} \frac{1}{(T-\tau)^{3/2}} \leq \frac{2}{\sqrt{T-t}}.$$

With these calculations, we come back to the bound on $\mathbb{E}[T_0]$, we notice that when $T$ is sufficiently large and $t = T - O(\log T)$, it holds that

$$\Theta\left(\frac{1}{T-1} + k \sum_{\tau=2}^{t-1} \sqrt{\frac{\log T}{(T-\tau)^2(\tau-1)}} + \sqrt{\frac{\log T}{T-t}}\right) + \frac{1}{T-t+1} = O(1) \leq \rho^{\min}.$$

Thus, we have

$$\mathbb{E}\left[T - T_0\right] = T - \mathbb{E}[T_0] \overset{(a)}{\leq} T - \Pr[T_0 \geq T - O(\log T)] \cdot (T - O(\log T))$$

$$\overset{(b)}{\leq} T - \left(1 - O\left(\frac{1}{T}\right)\right) \cdot (T - O(\log T)) = O(\log T). \tag{23}$$

In the above, (a) is because $\mathbb{E}[T_0] \geq \Pr[T_0 \geq t] \cdot t$ for any fixed $t$, and (b) is due to Lemma E.1. Consequently, we finish the analysis of the stopping time in (21).

### E.1.3 The Gap to the Optimal Reward

The rest part of (21) that we are left to consider is the following:

$$J(\boldsymbol{\rho}_1) - \mathbb{E}_\theta\left[\sum_{a \in A^+} \widehat{\phi}_t^*(\theta, a) R(\theta, a)\right]$$

$$= \left(J(\boldsymbol{\rho}_1) - \widehat{J}(\boldsymbol{\rho}_t, \mathcal{H}_t)\right) + \left(\widehat{J}(\boldsymbol{\rho}_t, \mathcal{H}_t) - \mathbb{E}_\theta\left[\sum_{a \in A^+} \widehat{\phi}_t^*(\theta, a) R(\theta, a)\right]\right). \tag{24}$$

Note that the second difference term in (24) reflects the estimation error on distributions of the context and the external factor, which leads to the following result as to be proved in Appendix E.4:

**Lemma E.2.** *We have for $t \geq 2$:*

$$\mathbb{E}\left[\widehat{J}(\boldsymbol{\rho}_t, \mathcal{H}_t) - \mathbb{E}_\theta\left[\sum_{a \in A^+} \widehat{\phi}_t^*(\theta, a) R(\theta, a)\right]\right] \leq O\left(k\sqrt{\frac{\log T}{t-1}} + \frac{k}{T}\right).$$

Lemma E.2 induces an $O(\sqrt{T \log T})$ accumulated regret considering (24) when summing from $t = 2$ to $T_0 \leq T$. While for the first term in (24), our main thread here is to bound $\widehat{J}(\boldsymbol{\rho}_t, \mathcal{H}_t)$ with $J(\boldsymbol{\rho}_t)$. To fix the idea, we compare these two optimization problems:

$$J(\boldsymbol{\rho}_t) := \max_{\phi:\Theta \times A^+ \to \mathbb{R}_+} \sum_{\theta \in \Theta, a \in A^+} u(\theta)\phi(\theta, a) \sum_\gamma r(\theta, a, \gamma) v(\gamma),$$

$$\text{s.t.} \quad \sum_{\theta \in \Theta, a \in A^+} u(\theta)\phi(\theta, a) \sum_\gamma c(\theta, a, \gamma) v(\gamma) \leq \boldsymbol{\rho}_t,$$

$$\sum_{a \in A^+} \phi(\theta, a) \leq 1, \quad \forall \theta \in \Theta,$$

$$\phi(\theta, a) \geq 0, \quad \forall(\theta, a) \in \Theta \times A^+.$$

$$\widehat{J}(\boldsymbol{\rho}_t, \mathcal{H}_t) := \max_{\phi:\Theta \times A^+ \to \mathbb{R}_+} \sum_{\theta \in \Theta, a \in A^+} \widehat{u}_t(\theta)\phi(\theta,a) \sum_\gamma r(\theta,a,\gamma)\widehat{v}_t(\gamma),$$

$$\text{s.t.} \quad \sum_{\theta \in \Theta, a \in A^+} \widehat{u}_t(\theta)\phi(\theta,a) \sum_\gamma \boldsymbol{c}(\theta,a,\gamma)\widehat{v}_t(\gamma) \le \boldsymbol{\rho}_t,$$

$$\sum_{a \in A^+} \phi(\theta,a) \le 1, \quad \forall \theta \in \Theta,$$

$$\phi(\theta,a) \ge 0, \quad \forall(\theta,a) \in \Theta \times A^+.$$

Now, conceptually, if there is a $0 < \eta_t \le 1$ such that for any $(\theta,a) \in \Theta \times A^+$,

$$u(\theta)\sum_\gamma \boldsymbol{c}(\theta,a,\gamma)v(\gamma) \ge \eta_t \widehat{u}_t(\theta) \sum_\gamma \boldsymbol{c}(\theta,a,\gamma)\widehat{v}_t(\gamma),$$

then for an optimal solution $\phi_t^*$ of $J(\boldsymbol{\rho}_t)$, we see that $\eta_t \phi_t^*$ is a feasible solution of the programming $\widehat{J}(\boldsymbol{\rho}_t, \mathcal{H}_t)$. Thus,

$$\widehat{J}(\boldsymbol{\rho}_t, \mathcal{H}_t) \ge \eta_t \sum_{\theta \in \Theta, a \in A^+} \widehat{u}_t(\theta)\phi_t^*(\theta,a) \sum_\gamma r(\theta,a,\gamma)\widehat{v}_t(\gamma)$$

$$\overset{(a)}{\ge} \eta_t \sum_{\theta \in \Theta, a \in A^+} u(\theta)\phi_t^*(\theta,a) \sum_\gamma r(\theta,a,\gamma)v(\gamma)$$

$$- \eta_t(\|(u(\theta) - \widehat{u}_t(\theta))_{\theta \in \Theta}\|_1 + \|(v(\gamma) - \widehat{v}_t(\gamma))_{\gamma \in \Gamma}\|_1)$$

$$= \eta_t J(\boldsymbol{\rho}_t) - (\|(u(\theta) - \widehat{u}_t(\theta))_{\theta \in \Theta}\|_1 + \|(v(\gamma) - \widehat{v}_t(\gamma))_{\gamma \in \Gamma}\|_1).$$

Here, since $r(\theta,a,\gamma) \le 1$ and $\sum_{a \in A^+} \widehat{\phi}_t^*(\theta,a) \le 1$ for any $\theta$, (a) is expanded as

$$\sum_{\theta \in \Theta, a \in A^+} \widehat{u}_t(\theta)\phi_t^*(\theta,a) \sum_\gamma \widehat{v}_t(\gamma)r(\theta,a,\gamma) - \sum_{\theta \in \Theta, a \in A^+} u(\theta)\phi_t^*(\theta,a) \sum_\gamma v(\gamma)r(\theta,a,\gamma)$$

$$= \sum_{\theta \in \Theta, a \in A^+} (\widehat{u}_t(\theta) - u(\theta))\phi_t^*(\theta,a) \sum_\gamma \widehat{v}_t(\gamma)r(\theta,a,\gamma)$$

$$+ \sum_{\theta \in \Theta, a \in A^+} u(\theta)\phi_t^*(\theta,a) \sum_\gamma (\widehat{v}_t(\gamma) - v(\gamma))r(\theta,a,\gamma)$$

$$\le \|(u(\theta) - \widehat{u}_t(\theta))_{\theta \in \Theta}\|_1 + \|(v(\gamma) - \widehat{v}_t(\gamma))_{\gamma \in \Gamma}\|_1.$$

Consequently,

$$J(\boldsymbol{\rho}_1) - \widehat{J}(\boldsymbol{\rho}_t, \mathcal{H}_t)$$
$$\le (1 - \eta_t)J(\boldsymbol{\rho}_1) + \eta_t(J(\boldsymbol{\rho}_1) - J(\boldsymbol{\rho}_t)) + \|(u(\theta) - \widehat{u}_t(\theta))_{\theta \in \Theta}\|_1 + \|(v(\gamma) - \widehat{v}_t(\gamma))_{\gamma \in \Gamma}\|_1. \quad (25)$$

On top of this, a key observation is that

$$J(\boldsymbol{\rho}_1) - J(\boldsymbol{\rho}_t) \le \frac{\max(\boldsymbol{\rho}_1 - \boldsymbol{\rho}_t, 0)}{\rho^{\min}} \cdot J(\boldsymbol{\rho}_1). \quad (26)$$

In fact, when $\boldsymbol{\rho}_1 \le \boldsymbol{\rho}_t$, (26) is natural as $J(\boldsymbol{\rho}_1) \le J(\boldsymbol{\rho}_t)$. Otherwise, let $\phi_1^*$ be the optimal solution to the programming $J(\boldsymbol{\rho}_1)$. Let $i^*$ be the index that minimizes $\boldsymbol{\rho}_t^{i^*}/\boldsymbol{\rho}_1^{i^*}$. We have $\boldsymbol{\rho}_1^{i^*} > \boldsymbol{\rho}_t^{i^*}$. Evidently, we know that $\phi_1^* \cdot \boldsymbol{\rho}_t^{i^*}/\boldsymbol{\rho}_1^{i^*}$ is a feasible solution to the programming of $J(\boldsymbol{\rho}_t)$. By the optimality of $J(\boldsymbol{\rho}_t)$, we have

$$J(\boldsymbol{\rho}_t) \ge \frac{\boldsymbol{\rho}_t^{i^*}}{\boldsymbol{\rho}_1^{i^*}} \cdot J(\boldsymbol{\rho}_1),$$

which leads to

$$J(\boldsymbol{\rho}_1) - J(\boldsymbol{\rho}_t) \le \left(1 - \frac{\boldsymbol{\rho}_t^{i^*}}{\boldsymbol{\rho}_1^{i^*}}\right) \cdot J(\boldsymbol{\rho}_1) = \frac{\boldsymbol{\rho}_1^{i^*} - \boldsymbol{\rho}_t^{i^*}}{\boldsymbol{\rho}_1^{i^*}} \cdot J(\boldsymbol{\rho}_1) \le \frac{\max(\boldsymbol{\rho}_1 - \boldsymbol{\rho}_t)}{\rho^{\min}} \cdot J(\boldsymbol{\rho}_1).$$

Synthesizing the above two parts, (26) is proved.

As for $\mathbb{E}[\max(\boldsymbol{\rho}_1 - \boldsymbol{\rho}_t, 0)]$, we note that for any non-negative random variable $X$ with upper bound $\bar{X}$ and any positive $\xi$, we have

$$\mathbb{E}[X] \leq \xi \Pr[X \leq \xi] + \bar{X}(1 - \Pr[X \leq \xi]) \leq \xi + \bar{X}(1 - \Pr[X \leq \xi]). \qquad (27)$$

Notice that $\max(\boldsymbol{\rho}_1 - \boldsymbol{\rho}_t, 0)$ is certainly upper bounded by 1. Therefore, as a corollary of Lemma E.1, we have

$$\mathbb{E}\left[\max\left(\boldsymbol{\rho}_1 - \boldsymbol{\rho}_t, 0\right)\right] \leq \begin{cases} O\left(k\dfrac{\sqrt{(t-2)\log T}}{T} + \sqrt{\dfrac{\log T}{T-t}} + \dfrac{k+n}{T}\right), & 2 \leq t \leq (T+1)/2; \\[4mm] O\left(k\sqrt{\dfrac{\log T}{T-t}} + \dfrac{k+n}{T}\right), & t > (T+1)/2. \end{cases}$$

We almost finish the bound now except for determining $\eta_t$ in (25), which we hope is as close to 1 as possible. Nevertheless, we leave the technical parts to Appendix E.5 which derives the following lemma on the total bound:

**Lemma E.3.**

$$\mathbb{E}\left[\sum_{t=1}^{T_0}\left(J(\boldsymbol{\rho}_1) - \widehat{J}(\boldsymbol{\rho}_t, \mathcal{H}_t)\right)\right] = O(k\sqrt{T\log T} + n).$$

Now, we sum the result in Lemma E.2 from $t = 2$ to $T_0$, and plus the constant term for $t = 1$ to obtain that

$$\mathbb{E}\left[\sum_{t=1}^{T_0}\left(\widehat{J}(\boldsymbol{\rho}_t, \mathcal{H}_t) - \mathbb{E}_\theta\left[\sum_{a \in A^+}\widehat{\phi}_t^*(\theta, a)R(\theta, a)\right]\right)\right] = O(k\sqrt{T\log T}).$$

Synthesizing Lemma E.3, (24), (23), and (21), we derive Theorem 5.1.

## E.2 Proof of Theorem 5.2

The proof of this theorem follows the line of Theorem 5.1, and the only difference is to adopt Lemma 4.1 when considering the concentration of estimates. On this side, we can disregard the cases when $t \leq \Theta(\log T)$, as the accumulated regret in this phase is bounded by $O(\log T)$. On the other hand, the time range that $t \geq \Theta(T)$ is asymptotically identical to the full information setting since the accessing frequency is a constant. We only need to consider the case that $\Theta(\log T) \leq t \leq \Theta(T)$, when we have

$$\begin{aligned} \Pr\left[\|(u(\theta) - \widehat{u}_t(\theta))_{\theta \in \Theta}\|_1 \leq -\Theta\left(k\sqrt{\dfrac{\log T}{t-1}}\right)\right] &\leq O\left(\dfrac{k}{T^2}\right), \\ \Pr\left[\|(v(\gamma) - \widehat{v}_t(\gamma))_{\gamma \in \Gamma}\|_1 \leq -\Theta\left(|\Gamma|\dfrac{\log T}{\sqrt{t-1}}\right)\right] &\leq O\left(\dfrac{|\Gamma|}{T^2}\right). \end{aligned} \qquad (28)$$

Taking into the proof of Lemma E.1 and then into the main body, we should find a sufficient large $t$ such that

$$\Theta\left(\frac{\log T}{T-1} + k\sum_{\tau=\Theta(\log T)}^{\Theta(T)}\frac{\log T}{\sqrt{(T-\tau)^2(\tau-1)}} + k\sum_{\tau=\Theta(T)}^{t-1}\sqrt{\frac{\log T}{(T-\tau)^2(\tau-1)}} + \sqrt{\frac{\log T}{T-t}}\right)$$

$$\leq \rho^{\min} - \frac{1}{T-t+1},$$

and $t = T - O(\log T)$ still suffices. Therefore, $\mathbb{E}[T - T_0] = O(\log T)$ also holds under partial information feedback.

Nevertheless, for the counterpart of Lemma E.2, by (28), when we sum from $t = 1$ to $T_0 \leq T$, we derive that

$$\mathbb{E}\left[\sum_{t=1}^{T_0}\left(\widehat{J}(\boldsymbol{\rho}_t, \mathcal{H}_t) - \mathbb{E}_{\theta \sim \mathcal{U}}\left[\sum_{a \in A^+}\widehat{\phi}_t^*(\theta, a)R(\theta, a)\right]\right)\right] \leq O(k\sqrt{T}\log T).$$

At last, for $J(\boldsymbol{\rho}_1) - \widehat{J}(\boldsymbol{\rho}_t, \mathcal{H}_t)$, we face the same degradation on the estimation accuracy, which leads to

$$\mathbb{E}\left[\sum_{t=1}^{T_0}\left(J(\boldsymbol{\rho}_1) - \widehat{J}(\boldsymbol{\rho}_t, \mathcal{H}_t)\right)\right] = O(k\sqrt{T}\log T + n).$$

Therefore, Theorem 5.2 is achieved.

### E.3 Proof of Lemma E.1

Now that we are going to bound $\max(\boldsymbol{\rho}_1 - \boldsymbol{\rho}_t, 0)$. Recall the definitions below which we give in Appendix C.5 when we prove Lemma C.3:

$$\boldsymbol{M}_t^C := \frac{\boldsymbol{\rho}_t - \mathbb{E}_{\theta\sim\mathcal{U}}\left[\sum_{a\in A^+}\widehat{\phi}_t^*(\theta, a)\boldsymbol{C}(\theta, a)\right]}{T - t}, \quad \boldsymbol{N}_t^C := \frac{\mathbb{E}_{\theta\sim\mathcal{U}}\left[\sum_{a\in A^+}\widehat{\phi}_t^*(\theta, a)\boldsymbol{C}(\theta, a)\right] - \boldsymbol{c}_t}{T - t}.$$

By (16), we have

$$\boldsymbol{\rho}_{t+1} - \boldsymbol{\rho}_t = \frac{\boldsymbol{\rho}_t - \boldsymbol{c}_t}{T - t} = \boldsymbol{M}_t^C + \boldsymbol{N}_t^C.$$

Consequently,

$$\max(\boldsymbol{\rho}_1 - \boldsymbol{\rho}_t) = \max\left(-\left(\sum_{\tau=1}^{t-1}\boldsymbol{M}_\tau^C + \sum_{\tau=1}^{t-1}\boldsymbol{N}_\tau^C\right)\right) \leq -\min\sum_{\tau=1}^{t-1}\boldsymbol{M}_\tau^C - \min\sum_{\tau=1}^{t-1}\boldsymbol{N}_\tau^C.$$

For the second term, we notice that each entry of $\{\sum_{\tau<t}\boldsymbol{N}_\tau^C\}_t$ is a martingale with the absolute value of the $\tau$-th increment bounded by $1/(T-\tau)$. Since

$$\sum_{\tau=1}^{t-1}\frac{1}{(T-\tau)^2} \leq \frac{1}{T-t},$$

by applying the Azuma–Hoeffding inequality and a union bound, we achieve that

$$\Pr\left[-\min\sum_{\tau=1}^{t-1}\boldsymbol{N}_\tau^C \geq \sqrt{\frac{2\log T}{T-t}}\right] \leq \frac{n}{T}. \tag{29}$$

On the other hand, for the first term, when $\tau = 1$, it is apparent that $-\min\boldsymbol{M}_1^C \leq 1/(T-1)$. When $\tau \geq 2$, we have for any $i \in [n]$,

$$(T-\tau)\left(\boldsymbol{M}_\tau^C\right)^i$$

$$= \boldsymbol{\rho}_\tau^i - \mathbb{E}_{\theta\sim\mathcal{U}}\left[\sum_{a\in A^+}\widehat{\phi}_\tau^*(\theta, a)\boldsymbol{C}^i(\theta, a)\right]$$

$$\overset{(a)}{\geq} \sum_{\theta\in\Theta,a\in A^+}\widehat{u}_\tau(\theta)\widehat{\phi}_\tau^*(\theta, a)\sum_\gamma \widehat{v}_\tau(\gamma)\boldsymbol{c}^i(\theta, a, \gamma) - \sum_{\theta\in\Theta,a\in A^+}u(\theta)\widehat{\phi}_\tau^*(\theta, a)\sum_\gamma v(\gamma)\boldsymbol{c}^i(\theta, a, \gamma)$$

$$= \sum_{\theta\in\Theta,a\in A^+}(\widehat{u}_\tau(\theta) - u(\theta))\widehat{\phi}_\tau^*(\theta, a)\sum_\gamma \widehat{v}_\tau(\gamma)\boldsymbol{c}^i(\theta, a, \gamma)$$

$$+ \sum_{\theta\in\Theta,a\in A^+}u(\theta)\widehat{\phi}_\tau^*(\theta, a)\sum_\gamma(\widehat{v}_\tau(\gamma) - v(\gamma))\boldsymbol{c}^i(\theta, a, \gamma)$$

$$\geq -\|(u(\theta) - \widehat{u}_\tau(\theta))_{\theta\in\Theta}\|_1 - \|(v(\gamma) - \widehat{v}_\tau(\gamma))_{\gamma\in\Gamma}\|_1.$$

In the above, (a) is because $\widehat{\phi}_\tau^*$ is feasible for $\widehat{J}(\boldsymbol{\rho}_\tau, \mathcal{H}_\tau)$. By Hoeffding's inequality and a union bound, we have

$$\Pr\left[\|(u(\theta) - \widehat{u}_\tau(\theta))_{\theta\in\Theta}\|_1 \leq -k\sqrt{\frac{\log T}{\tau-1}}\right] \leq \frac{k}{T^2},$$

$$\Pr\left[\|(v(\gamma) - \widehat{v}_\tau(\gamma))_{\gamma \in \Gamma}\|_1 \le -|\Gamma|\sqrt{\frac{\log T}{\tau - 1}}\right] \le \frac{|\Gamma|}{T^2}.$$

Thus, suppose the above events hold for all $\tau \le T$ with failure probability only $O(1/T)$,

$$\Pr\left[-\min\sum_{\tau=1}^{t-1} \boldsymbol{M}_\tau^C \ge \Theta\left(\frac{1}{T-1} + k\sum_{\tau=2}^{t-1}\sqrt{\frac{\log T}{(T-\tau)^2(\tau-1)}}\right)\right] \le O\left(\frac{k}{T}\right). \qquad (30)$$

Combining (29) and (30), we derive the lemma.

### E.4 Proof of Lemma E.2

We notice that

$$\widehat{J}(\boldsymbol{\rho}_t, \mathcal{H}_t) = \sum_{\theta \in \Theta, a \in A^+} \widehat{u}_t(\theta)\widehat{\phi}_t^*(\theta, a)\sum_\gamma r(\theta, a, \gamma)\widehat{v}_t(\gamma),$$

and

$$\sum_{\theta \in \Theta, a \in A^+} \widehat{u}_t(\theta)\widehat{\phi}_t^*(\theta, a)\sum_\gamma \widehat{v}_t(\gamma)r(\theta, a, \gamma) - \sum_{\theta \in \Theta, a \in A^+} u(\theta)\widehat{\phi}_t^*(\theta, a)\sum_\gamma v(\gamma)r(\theta, a, \gamma)$$

$$= \sum_{\theta \in \Theta, a \in A^+} (\widehat{u}_t(\theta) - u(\theta))\widehat{\phi}_t^*(\theta, a)\sum_\gamma \widehat{v}_t(\gamma)r(\theta, a, \gamma)$$

$$+ \sum_{\theta \in \Theta, a \in A^+} u(\theta)\widehat{\phi}_t^*(\theta, a)\sum_\gamma (\widehat{v}_t(\gamma) - v(\gamma))r(\theta, a, \gamma)$$

$$\le \|(u(\theta) - \widehat{u}_t(\theta))_{\theta \in \Theta}\|_1 + \|(v(\gamma) - \widehat{v}_t(\gamma))_{\gamma \in \Gamma}\|_1.$$

Thus,

$$\widehat{J}(\boldsymbol{\rho}_t, \mathcal{H}_t) - \mathbb{E}_{\theta \sim \mathcal{U}}\left[\sum_{a \in A^+} \widehat{\phi}_t^*(\theta, a)R(\theta, a)\right] \le \|(u(\theta) - \widehat{u}_t(\theta))_{\theta \in \Theta}\|_1 + \|(v(\gamma) - \widehat{v}_t(\gamma))_{\gamma \in \Gamma}\|_1.$$

By Hoeffding's inequality and a union bound, we have

$$\Pr\left[\|(u(\theta) - \widehat{u}_t(\theta))_{\theta \in \Theta}\|_1 \ge k\sqrt{\frac{\log T}{2(t-1)}}\right] \le \frac{k}{T},$$

$$\Pr\left[\|(v(\gamma) - \widehat{v}_t(\gamma))_{\gamma \in \Gamma}\|_1 \ge |\Gamma|\sqrt{\frac{\log T}{2(t-1)}}\right] \le \frac{|\Gamma|}{T}.$$

Further, the difference we hope to analyze is certainly upper bounded by 1. As a result, with (27), we finish the proof.

### E.5 Proof of Lemma E.3

We come to consider $J(\boldsymbol{\rho}_1) - \widehat{J}(\boldsymbol{\rho}_t, \mathcal{H}_t)$. As per the thread in the main body, we let

$$\delta_t := \frac{\|(u(\theta) - \widehat{u}_t(\theta))_{\theta \in \Theta}\|_\infty + \|(v(\gamma) - \widehat{v}_t(\gamma))_{\gamma \in \Gamma}\|_1}{\min_{\theta \in \Theta, a \in A^+}\{\min\{u(\theta)\boldsymbol{C}(\theta, a) > 0\}\}}.$$

We now claim that for any $(\theta, a, i) \in \Theta \times A^+ \times [n]$,

$$\widehat{u}_t(\theta)\sum_\gamma \boldsymbol{c}^i(\theta, a, \gamma)\widehat{v}_t(\gamma) \le (1 + \delta_t)u(\theta)\sum_\gamma \boldsymbol{c}^i(\theta, a, \gamma)v(\gamma).$$

The above is obvious if $\boldsymbol{C}^i(\theta, a) = \boldsymbol{0}$, or $\boldsymbol{c}^i(\theta, a, \gamma) = 0$ holds for any $\gamma$. When $\boldsymbol{C}(\theta, a) \ne \boldsymbol{0}$, then for any $i \in [n]$,

$$\widehat{u}_t(\theta)\sum_\gamma \boldsymbol{c}^i(\theta, a, \gamma)\widehat{v}_t(\gamma) - u(\theta)\sum_\gamma \boldsymbol{c}^i(\theta, a, \gamma)v(\gamma)$$

$$= (\widehat{u}_t(\theta) - u(\theta)) \sum_\gamma \boldsymbol{c}^i(\theta, a, \gamma)\widehat{v}_t(\gamma) + u(\theta) \sum_\gamma \boldsymbol{c}^i(\theta, a, \gamma)(\widehat{v}_t(\gamma) - v(\gamma))$$

$$\leq \|(u(\theta) - \widehat{u}_t(\theta))_{\theta \in \Theta}\|_\infty + \|(v(\gamma) - \widehat{v}_t(\gamma))_{\gamma \in \Gamma}\|_1$$

$$\leq \delta_t u(\theta) \sum_\gamma \boldsymbol{c}^i(\theta, a, \gamma)v(\gamma).$$

This finish the explanation of the claim. Upon that, if we let $\eta_t := 1 - \delta_t \leq 1/(1 + \delta_t)$, we derive that

$$u(\theta) \sum_\gamma \boldsymbol{c}(\theta, a, \gamma)v(\gamma) \leq \frac{1}{1 + \delta_t}\widehat{u}_t(\theta) \sum_\gamma \boldsymbol{c}(\theta, a, \gamma)\widehat{v}_t(\gamma)$$

$$\leq \eta_t \widehat{u}_t(\theta) \sum_\gamma \boldsymbol{c}(\theta, a, \gamma)\widehat{v}_t(\gamma).$$

With respect to (25) and (26), we obtain that

$$J(\boldsymbol{\rho}_1) - \widehat{J}(\boldsymbol{\rho}_t, \mathcal{H}_t)$$

$$\leq J(\boldsymbol{\rho}_1) \cdot \left(1 - \eta_t + \frac{\max(\boldsymbol{\rho}_1 - \boldsymbol{\rho}_t, 0)}{\rho^{\min}}\right) + \|(u(\theta) - \widehat{u}_t(\theta))_{\theta \in \Theta}\|_1 + \|(v(\gamma) - \widehat{v}_t(\gamma))_{\gamma \in \Gamma}\|_1$$

$$= J(\boldsymbol{\rho}_1) \cdot \left(\delta_t + \frac{\max(\boldsymbol{\rho}_1 - \boldsymbol{\rho}_t, 0)}{\rho^{\min}}\right) + \|(u(\theta) - \widehat{u}_t(\theta))_{\theta \in \Theta}\|_1 + \|(v(\gamma) - \widehat{v}_t(\gamma))_{\gamma \in \Gamma}\|_1. \quad (31)$$

As we have already shown in the main part that

$$\mathbb{E}\left[\max(\boldsymbol{\rho}_1 - \boldsymbol{\rho}_t, 0)\right] \leq \begin{cases} O\left(k\dfrac{\sqrt{(t-2)\log T}}{T} + \sqrt{\dfrac{\log T}{T - t}} + \dfrac{k + n}{T}\right), & 2 \leq t \leq (T+1)/2; \\[4mm] O\left(k\sqrt{\dfrac{\log T}{T - t}} + \dfrac{k + n}{T}\right), & t > (T+1)/2. \end{cases}$$

it suffices for us to bound

$$\mathbb{E}[\|(u(\theta) - \widehat{u}_t(\theta))_{\theta \in \Theta}\|_\infty], \mathbb{E}[\|(u(\theta) - \widehat{u}_t(\theta))_{\theta \in \Theta}\|_1], \mathbb{E}[(v(\gamma) - \widehat{v}_t(\gamma))_{\gamma \in \Gamma}\|_1].$$

On this side, as we have shown that

$$\Pr\left[\|(u(\theta) - \widehat{u}_t(\theta))_{\theta \in \Theta}\|_1 \geq k\sqrt{\frac{\log T}{2(t - 1)}}\right] \leq \frac{k}{T},$$

$$\Pr\left[\|(v(\gamma) - \widehat{v}_t(\gamma))_{\gamma \in \Gamma}\|_1 \geq |\Gamma|\sqrt{\frac{\log T}{2(t - 1)}}\right] \leq \frac{|\Gamma|}{T},$$

it is natural that

$$\{\mathbb{E}[\|(u(\theta) - \widehat{u}_t(\theta))_{\theta \in \Theta}\|_\infty], \mathbb{E}[\|(u(\theta) - \widehat{u}_t(\theta))_{\theta \in \Theta}\|_1], \mathbb{E}[(v(\gamma) - \widehat{v}_t(\gamma))_{\gamma \in \Gamma}\|_1]\}$$

$$\leq O\left(k\sqrt{\frac{\log T}{t - 1}} + \frac{k}{T}\right).$$

Thus, putting all the above into (31) and summing from $t = 1$ to $T_0 \leq T$, we have

$$\mathbb{E}\left[\sum_{t=1}^{T_0} \left(J(\boldsymbol{\rho}_1) - \widehat{J}(\boldsymbol{\rho}_t, \mathcal{H}_t)\right)\right] = O(k\sqrt{T \log T} + n).$$

This concludes the proof.

# F  Details of Numerical Validations

Specifically, we consider the following instance: There are two types of resources, three types of contexts, and two types of external factors. The unknown mass function of context and external factor are $(u(\theta_1), u(\theta_2), u(\theta_3)) = (0.3, 0.3, 0.4)$ and $(v(\gamma_1), v(\gamma_2)) = (0.5, 0.5)$, respectively. The resource consumption is represented by

$$C^{(1)} = \begin{bmatrix} 0.9 & 1.1 \\ 1.8 & 2.2 \\ 1.2 & 0.8 \end{bmatrix}, \quad C^{(2)} = \begin{bmatrix} 2.1 & 1.9 \\ 0.8 & 1.2 \\ 0.9 & 1.1 \end{bmatrix},$$

where $\boldsymbol{c}(\theta_i, 1, \gamma_j) = (C_{i,j}^{(1)}, C_{i,j}^{(2)})^\top$ for all $(i, j)$. The reward function is represented by

$$R = \begin{bmatrix} 1.2 & 0.8 \\ 1.3 & 1.1 \\ 0.7 & 0.9 \end{bmatrix},$$

where $r(\theta_i, 1, \gamma_j) = R_{i,j}$ for all $i, j$. Thus the underlying LP is

$$J(\boldsymbol{\rho}) = \max 0.3 \cdot x_1 + 0.36 \cdot x_2 + 0.32 \cdot x_3,$$
$$\text{s.t.} \quad 0.3 \cdot x_1 + 0.6 \cdot x_2 + 0.4 \cdot x_3 \leq \boldsymbol{\rho}^1,$$
$$0.6 \cdot x_1 + 0.3 \cdot x_2 + 0.4 \cdot x_3 \leq \boldsymbol{\rho}^2,$$
$$0 \leq x_i \leq 1, \ i = 1, 2, 3,$$

where $x_i = \phi(\theta_i)$ for all $i = 1, 2, 3$. For a degenerate instance, we set $\boldsymbol{\rho} = (1, 1.15)^\top$ with the optimal solution $\boldsymbol{x}^* = (1, 0.5, 1)^\top$. For a non-degenerate problem instance, we set the average resources as $\boldsymbol{\rho} = (1, 1)^\top$ and the unique optimal solution is $\boldsymbol{x}^* = (2/3, 2/3, 1)^\top$; Here we relax the restriction that the consumption and reward are upper bounded by 1, as this condition can be met easily by scaling, with the regret accordingly scaling. Such a relaxation is to make the LP form more visually appealing.

# G  Missing Details in Appendix A

## G.1  The Density Estimator

We now present details on the kernel density estimator which we apply in Appendix A for approximating continuous distributions, which comes from Wasserman [40]. We consider a one-dimensional kernel function $K$ such that

- $\int K(x) \, \mathrm{d}x = 1$;
- $\int x^s K(x) \, \mathrm{d}x = 0, \quad \forall 1 \leq s \leq \beta$;
- $\int |x|^\beta |K(x)| \, \mathrm{d}x < \infty$.

Now, given $\ell$ independent samples $X_1, \cdots, X_\ell$ from $P$ and a positive number $h$ called the bandwidth, the kernel density estimator is defined as

$$\widehat{p}_\ell(x) = \frac{1}{\ell} \sum_{i=1}^{\ell} \frac{1}{h^d} K\left( \frac{\|x - X_i\|_2}{h} \right).$$

Furthermore, to satisfy Proposition A.1, we should choose $h \asymp k^{1/(2\beta+d)} \log k$ when $p \in \Sigma(\beta, L)$ is the density of $\mathcal{P}$ on $\mathbb{R}^d$.

## G.2  Proof of Theorem A.1

By (21), we know that

$$V^{\mathrm{FL}} - Rew = J(\boldsymbol{\rho}_1) \cdot \mathbb{E}\left[T - T_0\right] + \mathbb{E}\left[ \sum_{t=1}^{T_0} \left( J(\boldsymbol{\rho}_1) - \mathbb{E}_\theta \left[ \sum_{a \in A^+} \widehat{\phi}_t^*(\theta, a) R(\theta, a) \right] \right) \right],$$

and we bound these terms in order. For the expected stopping time $\mathbb{E}[T_0]$, by the analysis in Section 5, our goal turns into bounding $\max(\boldsymbol{\rho}_1 - \boldsymbol{\rho}_t, 0)$, which further by (16) and (29), reduces to bound $\boldsymbol{M}_\tau^C$. With continuous randomness, we have for any $i \in [n]$,

$$
(T - \tau) \left( \boldsymbol{M}_\tau^C \right)^i
$$

$$
= \boldsymbol{\rho}_\tau^i - \mathbb{E}_\theta \left[ \sum_{a \in A^+} \widehat{\phi}_\tau^*(\theta, a) \boldsymbol{C}^i(\theta, a) \right]
$$

$$
\overset{(a)}{\geq} \int_\theta \sum_{a \in A^+} \widehat{\phi}_\tau^*(\theta, a) \int_\gamma \boldsymbol{c}^i(\theta, a, \gamma) \widehat{v}_\tau(\gamma) \widehat{u}_\tau(\theta) \, \mathrm{d}\gamma \, \mathrm{d}\theta
$$

$$
- \int_\theta \sum_{a \in A^+} \widehat{\phi}_\tau^*(\theta, a) \int_\gamma \boldsymbol{c}^i(\theta, a, \gamma) v(\gamma) u(\theta) \, \mathrm{d}\gamma \, \mathrm{d}\theta
$$

$$
= \int_\theta \sum_{a \in A^+} \widehat{\phi}_\tau^*(\theta, a) \int_\gamma \boldsymbol{c}^i(\theta, a, \gamma) \widehat{v}_\tau(\gamma) (\widehat{u}_\tau(\theta) - u(\theta)) \, \mathrm{d}\gamma \, \mathrm{d}\theta
$$

$$
+ \int_\theta \sum_{a \in A^+} \widehat{\phi}_\tau^*(\theta, a) \int_\gamma \boldsymbol{c}^i(\theta, a, \gamma) (\widehat{v}_\tau(\gamma) - v(\gamma)) u(\theta) \, \mathrm{d}\gamma \, \mathrm{d}\theta
$$

$$
\overset{(b)}{\geq} - \sup_\theta |u(\theta) - \widehat{u}_\tau(\theta)| - \sup_\gamma |(v(\gamma) - \widehat{v}_\tau(\gamma)|.
$$

In the above, (a) is by the constraint feasibility of $\widehat{\phi}_\tau^*$, and (b) is because $\sum_{a \in A^+} \widehat{\phi}_\tau^*(\theta, a) \leq 1$ holds for any $\theta \in \Theta$. Further, by Proposition A.1, we have for $\tau = \Omega(1)$,

$$
\Pr \left[ \sup_\theta |u(\theta) - \widehat{u}_\tau(\theta)| \leq -\Theta \left( \sqrt{\log T} (\tau - 1)^{\alpha_u - 1} \right) \right] \leq \frac{1}{T^2},
$$

$$
\Pr \left[ \sup_\gamma |v(\theta) - \widehat{v}_\tau(\theta)| \leq -\Theta \left( \sqrt{\log T} (\tau - 1)^{\alpha_v - 1} \right) \right] \leq \frac{1}{T^2}.
$$

Thus, when $t = \Omega(1)$, we derive that with failure probability $O(n/T)$, it holds that

$$
\max \left( \boldsymbol{\rho}_1 - \boldsymbol{\rho}_t, 0 \right) \leq \Theta \left( \frac{1}{T - 1} + \sqrt{\log T} \sum_{\tau = \Theta(1)}^{t-1} \left( \frac{(\tau - 1)^{\alpha_u - 1}}{T - \tau} + \frac{(\tau - 1)^{\alpha_v - 1}}{T - \tau} \right) + \sqrt{\frac{\log T}{T - t}} \right).
$$

Further, for $p \in \{u, v\}$, when $t \leq (T + 1)/2$,

$$
\sum_{\tau = \Theta(1)}^{t-1} \frac{(\tau - 1)^{\alpha_p - 1}}{T - \tau} \leq \frac{2}{T - 1} \sum_{\tau = 2}^{t-1} (\tau - 1)^{\alpha_p - 1} \leq \frac{2(t - 2)^{\alpha_p}}{\alpha_p (T - 1)};
$$

and when $t > (T + 1)/2$, we have

$$
\sum_{\tau = \Theta(1)}^{t-1} \frac{(\tau - 1)^{\alpha_p - 1}}{T - \tau} \leq \frac{1}{\alpha_p} \left( \frac{2}{T - 1} \right)^{1 - \alpha_p} + \sum_{\tau = (T+1)/2}^{t-1} (T - \tau)^{\alpha_p - 2} \leq \frac{(T - t)^{\alpha_p - 1}}{1 - \alpha_p}.
$$

Thus, when $t = T - \Theta(\log^{(2(1 - \max\{1/2, \alpha_u, \alpha_v\}))^{-1}} T)$, we have

$$
\Theta \left( \frac{1}{T - 1} + \sqrt{\log T} \sum_{\tau = \Theta(1)}^{t-1} \left( \frac{(\tau - 1)^{\alpha_u - 1}}{T - \tau} + \frac{(\tau - 1)^{\alpha_v - 1}}{T - \tau} \right) + \sqrt{\frac{\log T}{T - t}} \right)
$$

$$
\leq \rho^{\min} - \frac{1}{T - t + 1},
$$

which leads to

$$
\mathbb{E}\left[ T - T_0 \right] = O \left( \log^{(2(1 - \max\{1/2, \alpha_u, \alpha_v\}))^{-1}} T \right).
$$

This concludes the analysis of the stopping time.

For the second part, By (24), we have

$$J(\boldsymbol{\rho}_1) - \mathbb{E}_\theta \left[ \sum_{a \in A^+} \widehat{\phi}_t^*(\theta, a) R(\theta, a) \right]$$

$$= \left( J(\boldsymbol{\rho}_1) - \widehat{J}(\boldsymbol{\rho}_t, \mathcal{H}_t) \right) + \left( \widehat{J}(\boldsymbol{\rho}_t, \mathcal{H}_t) - \mathbb{E}_\theta \left[ \sum_{a \in A^+} \widehat{\phi}_t^*(\theta, a) R(\theta, a) \right] \right).$$

On the second difference term, similar to the proof of Lemma E.2, we have

$$\widehat{J}(\boldsymbol{\rho}_t, \mathcal{H}_t) - \mathbb{E}_\theta \left[ \sum_{a \in A^+} \widehat{\phi}_t^*(\theta, a) R(\theta, a) \right]$$

$$= \int_\theta \sum_{a \in A^+} \widehat{\phi}_t^*(\theta, a) \int_\gamma r(\theta, a, \gamma) \widehat{v}_t(\gamma) \widehat{u}_t(\theta) \, d\gamma \, d\theta - \int_\theta \sum_{a \in A^+} \widehat{\phi}_t^*(\theta, a) \int_\gamma r(\theta, a, \gamma) v(\gamma) u(\theta) \, d\gamma \, d\theta$$

$$\leq \sup_\theta |u(\theta) - \widehat{u}_t(\theta)| + \sup_\gamma |(v(\gamma) - \widehat{v}_t(\gamma)|.$$

Thus, when $t = \Omega(1)$, by taking $\epsilon = 1/T$ in Proposition A.1 and (27), we arrive that

$$\mathbb{E} \left[ \widehat{J}(\boldsymbol{\rho}_t, \mathcal{H}_t) - \mathbb{E}_\theta \left[ \sum_{a \in A^+} \widehat{\phi}_t^*(\theta, a) R(\theta, a) \right] \right] = O \left( \sqrt{\log T} \left( (t-1)^{\alpha_u - 1} + (t-1)^{\alpha_v - 1} \right) + \frac{1}{T} \right).$$

We now focus on $J(\boldsymbol{\rho}_1) - \widehat{J}(\boldsymbol{\rho}_t, \mathcal{H}_t)$. We let

$$\delta_t := \frac{\sup_\theta |u(\theta) - \widehat{u}_t(\theta)| + \sup_\gamma |(v(\gamma) - \widehat{v}_t(\gamma)|}{\min_{\theta \in \Theta, a \in A^+} \{\min\{u(\theta) \boldsymbol{C}(\theta, a) > 0\}\}}.$$

We prove that

$$\widehat{u}_t(\theta) \int_\gamma \boldsymbol{c}^i(\theta, a, \gamma) \widehat{v}_t(\gamma) \, d\gamma \leq (1 + \delta_t) u(\theta) \int_\gamma \boldsymbol{c}^i(\theta, a, \gamma) v(\gamma) \, d\gamma$$

holds for any $(\theta, a, i)$ tuple, which is obvious if $\boldsymbol{c}^i(\theta, a, \gamma)$ is almost surely zero with respect to $\gamma$. Otherwise, we observe that

$$\widehat{u}_t(\theta) \int_\gamma \boldsymbol{c}^i(\theta, a, \gamma) \widehat{v}_t(\gamma) \, d\gamma - u(\theta) \int_\gamma \boldsymbol{c}^i(\theta, a, \gamma) v(\gamma) \, d\gamma$$

$$= (\widehat{u}_t(\theta) - u(\theta)) \int_\gamma \boldsymbol{c}^i(\theta, a, \gamma) \widehat{v}_t(\gamma) \, d\gamma + u(\theta) \int_\gamma \boldsymbol{c}^i(\theta, a, \gamma) (\widehat{v}_t(\gamma) - v(\gamma)) \, d\gamma$$

$$\leq \sup_\theta |u(\theta) - \widehat{u}_t(\theta)| + \sup_\gamma |(v(\gamma) - \widehat{v}_t(\gamma)|$$

$$\leq \delta_t u(\theta) \int_\gamma \boldsymbol{c}^i(\theta, a, \gamma) v(\gamma) \, d\gamma.$$

and thus, with $\eta_t := 1 - \delta_t \leq 1/(1 + \delta_t)$, we derive that

$$u(\theta) \int_\gamma \boldsymbol{c}^i(\theta, a, \gamma) v(\gamma) \, d\gamma \leq \frac{1}{1 + \delta_t} \widehat{u}_t(\theta) \int_\gamma \boldsymbol{c}^i(\theta, a, \gamma) \widehat{v}_t(\gamma) \, d\gamma$$

$$\leq \eta_t \widehat{u}_t(\theta) \int_\gamma \boldsymbol{c}^i(\theta, a, \gamma) \widehat{v}_t(\gamma) \, d\gamma.$$

This proves the above inequality. Thus, for an optimal solution $\phi_t^*$ of $J(\boldsymbol{\rho}_t)$, we see that $\eta_t \phi_t^*$ is a feasible solution of the programming $\widehat{J}(\boldsymbol{\rho}_t, \mathcal{H}_t)$. Thus, we notice that

$$\widehat{J}(\boldsymbol{\rho}_t, \mathcal{H}_t) \geq \eta_t \int_\theta \sum_{a \in A^+} \phi_t^*(\theta, a) \int_\gamma r(\theta, a, \gamma) \widehat{v}_t(\gamma) \widehat{u}_t(\theta) \, d\gamma \, d\theta$$

$$\geq \eta_t \int_\theta \sum_{a \in A^+} \phi_t^*(\theta, a) \int_\gamma r(\theta, a, \gamma) v(\gamma) u(\theta) \, \mathrm{d}\gamma \, \mathrm{d}\theta$$
$$- \eta_t (\sup_\theta |u(\theta) - \widehat{u}_t(\theta)| + \sup_\gamma |(v(\gamma) - \widehat{v}_t(\gamma)|)$$
$$= \eta_t J(\boldsymbol{\rho}_t) - (\sup_\theta |u(\theta) - \widehat{u}_t(\theta)| + \sup_\gamma |(v(\gamma) - \widehat{v}_t(\gamma)|).$$

With respect to (26), we obtain that

$$J(\boldsymbol{\rho}_1) - \widehat{J}(\boldsymbol{\rho}_t, \mathcal{H}_t)$$
$$\leq J(\boldsymbol{\rho}_1) \cdot \left( 1 - \eta_t + \frac{\max(\boldsymbol{\rho}_1 - \boldsymbol{\rho}_t, 0)}{\rho^{\min}} \right) + \sup_\theta |u(\theta) - \widehat{u}_t(\theta)| + \sup_\gamma |(v(\gamma) - \widehat{v}_t(\gamma)|$$
$$= J(\boldsymbol{\rho}_1) \cdot \left( \delta_t + \frac{\max(\boldsymbol{\rho}_1 - \boldsymbol{\rho}_t, 0)}{\rho^{\min}} \right) + \sup_\theta |u(\theta) - \widehat{u}_t(\theta)| + \sup_\gamma |(v(\gamma) - \widehat{v}_t(\gamma)|.$$

Now, when $t = \Theta(1)$, we have

$$\mathbb{E} \left[ \sup_\theta |u(\theta) - \widehat{u}_t(\theta)| \right] = O \left( \sqrt{\log T}(t-1)^{\alpha_u - 1} + \frac{1}{T} \right),$$
$$\mathbb{E} \left[ \sup_\gamma |v(\gamma) - \widehat{v}_t(\gamma)| \right] = O \left( \sqrt{\log T}(t-1)^{\alpha_v - 1} + \frac{1}{T} \right).$$

By the previous reasoning on $\max(\boldsymbol{\rho}_1 - \boldsymbol{\rho}_t, 0)$, we obtain that when $t = \Omega(1)$,

$$\mathbb{E} \left[ \max(\boldsymbol{\rho}_1 - \boldsymbol{\rho}_t, 0) \right]$$
$$\leq \Theta \left( \frac{1}{T-1} + \sqrt{\log T} \sum_{\tau=\Theta(1)}^{t-1} \left( \frac{(\tau-1)^{\alpha_u - 1}}{T - \tau} + \frac{(\tau-1)^{\alpha_v - 1}}{T - \tau} \right) + \sqrt{\frac{\log T}{T - t}} + \frac{n}{T} \right).$$

Therefore, summing from $t = 1$ to $T_0 \leq T$, we achieve that

$$\mathbb{E} \left[ \sum_{t=1}^{T_0} \left( J(\boldsymbol{\rho}_1) - \widehat{J}(\boldsymbol{\rho}_t, \mathcal{H}_t) \right) \right] = O \left( (T^{1/2} + T^{\alpha_u} + T^{\alpha_v}) \sqrt{\log T} + n \right).$$

Combining with previous bounds on $\mathbb{E}[T - T_0]$ and the estimation errors, we derive the theorem.

### G.3 Proof of Theorem A.2

Similar to the proof of Theorem 5.2, we concentrate on re-bounding the three terms under partial information feedback, respectively $\mathbb{E}[T - T_0]$, $\widehat{J}(\boldsymbol{\rho}_t, \mathcal{H}_t) - \mathbb{E}_\theta [\sum_{a \in A^+} \widehat{\phi}_t^*(\theta, a) R(\theta, a)]$, and $J(\boldsymbol{\rho}_1) - \widehat{J}(\boldsymbol{\rho}_t, \mathcal{H}_t)$. As for $\mathbb{E}[T - T_0]$, with Lemma 4.1, we argue here that the main term in bounding $\max(\boldsymbol{\rho}_1 - \boldsymbol{\rho}_t, 0)$ when $t = \Theta(T)$ becomes

$$\Theta \left( \sqrt{\log T} \left( \sum_{\tau=\Theta(1)}^{t-1} \frac{(\tau-1)^{\alpha_u - 1}}{T - \tau} + \sum_{\tau=\Theta(1)}^{\Theta(T)} \frac{((\tau-1)/\log T)^{\alpha_v - 1}}{T - \tau} + \sum_{\tau=\Theta(T)}^{t-1} \frac{(\tau-1)^{\alpha_v - 1}}{T - \tau} \right) \right).$$

Consequently, when $t$ is close to $T$, we have with failure probability $O(1/T)$,

$$\max(\boldsymbol{\rho}_1 - \boldsymbol{\rho}_t, 0)$$
$$\leq \Theta \left( \frac{1}{T-1} + \sqrt{\log T} \left( (T-t)^{\alpha_u - 1} + (T-t)^{-1/2} \right) + (T-t)^{\alpha_v - 1} \log^{3/2 - \alpha_v} T \right).$$

This leads to
$$\mathbb{E}[T - T_0] = O \left( \log^{\max(1, 1/(2 - 2\alpha_u), (3 - 2\alpha_v)/(2 - 2\alpha_v))} T \right).$$

For the estimation error term $\widehat{J}(\boldsymbol{\rho}_t, \mathcal{H}_t) - \mathbb{E}_\theta[\sum_{a \in A^+} \widehat{\phi}_t^*(\theta, a) R(\theta, a)]$, when $\Omega(1) \le t \le \Theta(T)$, the bound now becomes

$$\mathbb{E}\left[\widehat{J}(\boldsymbol{\rho}_t, \mathcal{H}_t) - \mathbb{E}_\theta\left[\sum_{a \in A^+} \widehat{\phi}_t^*(\theta, a) R(\theta, a)\right]\right]$$

$$= O\left(\sqrt{\log T}(t-1)^{\alpha_u - 1} + \log^{3/2 - \alpha_v} T \cdot (t-1)^{\alpha_v - 1} + \frac{1}{T}\right).$$

At last, for $J(\boldsymbol{\rho}_1) - \widehat{J}(\boldsymbol{\rho}_t, \mathcal{H}_t)$, we derive that

$$\mathbb{E}\left[\sum_{t=1}^{T_0} \max(\boldsymbol{\rho}_1 - \boldsymbol{\rho}_t, 0)\right] = O\left(\sqrt{\log T}\left(T^{\alpha_u} + T^{1/2}\right) + \log^{3/2 - \alpha_v} T \cdot T^{\alpha_v} + n\right),$$

$$\mathbb{E}\left[\sum_{t=1}^{T_0} \sup_\theta |u(\theta) - \widehat{u}_t(\theta)|\right] = O\left(\sqrt{\log T} \cdot T^{\alpha_u}\right),$$

$$\mathbb{E}\left[\sum_{t=1}^{T_0} \sup_\gamma |v(\gamma) - \widehat{v}_t(\gamma)|\right] = O\left(\log^{3/2 - \alpha_v} T \cdot T^{\alpha_v}\right).$$

Putting together, we obtain that

$$\mathbb{E}\left[\sum_{t=1}^{T_0} \left(J(\boldsymbol{\rho}_1) - \widehat{J}(\boldsymbol{\rho}_t, \mathcal{H}_t)\right)\right] = O\left(\sqrt{\log T}\left(T^{\alpha_u} + T^{1/2}\right) + \log^{3/2 - \alpha_v} T \cdot T^{\alpha_v} + n\right).$$

Synthesizing all the above, we finish the proof of the theorem.

