# OpenReview forum: "Contextual Decision-Making with Knapsacks Beyond the Worst Case"
_NeurIPS.cc/2024/Conference — NeurIPS 2024 poster_

### Official Review · Reviewer_ZSDZ · 2024-06-19

**Soundness:** 3
**Presentation:** 3
**Contribution:** 2
**Rating:** 5
**Confidence:** 3

**Summary:**

This paper considers the problem of dynamic decision making with resource constraints. They show that under certain conditions they can achieve O(1) regret with respect to the time horizon. However, I am not sure whether this is true regret (i.e. with respect to the best policy in hindsight) or regret with respect to a number $V^{FL}$. My final score is highly dependent on the author’s response to this question.

**Strengths:**

I am wondering if in Theorem 3.1 the display equation should say $V^{ON}$ instead of $V^{FL}$. If it is indeed $V^{FL}$ then your goal is only comparing to $V^{FL}$ so what is the point of the “worst case” discussion comparing $V^{FL}$ to $V^{ON}$? If is is meant to be $V^{ON}$ in Theorem 3.1 then I think that this is an interesting result. However, if it is indeed $V^{FL}$ then the result is much weaker as you have not shown any improvement over $O(\sqrt{T})$ with respect to the optimal policy. My final score will be highly dependent on your answer to this question.

**Weaknesses:**

If Theorem 3.1 is written correctly then please see the “strengths” section for a major weakness.

In Theorem 3.1 I believe that the O(1) must be hiding problem-dependent constants. I don’t just mean constants like the sizes of the sets but constants that are based on the linear program itself. I.e. are there some non-degenerate linear programs in which the hidden constant factor blows up arbitrarily high - e.g. when we limit to degeneracy. This is the same concept as in stochastic bandits: the $O(Kln(T))$ factor hides the gap between the mean rewards of the optimal and second-optimal arms - which limits to infinity as the gap limits to zero (although the regret actually never goes above $O(\sqrt{KT})$). You should certainly include any problem-dependent constants in your bound (although I do understand they may be complex to bound - so at least point out that they exist (if they do)).

Line 226 (and line 78): You say that any LP can easily escape from degeneracy. Doe this mean that you can improve on the sqrt{T} bound when $J(\rho_1)$ is degenerate? Otherwise this statement (which appears twice) is highly misleading.

Line 171: Theorem 2.1 does not show an \Omega(\sqrt{T}) regret lower bound when you define the regret relative to $V^{FL}$.

**Questions:**

Can you give examples to show that the non-degeneracy of $J(\rho_1)$ is common.

---

> ### Author Rebuttal · Authors · 2024-07-31
>
> Thanks for your review and comments.
> We now respond to your concerns and questions.
>
> **Theorem 3.1.**
> It is *completely correct* for you to say that in the equation of Theorem 3.1, $V^{\mathrm{FL}}$ could be replaced by $V^{\mathrm{ON}}$.
> In other words, we are talking about the *true regret*, and we indeed obtain an $O(1)$ true regret under Assumption 3.1.
> This is also true for *all* our regret results, including Theorems 4.1, 5.1, 5.2, B.1, and B.2.
> We hope that our claim here could eliminate your concerns.
>
> Nevertheless, we should notice that from Proposition 2.1, we know that $V^{\mathrm{FL}} \geq V^{\mathrm{ON}}$, and of course, we have $V^{\mathrm{ON}} \geq Rew$.
> Thus, the formulae we write in this paper are, in fact, stronger in the sense that if $V^{\mathrm{FL}} - Rew = O(f(T))$, then $V^{\mathrm{ON}} - Rew = O(f(T))$ also holds.
> For Theorem 2.1, we only want to know how far $V^{\mathrm{FL}}$ could be above $V^{\mathrm{ON}}$ at worst and the corresponding conditions so as to make our work more complete.
>
> **Problem-dependent constants in Theorem 3.1.**
> It is true that our regret bound consists of problem-dependent variables, as we discussed in *Lines 233--258*.
> More specifically, as you mentioned, we have a constant $D$ in the bound based on the linear program $J(\rho_1)$.
> This constant represents the minimum $L_\infty$ distance between the LP $J(\rho_1)$ and any LP with multiple optimal solutions or a degenerate optimal solution (Lines 237--238).
> When our CDMK problem degenerates to stochastic decision-making by removing the context and resource constraints, $D$ is precisely *half the gap between the mean rewards of the best and second-best arm*.
> Thus, $D$ resembles the reward-gap-like parameter in the multi-armed bandit literature (Lines 252--253).
> And our bound goes quadratically with $1/D$ (Line 258, with a typo, we will correct that.).
> We will work on to make our arguments clearer.
>
> **Escaping from degeneracy.**
> In fact, we are not saying that we can improve the $O(\sqrt{T})$ regret under degeneracy.
> We mean that in practice, we can prevent the real-world problem from $J(\rho_1)$ being degenerate by minorly changing the resource constraints, e.g., neglecting some of the resources.
> We will show in the later response that non-degeneracy for $J(\rho_1)$ is common.
> We will clarify this part, of course, to eliminate any misleading information. Thanks a lot for pointing that out.
>
> **Regret lower-bound.**
> You are correct.
> We will surely clarify these arguments by clearly differentiating the "true regret" and the regret we define in this work's main body.
>
> **Non-degeneracy for $J(\rho_1)$ is common.**
> We simplify the example that we use in our numeric validations (Appendix G) by disregarding the third context, renormalizing, and letting the first resource constraint be $0.75$ and the second be $y$. Then the LP $J(\rho_1)$ is now maximizing $0.3x_1 + 0.36x_2$, under the constraint that $0.3x_1 + 0.6x_2 \leq 0.75$, $0.6x_1 + 0.3x_2\leq y$, and $0\leq x_1, x_2\leq 1$.
> This LP always has a unique solution, and the *only* value $y > 0.6$ such that $J(\rho_1)$ is degenerate is $y = 0.825$, with the degenerate solution being $(x_1, x_2) = (1, 0.75)$.
> That is to say, if we suppose that $y$ is uniformly located between 0.6 and 1, for example, then $J(\rho_1)$ is almost surely non-degenerate.
> I hope this can address your question, and this is the reason why we say that any LP can easily "escape" from being degenerate.
> Nevertheless, we will correct the misleading sentences.

---

> ### Author Response · Authors · 2024-08-12
>
> Hi, we are looking forward to seeing your opinion on our rebuttal!

---

> > ### Comment · Reviewer_ZSDZ · 2024-08-12
> >
> > Yes - sorry - of course you can replace $V^{\mathrm{FL}}$ with $V^{\mathrm{ON}}$ - I was thinking "losses" rather than "rewards". I am therefore increasing my score - but I feel strongly that, if this paper is accepted, you should include all problem-dependent constants in Theorem 3.1

---

> > > ### Author Response · Authors · 2024-08-12
> > >
> > > Thanks a lot! We will surely improve our manuscript on noting more about problem-dependent constants.

---

### Official Review · Reviewer_TgME · 2024-06-28

**Soundness:** 3
**Presentation:** 2
**Contribution:** 3
**Rating:** 5
**Confidence:** 2

**Summary:**

This paper studies contextual decision making with Knapsack constraints assuming that the requests and the contexts follows some distributions. It studies the full information setting when each context is revealed after the decision is made and the partial information setting when the context is revealed only when a non-null decision is taken. The paper provides regret bounds under various settings.

**Strengths:**

Under the unique non degenerate LP assumption, the paper has provided regret bounds that are better than the worst cases in both full information and partial information setting, which is novel. It further shows that without this assumption, the regret bound is indeed $O(\sqrt{T})$ that matches the lower bounds. The paper further provides regret bounds when the request and context are continuously distributed. It greatly enhances the understanding of the problem.

**Weaknesses:**

1. I found the presentation can be further improved, particularly about why the unique non-degenerate solution can greatly improve the regret bound and provide a proof sketch. This is the main contribution of the work yet the intuition is not fully clear.
2. See questions below.

**Questions:**

1. In the motivating example, it is unclear why for the dynamic bidding problem, the highest competing bid would follow a distribution. It seems to be more adversarially chosen.
2. In the partial information setting, since choosing a null action will not be able to observe the context, does the $\hat v_t(\gamma)$ still provide an unbiased empirical estimator of the distribution of $\gamma$? Since the decision depends on previous observed contexts, observing the next context or not depends on the previous observations. Thus $\hat v_t(\gamma)$ does not seem to be an unbiased empirical estimator anymore. What are the techniques used to address such bias?
3. Since the context $\gamma$ is revealed after the decision is made, is it worthy to take more non-null decisions in the early stages and try to learn the distribution faster?
4. The relationship of the proposed problem comparing to the NRM problem is not explicitly discussed. Please compare the similarities and differences, particularly regarding the measurements.
5. The linear programming is only solvable when $\theta$ and $a$ admit finite support. What if they are continuous variables? Continuous variables are quite common in both inventory and bidding situation.

I am happy to raise the score if these issues are well addressed.

**Limitations:**

Comparison to continuous optimization methods in NRM problems are not discussed.

---

> ### Author Rebuttal · Authors · 2024-07-31
>
> Thanks for your comments and questions. We will now answer them.
>
> **Why uniqueness and non-degeneracy is important.**
> The corresponding discussion is located in *Lines 233--243* in our paper.
> In short words, under this assumption (Assumption 3.1), we have the stability property that if all estimated parameters in $\hat{J}(\rho_t, \mathcal{H}_t)$ are not far away from the true parameters, then $\hat{J}(\rho_t, \mathcal{H}_t)$ and $J(\rho_1)$ have the same set of basic variables and binding constraints, respectively.
> This property is given in Lemma D.2 in the appendix and plays a crucial role in our analysis.
> We will work on to make that clearer.
>
> **Stochastic highest competing bids in the dynamic bidding problem.**
> This is a good question.
> Suppose we stand in the view of a single bidder and consider a large auction market.
> In this scenario, even if each of the other bidders bid strategically, their highest bid can still be regarded as being stochastic due to the large market.
> This is sometimes referred to as the "mean-field approximation" in economics.
> In the related literature, such a stochastic assumption on the highest competing bids is also widely taken, for example, Balseiro and Gur (2019), Feng, Padmanabhan, and Wang (2023), and Chen et al. (2024).
>
> **Unbiased estimation of $\hat{v}_t$.**
> This is an important question.
> In fact, $\hat{\mathcal{V}}_t$ is still an unbiased estimation of $\mathcal{V}$ even under partial feedback.
> The reasoning here is that $\gamma_t$ is drawn independently with the context $\theta_t$, and the action $a_t$ is also chosen before $\gamma_t$ is revealed.
> Thus, we have that $\Pr[\gamma_t | \theta_t, a_t] = \Pr[\gamma_t]$, or that $\gamma_t$ is independent with $(\theta_t, a_t)$.
> This guarantees that $\hat{\mathcal{V}}_t$ is unbiased.
>
> **More non-null decisions in the beginning.**
> This is a very good thought experiment.
> However, although it seems delightful to make more non-null decisions initially, we have to ensure that we are not "exploring" too much or that we could be too far from the optimal and may not "rescue back" in the later time slots.
> In fact, our key ingredient for the proof is Lemma 4.1, which indicates that under the re-solving heuristic, we explore in every $O(\log T)$ rounds and have an $O(1)$ overall exploring frequency in the beginning.
> (Please also refer to Figure 1.)
> This is already asymptotically much efficient for getting samples.
> You raised a very important question, and we will see if we can do better by slightly modifying the algorithm and obtaining a better regret bound.
>
> **Relationship between CDMK and NRM.**
> Due to the space limit, we defer the discussion to Appendix A, Lines 499--512.
> Put simply, NRM is a sub-problem of our CDMK problem with no external factors and only a binary action space $A = \{0, 1\}$, where $0$ is the null action and $1$ is the "accept" action.
> Therefore, our CDMK is generally "harder" than the NRM problem.
> As for the measurements, our fluid benchmark is identical to the deterministic LP (DLP) benchmark introduced in classical works on NRM, e.g., Jasin and Kumar (2012) and Balseiro, Besbes, and Pizarro (2023).
> Thus, we believe that our work is built on a strong literature basis.
>
> **Continuous variables.**
> This is also an important question.
> In Appendix B, we suppose that there is an outside oracle that can help us solve the programming in each round for continuous variables.
> Yet, this could be hard in practice if we are not facing a convex continuous programming.
> Related works in the literature would mostly suppose a finite context and external factor set or similarly suppose an oracle for continuous variables, e.g., Balseiro, Besbes, and Pizarro (2023).
> Also, in real-life scenarios, e.g., inventory or bidding, even though the action space is large, it is still finite since the allocation/bid is usually required to be a multiple of a minimum unit.
>
> **Reference:**
> Balseiro, S. R., & Gur, Y. (2019). Learning in repeated auctions with budgets: Regret minimization and equilibrium. Management Science, 65(9), 3952-3968.
> Feng, Z., Padmanabhan, S., & Wang, D. (2023, April). Online Bidding Algorithms for Return-on-Spend Constrained Advertisers. In Proceedings of the ACM Web Conference 2023 (pp. 3550-3560).
> Chen, Z., Wang, C., Wang, Q., Pan, Y., Shi, Z., Cai, Z., ... & Deng, X. (2024, March). Dynamic budget throttling in repeated second-price auctions. In Proceedings of the AAAI Conference on Artificial Intelligence (Vol. 38, No. 9, pp. 9598-9606).
> Jasin, S., & Kumar, S. (2012). A re-solving heuristic with bounded revenue loss for network revenue management with customer choice. Mathematics of Operations Research, 37(2), 313-345.
> Balseiro, S. R., Besbes, O., & Pizarro, D. (2023). Survey of dynamic resource-constrained reward collection problems: Unified model and analysis. Operations Research.

---

> > ### Comment · Reviewer_TgME · 2024-08-08
> > **Acknowledgement**
> >
> > Thanks for your responses! It addresses my questions well. I will raise the score to 5.

---

> > > ### Author Response · Authors · 2024-08-12
> > >
> > > Thanks a lot!

---

### Official Review · Reviewer_m7z3 · 2024-07-07

**Soundness:** 3
**Presentation:** 3
**Contribution:** 3
**Rating:** 6
**Confidence:** 3

**Summary:**

This paper studies an online contextual optimization problem with resource constraints. The paper provides a sufficient condition (worst case condition) under which the fundamental limit on the regret bound is reached. The paper further provides an algorithm that can achieve \tilde O(1) regret when the worst case condition does not hold. Numerical results are also provided to validate the theory.

**Strengths:**

The paper is well written. The relation between the worst case condition and the degeneracy of linear constraints is novel and inspiring. The intuition behind the proposed algorithm is clearly explained.

**Weaknesses:**

1.  Is $T$ known beforehand? If $T$ is unknown, how to compute the leftover budget $\rho_t$? If $T$ is known, how to design an algorithm with unknown $T$?

2. What's the difference between Rew and V^{on}?

3. if there is $\rho^i$ amount of resources per stage, why does the problem formulation only respect the resource constraint at stage $T$, instead of considering resource constraint at every stage $t$, i.e. total resources available to be used in stages $t$ <= $\rho^i t$ for all $t\geq T$?

4. In the simulation, how does the proposed algorithm compare with the existing algorithms proposed for this problem?

**Questions:**

1. It is slightly confusing to use $\rho^i$ to denote the $i$th entry since it also means exponent.

---

> ### Author Rebuttal · Authors · 2024-07-31
>
> Thanks for appreciating our work.
> We will now answer your questions.
>
> **Known $T$.**
> $T$ is known beforehand in this work, which is a common assumption in the literature, as supposed by the survey of Balseiro, Besbes, and Pizarro (2023).
> We have not yet considered the problem of unknown $T$, which is certainly an important future direction.
>
> **$Rew$ and $V^{\mathrm{ON}}$.**
> $Rew$ is the expected total reward of *our algorithm*, while $V^{\mathrm{ON}}$ is the expected total reward of the *optimal algorithm*.
>
> **Resource constraint for each stage.**
> This is a very interesting question.
> In practice, only the global constraint is taken since we usually have an initial inventory of resources and only require that all the allocations should be done concerning this initial inventory.
> Thus, a resource constraint for the later stages is unnecessary.
> Considering resource constraints for every stage will make the problem much more difficult, as an algorithm cannot sufficiently explore all actions in the beginning stages.
> This is certainly a future extension of our model.
>
> **Simulation.**
> In this work, we do not compare our algorithm with other existing algorithms mainly because CDMK with partial feedback is a new model.
> Further, the regret results highly rely on instance-dependent factors for both existing algorithms and ours.
> Altogether, it is a bit unfair to conduct such a comparison under only some instances since they could be biased.
> Our simulation results mainly aim to justify our theoretical regret bounds, and we do see a match there.
>
> **Usage of $\rho^i$.**
> We are sorry for causing such confusion. We will work on improving the notation.
>
> **Reference:**
> Balseiro, S. R., Besbes, O., & Pizarro, D. (2023). Survey of dynamic resource-constrained reward collection problems: Unified model and analysis. Operations Research.

---

> > ### Comment · Reviewer_m7z3 · 2024-08-08
> >
> > Thanks for your responses! I will keep my score.

---

> > > ### Author Response · Authors · 2024-08-12
> > >
> > > Thanks a lot!

---

### Official Review · Reviewer_c8Hc · 2024-07-11

**Soundness:** 3
**Presentation:** 3
**Contribution:** 3
**Rating:** 6
**Confidence:** 3

**Summary:**

This paper considers a new contextual decision-making model with knapsack constraints, which is highly related to the CBwK setting but features a different information feedback structure. Under this model, the authors nearly characterize the conditions under which $\tilde{O}(1)$ regret can be achieved based on the degeneracy of the optimal solution of the fluid LP. Specifically, the re-solving heuristic is proposed to achieve $\tilde{O}(1)$ regret under these conditions. Additionally, the $\tilde{O}(\sqrt{T})$ worst-case regret of the algorithm is provided.

**Strengths:**

1. The model considered in the paper is quite general to cover several interesting problems.

2. The results provided under the proposed model is relative sharp and complete.

**Weaknesses:**

1. The assumption on the randomness is stronger than those made in the contextual decision-making literature.

2. While I understand that this paper considers a new setting and that the works in CBwK are the most related, it is acceptable that the authors mainly compare their results with those in CBwK. However, since the information feedback assumption in this paper is strictly stronger than in CBwK, I suggest the authors provide more comments regarding this difference to make the comparison fairer.

**Questions:**

Both the feedback structure and the degeneracy-based condition for breaking $O(\sqrt{T})$ regret make me think it is similar to the corresponding results in online linear programming [1,2,3]. While several structures are different (e.g., the context and the action set), are there any high-level heuristics that can explain such similarity? Moreover, [3] provides a condition for breaking $O(\sqrt{T})$ regret beyond degeneracy. Can the idea in [3] be utilized in this setting to further characterize the conditions for breaking $O(\sqrt{T})$ regret in this paper?



Ref:

[1] Arlotto A, Gurvich I. Uniformly bounded regret in the multisecretary problem[J]. Stochastic Systems, 2019, 9(3): 231-260.

[2] Bray R L. Logarithmic Regret in Multisecretary and Online Linear Programs with Continuous Valuations[J]. arXiv preprint arXiv:1912.08917, 2019.

[3] Jiang J, Ma W, Zhang J. Degeneracy is ok: Logarithmic regret for network revenue management with indiscrete distributions[J]. arXiv preprint arXiv:2210.07996, 2022.

**Limitations:**

N.A.

---

> ### Author Rebuttal · Authors · 2024-07-31
>
> Thanks for your kind comments and suggestions.
> We now respond to your concerns and questions.
>
> **Stronger assumptions on the randomness.**
> Indeed, our model requires an explicit form of randomness $\gamma$ for the external factor.
> This explicit model is crucial for us in this work to learn its distribution.
> We will work on to see how to relax this assumption.
>
> **Comparison with CBwK.**
> Due to the space limit, we have to delay the comparison between our model and CBwK to Appendix A, Lines 469--498.
> There, we emphasize the difference between our results and results in CBwK, as well as the difference in feedback models.
> In short, although our full/partial feedback models are stronger than the bandit feedback model, techniques in the literature for CBwK would require assumptions on the problem structure, e.g., linear dependence between expected rewards/cost vectors and the action (linear CBwK).
> With these model assumptions, existing regression modules can help to address the problem of learning unknown parameters.
> In our work, we do not take any of these simplifying assumptions and no learning oracles (Line 492); therefore, we would require a stronger feedback model to ensure the efficient learning of randomness.
>
> In fact, we give a novel, important, and non-trivial analysis for the learning guarantee under partial feedback (Lemma 4.1), which is crucial for our regret results in this model.
> Lemma 4.1 itself could also be of independent interest.
>
> **Degeneracy-based conditions, and other heuristics.**
> This is a very interesting topic.
> In general, we believe that non-degeneracy is a critical factor to an $O(\sqrt{T})$ regret bound in the CDMK/CDwK problems, at least for the re-solving technique we use in this work (e.g., Jasin and Kumar (2012)), as pointed out by Bumpensanti and Wang (2020).
>
> To break the $O(\sqrt{T})$ regret without the degeneracy assumption, we should explore other heuristics that are already discovered in the online linear programming problem, as you mentioned, and see whether other conditions could be characterized for breaking the $O(\sqrt{T})$ regret.
> Despite this, our work is an important first trial of extending the re-solving technique to the CDMK problem.
>
> **Reference:**
> Jasin, S., & Kumar, S. (2012). A re-solving heuristic with bounded revenue loss for network revenue management with customer choice. Mathematics of Operations Research, 37(2), 313-345.
> Bumpensanti, P., & Wang, H. (2020). A re-solving heuristic with uniformly bounded loss for network revenue management. Management Science, 66(7), 2993-3009.

---

> > ### Comment · Reviewer_c8Hc · 2024-08-12
> >
> > Thank you for your response. I will keep my score positive.

---

> > > ### Author Response · Authors · 2024-08-12
> > >
> > > Thanks a lot!

---

> ### Author Response · Authors · 2024-08-12
>
> Hi, we are looking forward to seeing your opinion on our rebuttal!

---

### Decision · Program_Chairs · 2024-09-25

**Decision:**

Accept (poster)

**Comment:**

This paper studies the problem of maximizing the total reward under resource constraints in the online setting. The paper establishes an $\Omega(\sqrt{T})$ lower-bound on the worst-case regret. Under a non-degeneracy assumption (that helps avoid the worst-case scenario), the paper proposes an algorithm that achieves $O(1)$ regret in the full information setting, and $O(\log(T))$) regret in the partial information setting.

The reviewers agree that the paper is well-written, and its contributions merit acceptance. After carefully reading the paper and the corresponding discussion, I tend to agree. Please incorporate the reviewers' feedback. In particular, addressing the following concerns will help strengthen the paper:
- Move the comparison to the CBwK from the Appendix to the main paper (response to Rev. c8Hc)
- Add a more detailed comparison to the papers (pointed by Rev. c8Hc) that break the $O(\sqrt{T})$ lower-bound without the non-degeneracy assumption.
- More intuition/explanation on why uniqueness and non-degeneracy help break the $O(\sqrt{T})$ lower-bound (response to Rev. TgME).
- Move the comparison to NRM from Appendix A to the main paper  (response to Rev. TgME).
- Include problem-dependent constants in the theorems, and explain their dependence (response to Rev. ZSDZ)